

# Annual to seasonal glacier mass balance in High Mountain Asia derived from Pléiades stereo images: examples from the Pamir and the Tibetan Plateau

Daniel Falaschi[1,2], Atanu Bhattacharya[3], Gregoire Guillet[4], Lei Huang[5], Owen King[1], Kriti Mukherjee[6], Philipp Rastner[7], Tandong Yao[8], Tobias Bolch[1]

[1]School of Geography and Sustainable Development, University of St Andrews, Irvine
Building, North Street St Andrews KY16 9AL, Scotland, UK.
[2]Instituto Argentino de Nivología, Glaciología y Ciencias Ambientales (IANIGLA), CCT-
CONICET Mendoza, C.C. 330, 5500 Mendoza, Argentina
[3]Department of Earth Sciences and Remote Sensing, JIS University, Kolkata, India-700109
[4]Civil and Environmental Engineering, University of Washington, Seattle, WA, USA
[5]Aerospace Information Research Institute, Chinese Academy of Sciences, Dengzhuang south
road 9, Haidian District, Beijing, China-100094
[6]Cranfield Environment Centre, Cranfield University, College Road, Bedford, MK43 0AL,
UK
[7]Department of Geography, University of Zurich, Switzerland
[8]Institute of Tibet Plateau Research, Chinese Academy of Sciences, Beijing, China

*Correspondence to*: Daniel Falaschi (dfalaschi@mendoza-conicet.gob.ar)

**Abstract.** Glaciers are crucial sources of freshwater in particular for the arid lowlands surrounding High Mountain
Asia. In order to better constrain glacio-hydrological models, annual, or even better, seasonal information about
glacier mass changes is highly beneficial. In this study, we test the suitability of very high-resolution Pleiades
DEMs to measure glacier-wide mass balance at annual and seasonal scales in two regions of High Mountain Asia
(Muztagh Ata in Eastern Pamir and parts of Western Nyainqêntanglha, South-central Tibetan Plateau), where
recent estimates have shown contrasting glacier behavior. We find that the average annual mass balance in
Muztagh Ata between 2020 and 2022 was -0.11 ±0.21 m w.e. a$^{-1}$, suggesting the continuation of a recent phase of
slight mass loss following a prolonged period of balanced mass budgets previously observed. The mean annual
mass balance in Western Nyainqêntanglha for the same period was highly negative (-0.60 ±0.15 m w.e. a$^{-1}$ on
average), suggesting increased mass loss rates. The 2022 winter (+0.21 ±0.24 m w.e.) and summer (-0.31 ±0.15
m w.e.) mass budgets in Muztag Ata and Western Nyainqêntanglha (-0.04 ±0.27 m w.e. [winter]; -0.66 ±0.07 m
w.e. [summer]) suggest winter and summer accumulation-type regimes, respectively. We support our findings by
implementing a Sentinel-1–based Glacier Index to identify the firn and wet snow areas on glaciers and characterize
accumulation type. The good match between the geodetic and Glacier Index results demonstrates the potential of
very high-resolution Pleiades data to monitor mass balance at short time scales and improves our understanding
of glacier accumulation regimes across High Mountain Asia.


## 1 Introduction

Fluctuations of glaciers across High Mountain Asia are at the core of both scientific and public debate due to their
major relevance as sources of freshwater for human consumption, their regulatory role of river runoff, and their
contribution to sea level rise (Immerzeel et al. 2020; Vishwakarma et al., 2022; Yao et al., 2022). The combination
of varying climatic and accumulation regimes (Yao et al., 2012; Huang et al., 2022), debris-cover fraction
(Scherler et al., 2018) glacier surges (Farinotti et al., 2020; King et al., 2021; Guillet et al., 2022), presence of
supra-glacial and proglacial lakes (Brun et al. 2019; Maurer et al., 2016; King et al., 2019), and inherent dynamic
factors interact with ongoing climate change (Armstrong et al. 2021) and result in spatially and temporally variable
mass loss rates (Brun et al, 2019; Dehecq et al., 2019). In recent decades, however, consistent increases in glacier
wastage have been observed throughout most of High Mountain Asia (Bhattacharya et al., 2021, Hugonnet et al.
2021). The most notable exception to this trend is the glaciers within the 'Pamir-Karakorum' anomaly, which also
extends into Eastern Pamir and the Western Kunlun Shan mountains (Kääb et al., 2015, Liang et al., 2022). In this
region, glaciers have been in balance or have had slight mass gains since at least the 1970s (Kääb et al., 2015;



Bolch et al., 2017, 2019; Brun et al., 2017; Berthier and Brun, 2019). However, recent research hints to an end of this anomaly (Hugonnet et al. 2021; Bhattacharya et al., 2021). Monitoring glacier changes from both remotely–sensed and in situ observations has thus been fundamental for an improved understanding of the relation between climate change and glaciers and has yielded solid evidence of glacier mass loss worldwide (Zemp et al., 2015, Hugonnet et al., 2021). Glacier mass budget is the summation of accumulation and ablation over a specific period of time (Cogley et al., 2011). Presently, the two most widely used methods for determining glacier mass balance are the glaciological and geodetic methods (Cogley, 2009). The glaciological method measures surface mass balance, typically on a seasonal scale, using a combination of conveniently distributed snow pits and stakes to measure accumulation and ablation, respectively (Cogley, 2009). Because of the high costs and logistical constraints involved, less than 0.1% of glaciers spread across the many glacierized mountains in the World have long-term glaciological records that span more than a decade, whilst ~170 glaciers have currently active in situ glaciological mass balance observations (Maussion et al., 2019; WGMs, 2021). On the other hand, the geodetic method consists essentially in the differentiation of multi-temporal and often multi-sourced elevation data (Digital Elevation Models -DEMs), covering usually longer time intervals (>5 years) and larger regions compared to the glaciological method (e.g. Braun et al, 2019; Dussaillant et al., 2019; Shean et al, 2019; Davaze et al., 2020; Hugonnet et al., 2021). In contrast to the glaciological method, the geodetic approach does not only measure the surface mass balance, but also englacial and subglacial processes (Zemp et al., 2013; Andreassen et al., 2016). Moreover, although the glaciological and geodetic methods provide independent observations of glacier mass balance, geodetic surveys of variable temporal resolution can potentially serve to validate and calibrate glaciological records (Zemp et al., 2013; Xu et al., 2018; Wagnon et al., 2021).

When shorter time intervals are considered, the geodetic method reveals two fundamental limitations: 1) the DEM precision in relation to the magnitude of the elevation change signal during an annual to seasonal time interval, and 2) the uncertainties of the snow, ice and firn density and densification linked to the volume to mass conversion factor, which are also high over short timescales (Huss, 2013; Pelto et al., 2019). This information is, as noted, only available for a restricted glacier sample worldwide.

The increased availability of sub-meter, very-high resolution (VHR) satellite imagery (e.g. Pléiades, WorldView-2) and associated DEMs allows for the quantification of small scale, low magnitude (meter scale) changes at the Earth's surface whilst retaining a suitable level of precision (Berthier et al., 2014). Testing of these DEMs over several mountain sites across the world has confirmed accuracies ranging between 0.2 to 1 m (Berthier et al., 2014; Shean et al., 2016), indicating high potential for assessing glacier elevation changes over short (<3 yrs) time intervals. Glaciers with relatively high mass balance amplitude and heavily snowbound areas are particularly suited to this purpose (Belart et al., 2017, Deshamps-Berger et al., 2020). Additionally, other very-high resolution DEMs can be derived from aerial photographs captured by terrestrial surveys, unmanned aerial vehicles (UAV) and terrestrial or airborne Lidar scanners and laser altimetry (e.g. Ice, Cloud and land Elevation Satellite-2 - ICESat-2) to derive short-term glacier elevation changes (Huss et al., 2013; Fischer et al., 2016; Pelto et al., 2019; Wand et al., 2021; Wu et al., 2022).

The major aims of this paper are therefore to investigate the potential and limitations of geodetic mass balance estimates derived from VHR Pleiades satellite data (using 5 DEMs over the 3-year period 2020-2022). We obtain annual and seasonal mass changes in order to make inferences about glacier accumulation regimes in the selected sites based on the geodetic results and by the SAR–derived glacier index of Huang et al (2022), supported by climatic data. For these investigations we have selected two specific regions in High Mountain Asia which have displayed dissimilar mass change rates. The Muztag Ata (Eastern Pamir), which is predominantly influenced by the westerlies and where glaciers have been in a state of mass balance for a longer period, and Western Nyainqêntanglha (Central–Eastern Tibet), which is more influenced by the monsoon and has strongly negative mass budgets (Bhattacharya et al. (2021). Moreover, we aim to monitor the ongoing mass balance trends in the Muztag Ata and Western Nyainqêntanglha regions.

## 2 Study areas

### 2.1 Muztag Ata

The Muztag Ata massif (38°17′ N, 75°07′ E; 7546 m asl.) is situated in Eastern Pamir, west of the Taklamakan Desert (Xinjiang Uighur Autonomous province, China, Fig. 1, left panel). The massif is a result of the exhumation of the Paleozoic metamorphic rocks (along with the nearby Kongur Shan mountains). These peaks represent an area of anomalously high topography at the northwestern tip of the Tibetan Plateau, rising ~4000 m above the ~3500 m asl. Plateau (Seong et al., 2009).

A relevant aspect of the atmospheric circulation over High Mountain Asia is its influence on the glaciers' thermal regime. Glaciers in the arid NW Tibetan Plateau are predominantly continental-type, cold–based (with their basal part entirely below the pressure melting point) and receive little precipitation. Climatically, the Muztag Ata region is characterized by a semi-arid continental-type regime, said to be one of the driest and coldest glacierized regions



in low- to mid-latitude China (Zhou et al., 2014). Precipitation is mainly driven by the mid-latitude westerly flow (Yao et al., 2012) and is concentrated between the spring and summer seasons (April-September). Data from the Taxikorgan meteorological station (37°46′ N, 75°14′ E, ~50 km south of Muztag Ata and placed at ~3100 m asl.), collected between 1957-2010, indicates a mean annual temperature of +3.4 °C (Yan et al., 2013; Yang et al., 2013). Based on an ice core drilled at ~7000 m, Duan et al. (2015) estimated mean annual snow accumulation of

605 mm w.e.. Glaciers in Muztag Ata represent a relevant water resource on a seasonal to long-term basis, regulating the freshwater supply into the Taxkorgan and Gezhe Rivers, tributaries of the Yarkand and Kaxgar Rivers respectively, and at the headwater of the Tarim River that runs through the Taklamakan desert (Holzer et al., 2015; Zhu et al., 2018a).

As per the year 2013, the Muztag Ata massif contains ~273 km$^2$ of glacier ice; the largest individual glacier here

is Kekesayi Glacier at ~86 km$^2$ (Holzer et al., 2015). Valley glaciers are cold–based (Zhou et al., 2014), and several of them grade distally from debris-free ice into debris-covered and subsequently ice-debris complexes similar to the landforms in Central Tien Shan (Bolch et al., 2019). These landforms terminate as ice-cored moraines that evidence the presence of buried ice such as thermokarst depressions and a steep front resemblant of intact rock glaciers. In addition, some of these glaciers (RGI60-13.41404, RGI60-13.41407, RGI60-13.41411,

RGI60-13.41900; Fig.1) currently display behaviour typical of surge-type glaciers (Guillet et al., 2022).

Previous assessments of glacier mass balance in Muztag Ata (Yao et al., 2012) had detected balanced conditions and even slight mass gains in the early 2000's, placing the Muztag Ata area within the Pamir-Karakorum anomaly. More recent studies, however, suggest that although glacier area changes are still negligible, the Muztag Ata region has recently shifted from approximately balanced conditions to slightly negative glacier mass budgets.

Holzer et al. (2015) and Bhattacharya et al. (2021) found slight mass losses over the last 6 decades (-0.03 m ±0.33 w.e. a$^{-1}$ and -0.06 m ±0.07 w.e. a$^{-1}$, respectively). Extending the time series will allow for the further monitoring of this development.

.

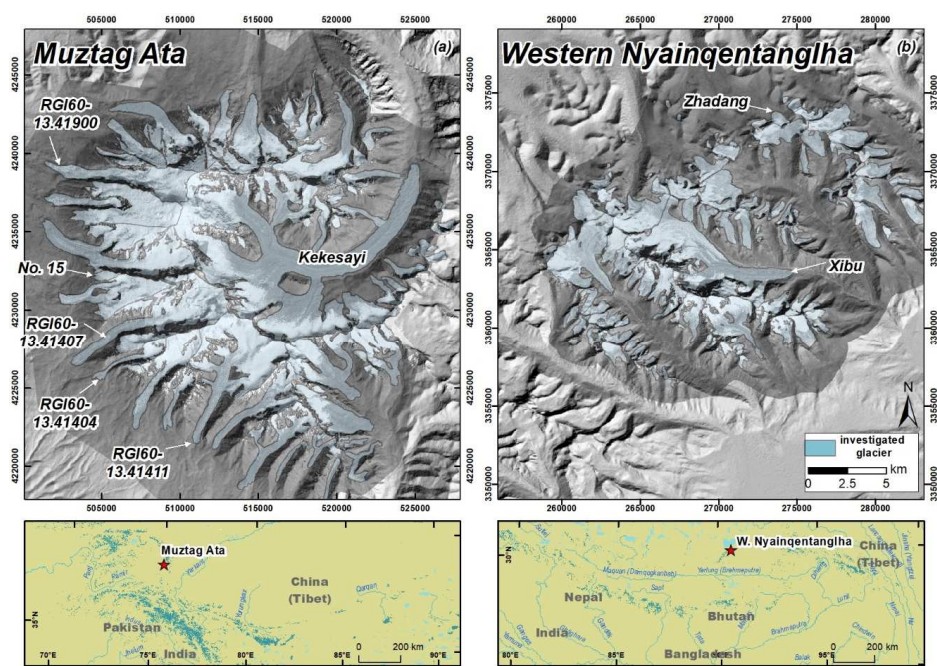

**Figure 1. Overview of the study sites with specific glaciers mentioned throughout the study and location in High Mountain Asia. Background images are Pleiades 2019 panchromatic band mosaics overlayed on top of Shuttle Radar Topography Mission (SRTM) DEM hillshades.**

## 2.2 Western Nyainqêntanglha


The Western Nyainqêntanglha mountains are located in the southeastern center of the Tibetan Plateau. The SW-NE oriented mountain range ~230 km in length reaches a maximum elevation of 7162 m asl. (Mount Nyainqêntanglha) (Yao et al., 2010). The main water divide in the Western Nyainqêntanglha range drains into the



Yangbajain-Damxung valley to the NW and into the Tsangpo-Brahmaputra River (Yao, 2008) further down, and into the Nam Co saltwater lake to the SE. Our study site covers a ~750 km² portion situated at the center of the main high mountain range (Fig. 1, right panel).

In general, glaciers to the SE of the Tibetan Plateau are located in a maritime regime under the influence of the Indian summer Monsoon with abundant precipitation. However, the location of the Western Nyainqêntanglha mountains with respect to major atmospheric circulation patterns in High Mountain Asia results in a highly
complex climate setting. The NW slopes lie on the windward side of the western winds (which prevail during the dry season), whereas the SE flank is exposed to the easterly winds of the Indian summer Monsoon that dominate in the wet season (Kang et al., 2009). Western Nyainqêntanglha glaciers therefore lie in a transitional zone between the more continental regime to the NW plateau and the maritime regime to the SE, and so polythermal glaciers (mixed basal thermal conditions) are common here (Shi and Menenti, 2013). Overall, the climate around
these glaciers has been described as continental summer-precipitation-type, with both accumulation and ablation maxima occurring during the summer months (Kang et al., 2009). The mean annual air temperature at Ambo meteorological station (4820 m asl., ~220 km to the NE of the Nyainqêntanglha range) is −3.0 ◦C (Bolch et al., 2010). In their energy balance model, Caidong and Sortenberg (2010) estimated the annual precipitation between 700 mm and >900 mm for Xibu Glacier.
Bolch et al. (2010) inventoried >1000 individual glaciers covering about 800 km² here in 2001, whereas for our specific study site the glacier area accounted for <150 km². The debris-covered Xibu Glacier (~23 km²) is the largest sampled glacier. In contrast to the Muztag Ata massif, glaciers in the Western Nyainqêntanglha region have been rapidly shrinking [27% between 1970-2014; Wu et al. (2016)] and losing mass at an accelerated pace (from -0.24 m w.e a⁻¹ to -0.47 m w.e a⁻¹) since the 1960's (Zhang and Zhang, 2017; Luo et al., 2020; Ren et al.,
2020; Bhattacharya et al., 2021). Both modelled glacier mass balance (–1.0 m w.e. a⁻¹ for the period 2001 – 2011; Huintjes et al., 2015) and in situ glaciological measurements (-1.2 m w.e. a⁻¹ to -1.6 m w.e. a⁻¹ between 2009 and 2014; Zhang et al., 2013 and 2016) on Zhadang Glacier have shown even greater mass loss rates, whilst the in situ mass balance averaged over the 2006-2017 period was of -1.35 m w.e. a⁻¹ (Yao et al., 2012 and unpublished data).


**3 Data and methods**

**3.1 Geodetic mass balance**

To derive annual and seasonal geodetic mass balance estimates, we produced a time series of glacier surface elevation change estimates through DEM differencing. To ensure the robustness of these data, we have considered, and where necessary corrected for, biases related to DEM misalignment, data voids in the elevation change grids, contrasting acquisition dates and seasonal snow conditions. The utilised data and individual processing steps are described in more detail below.

**3.1.1 DEMs derived from Pléiades tri-stereo imagery**

The commercial Pléiades 1A and 1B twin satellites were launched in December 2011 and 2012, respectively. Images are acquired at 0.7 m pixel resolution and delivered by Airbus Intelligence at an oversampled ground
sampling distance (GSD) of 0.5 m for the panchromatic band (Gleyzes et al., 2012). In this study, we used Pléiades tri-stereo panchromatic scenes to cover the pre- and post-monsoon seasons in Muztag Ata and Western Nyainqêntanglha between 2019 and 2022 (Table 1). Because of the relatively limited Pléiades footprint of 20 x 20 km (Berthier et al., 2014), both study areas were covered by two or three acquisitions separated in time on most occasions (Table 1). In all cases, partial acquisition dates were no more than 2 weeks apart. From these
images, we generated 4 m resolution DEMs and 0.5 m orthoimages using the NASA Ames Stereo Pipeline (ASP, Shean and others, 2016), implementing the Semi-Global-Matching algorithm (Shean et al., 2020).

Although no field-surveyed ground control points (GCPs) were used in the DEM extraction process, the vertical bias of Pléiades–derived DEMs with no GCPs has been reported typically between 1-2 m (occasionally up to 7 m) (Berthier et al., 2014; Belart et al., 2017; Falaschi et al., 2022), though this can be reduced to a few decimeters
after DEM coregistration. Pléiades DEMs are currently amongst the most common very high resolution DEMs used in geodetic mass balance assessments (e.g. Denzinger et al., 2021; Wagnon et al., 2021; Bhattacharya et al., 2021; Bolch et al., 2022). One of the main advantages of the Pléiades satellites with respect to other sensors used in glaciological applications (e.g. ASTER, SPOT5 HRS, Worldview-1 and Worldview -2), beyond the higher spatial resolution, is that panchromatic bands are coded in a finer 12-bit radiometric resolution, providing a better
optical contrast over low contrast and optically saturated areas. Nonetheless, when fresh snow has fallen very shortly before image acquisition (leading to a very high reflection over a large area and lack of contrast in the images), a relatively large number of data voids can be present in the derived DEMs. Such is the case of the





September 2021, March 2022 Western Nyainqêntanglha and April 2022 Muztag Ata Pléiades DEMs, where voids account for 20-23% of the glacier area. The remaining DEMs contain less than 9% data voids.


**Table 1.** Acquisition dates of the Pléiades imagery. *Full area was covered by two scenes acquired on the same day. **Partial coverage only.

| Muztag Ata | | Western Nyainqêntanglha | |
|---|---|---|---|
| Pléiades | Sentinel-2 | Pléiades | Sentinel-2 |
| 05-09-2019 | 03-09-2019 | 29-10-2019 | 27-10-2019 |
| 11-09-2019 | 13-09-2019 | 11-11-2019 | 11-11-2019 |
| 09-09-2020 | 07-09-2020 | 08-10-2020* | 11-10-2020* |
| 22-09-2020 | 22-09-2020 | | |
| 08-09-2021 | 07-09-2021 | 30-09-2021** | 06-10-2021 |
| 21-09-2021 | 22-09-2021 | | |
| 22-09-2021 | | | |
| 10-04-2022 | 10-04-2022 | 18-03-2022 | 20-03-2022* |
| 17-04-2022 | 15-04-2022 | 19-03-2022 | |
| 31-08-2022 | 02-09-2022 | 01-11-2022 | 31-10-2022 |
| 27-09-2022 | 27-09-2022 | 07-11-2022 | 05-11-2022 |

**3.1.2 Classification of snow and ice using Sentinel-2 scenes**

The European Copernicus Sentinel-2 satellites (A and B), carrying the MultiSpectral sensor (MSI), were launched in 2015 and 2017. The instruments onboard Sentinel-2A and 2B (swath width = 290 km, orbit repeat rate = 10 days) acquire data in four visible (VIS) and near-infrared (VNIR) bands at 10 m spatial resolution, and six VNIR

and short-wave infrared bands (SWIR) at 20 m resolution (Kääb et al, 2016). We used Sentinel-2 imagery to classify surface cover (ice, firn, snow) over Pléiades imagery due to the availability of an automatic, robust method such as ASMAG (Rastner et al., 2019) to identify snow characteristics. Also, the lower spatial resolution and larger swath width (i.e. smaller number of images to be processed) of Sentinel-2 compared to Pléiades are much less demanding from a computational point of view. The distribution of snow, firn and ice surfaces is fundamental

for a realistic estimation of annual geodetic glacier mass balance. In this regard, Pelto et al. (2019) judged this distribution to have an even greater impact on geodetic mass changes than snow, firn and ice densities themselves. Whilst snow and bare glacier ice are readily identifiable on multispectral satellite imagery using conventional methods (Paul et al., 2015), the distinction between snow, firn and ice on glacier surfaces can be more difficult. Indeed, several methodological approaches to identify snow line altitudes have shown that firn can be classified

as either snow or ice (depending e.g. on the amount of impurities) when using normalized snow indexes or thresholding of single bands/band ratios (e.g. Racoviteanu et al., 2019). Here we implemented the ASMAG algorithm (Rastner et al., 2019, see also Falaschi et al., 2021), using Sentinel-2 images to distinguish between ice and snow/firn areas. In summary, the algorithm converts the raw digital numbers to top of atmosphere reflectance and maps snow cover ratio (SCR). ASMAG then follows the histogram approach of Bippus (2011) applied to a

glacier mask to differentiate ice from snow. To this end, it implements an automatic threshold to the near-Infrared (NIR) band, in turn based on the Otsu (1979) thresholding algorithm (Rastner et al., 2019).

We selected Sentinel-2 scenes acquired as close to the Pleiades imagery as possible. Out of 17 images used for the mapping of snow and ice, the acquisition date of only two Sentinel-2 images differed by more than 2 days (see Table 1) from the Pléiades imagery. We inspected ERA5-Land daily temperature and precipitation data to ensure

that seasonal snow conditions had not been significantly altered between the Pléiades and corresponding Sentinel-2 scenes (due to snowfall or unusually high temperature events). According to ERA5-Land (Muñoz-Sabater et al., 2021) daily data, there were around 3 cm of fresh snow between the 30 September and 6 October 2021 Pléiades and Sentinel-2 images in the Western Nyainqêntanglha range. Because the 30 September Pleiades scene is anyhow snowbound, the overall seasonal snow conditions appear consistent among the two scenes. As for temperature,

the summers of 2019 to 2021 show monthly temperature anomalies below +0.6 °C, which is far below the historical +2.4 °C maximum for the full 1950-2022 series.

Despite the topographic correction featured in the ASMAG algorithm, some glacier parts in cast shadows were initially misclassified as ice in the accumulation regions. To remove these misclassifications, we masked out pixels located above the mean snowline altitude (SLA) plus 2 standard deviations and reassigned them to the snow

class and implemented a low-pass filter.

**3.1.3 Glacier outlines**



We used the glacier outlines of the year 2019 available from Bhattacharya et al. (2021), which were adjusted
based on the Randolph Glacier Inventory 6.0 (Randolph Consortium, 2017), as the basis for our glacier inventory.
For geodetic mass balance studies of small regional extent, varying quality of glacier polygons may have a
relatively large effect on the final results (see e,g, Sommer et al., 2020; Falaschi et al., 2022). Thus, the RGI glacier
polygons were manually adjusted (accounting for glacier length changes, removing non-glacierised stable ground,
reinterpreting debris-covered ice) to fit the glacier extent on each of the Pléiades acquisition dates (Table 1) by
visual interpretation of the 0.5 m orthophotos (UTM 43N and 46N for Muztag Ata and Western Nyainqêntanglha,
respectively). The glacier area uncertainty was conservatively calculated using the approach followed by Wagnon
et al. (2021), as the product of the glacier outline initial perimeter of each sampled time interval and two times the
0.5 m ground sampling distance of the Pléiades panchromatic band.

### 3.1.4 DEM differencing and generation of elevation change maps

We divided the full 2019-2022 study period into three annual time intervals: 2019-2020; 2020-2021; 2021-2022,
and two seasonal (2022 winter and summer) time intervals. Before differencing, we coregistered all DEMs to the
reprocessed 2019 Pléiades DEM from Bhattacharya et al. (2021) following Nuth and Kääb (2011). This way, any
remaining vertical and horizontal shifts between DEMs are minimized before the *dDEM* grids are generated by
subtracting the latter from the initial (*reference*) DEMs.
Elevation change grids derived from Pléiades DEMs often show low-frequency biases in off-glacier elevation
change residuals (often termed '*undulations,*' Hugonnet et al., 2022), which have been attributed to satellite along-
track attitude oscillations [jitter] (Girod et al., 2017; Deschamps-Berger et al., 2020). We implemented the bias
correction procedure described in Deschamps-Berger et al. (2020) to remove the low-frequency undulation
patterns (Fig. 2a, b).
To eliminate on-and off-glacier anomalous cell values from the *dDEM* grids, we considered all values exceeding
±150 m as outliers and removed them. A second outlier removal step was carried out where the neighboring cells
of data gaps in the *dDEM* grids contained high-magnitude elevation differences, which shared statistical
characteristics with real glacier thinking/thickening. To filter this noise, we first implemented a 3-cell buffer
around the data gaps and removed the cells within the buffered areas. We then filled the resulting data gaps using
the glacier-wide hypsometric approach of McNabb et al. (2019), fitting a fifth-degree polynomial function to the
mean elevation change on 50 m elevation bins (Fig. 2c-f). We chose a 3-cell buffer so as not to remove valid cells
from the original *dDEM* grids. Lastly, we mosaiced these tiles to generate the final elevation change maps. Basic
statistical parameters of our *dDEM* grids (Table 2) after coregistration are akin to previously published geodetic
mass balance assessments of annual to seasonal scale in other glacierized regions using Pléiades DEMs (e.g.
Beraud et al., 2022).

**Table 2.** Statistics of *dDEM* grids on- and off-glacier after coregistration (SD = standard deviation, SE = standard
error, NMAD = normalised median absolution deviation).

| | Percentage of data voids on-glacier (%) | off-glacier mean elevation difference (m) | off-glacier SD (m) | off-glacier SE (m) | NMAD (m) |
|---|---|---|---|---|---|
| **Muztag Ata** | | | | | |
| 2019-2020 | 8.6 | -0.10 | 2.9 | 9.0 x10$^{-4}$ | 1.3 |
| 2020-2021 | 10.4 | 0.06 | 2.4 | 4.8 x10$^{-4}$ | 1.3 |
| winter 2022 | 20.8 | 0.06 | 2.9 | 6.4 x10$^{-4}$ | 1.3 |
| summer 2022 | 28.1 | -0.11 | 1.7 | 3.7 x10$^{-4}$ | 1.3 |
| 2021-2022 | 16.4 | <0.01 | 1.9 | 4.2 x10$^{-4}$ | 1.2 |
| 2019-2022 | 14.4 | -0.02 | 0.7 | 4.4 x10$^{-4}$ | 1.3 |
| **Western Nyainqêntanglha** | | | | | |
| 2019-2020 | 6.8 | -0.02 | 1.4 | 3.1 x10$^{-4}$ | 1.2 |
| 2020-2021 | 23.5 | 0.03 | 1.4 | 3.7 x10$^{-4}$ | 1.1 |
| winter 2022 | 23.7 | -0.11 | 1.4 | 3.7 x10$^{-4}$ | 1.0 |
| summer 2022 | 14.7 | 0.03 | 1.0 | 2.4 x10$^{-4}$ | 0.8 |
| 2021-2022 | 29.7 | -0.07 | 1.0 | 2.6 x10$^{-4}$ | 0.9 |
| 2019-2022 | 9.2 | -0.03 | 1.0 | 2.3 x10$^{-4}$ | 0.6 |

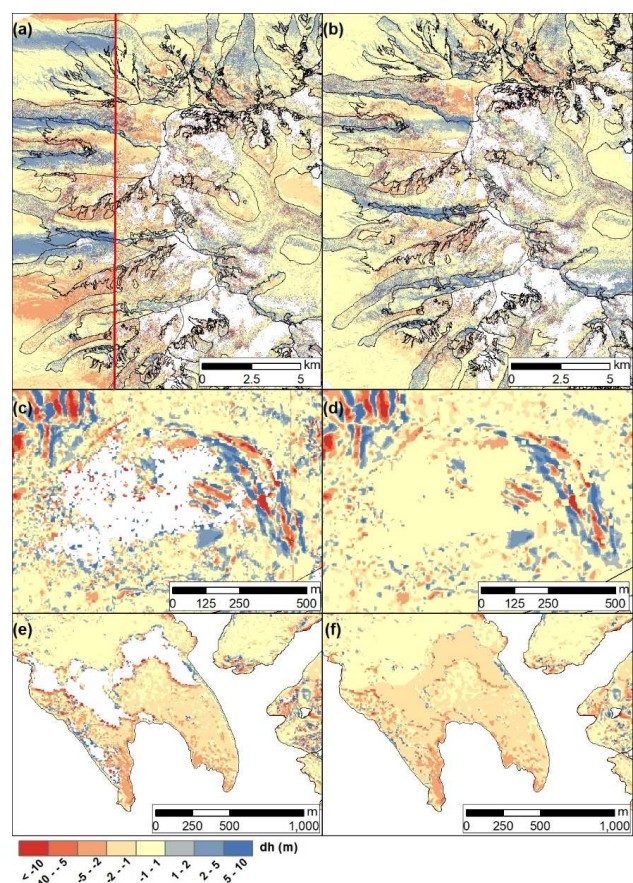

**Figure 2. Along-track bias removal in Muztag Ata *dDEM* grids before (a) and after (b) bias correction. The diverse impact of the jitter effect on individual *dDEM* grids can be appreciated left and right of the red line in panel (a). (c, d) and (e, f) depict examples of outlier removal and gap-filling before and after the implementation of the buffer approach in the Muztag Ata and Western Nyainqêntanglha *dDEM* grids, respectively. Note the crevasse motion captured as alternating blue (elevation gain) and red (elevation loss) in the in the Muztag Ata panels.**

### 3.1.5 Bulk density

Conversion from geodetic volumetric change (in meters) based on the DEM differencing technique to water equivalent (in m w.e.) requires consideration of the densities of the material involved. With no field-surveyed snow density measurements contemporary to our surveyed time periods available, we used the 410 ±60 kg m$^{-3}$ snow density value used in the energy balance model comprising Muztag Ata N15 and Zhadang glaciers (in Muztag Ata and Western Nyainqêntanglha districts, respectively) by Zhu et al. (2018a). According to the authors, this density value was retrieved from snow pits. We assumed a ±60 kg m$^{-3}$ uncertainty value, which is a standard value used in geodetic mass balance assessments (Huss., 2013). For glacier ice areas, we used a density of 900 kg m$^{-3}$ (Zhu et al., 2018a) and an uncertainty of ±10 kg m$^{-3}$ (Clarke et al., 2013).

### 3.1.6 Mass balance calculation

We present annual mass balance estimates for the Muztag Ata and Western Nyainqêntanglha regions starting in the winter of 2019 up to the summer of 2022 but provide also estimates of the 2022 winter and summer seasons. The annual mass balance $B_a$ can be defined as the algebraic sum of the winter $B_w$ and summer $B_s$ balances



$$B_a = B_w + B_s \qquad (1)$$

The total volume change $\Delta v$ (m$^3$) over a given time interval $t$ is then derived from the respective elevation difference $dh$ of the two grids at pixel $k$ with cell size $r$ of the DEMs, summed over the number of pixels covering the glacier, and is expressed as (cf. Zemp et al., 2013):

$$\Delta v = r^2 \times \sum dh_i \qquad (2)$$

Applying the snow and ice masks described in Sect. 4.3, we then assessed the annual and seasonal geodetic mass balance $\Delta m$ (m w.e.) considering the volume change $\Delta v$, the density of the involved material $\rho$, and both the snow and ice areas $A_i$

$$\Delta m = \sum \frac{\Delta v \times \rho}{A_i} \times t^{-1} \qquad (3)$$

### 3.1.7 Firn densification correction

Lowering of the annual snowpack on glaciers occurs as a result of firn densification, a process that leads to underestimated volume changes (Sold et al., 2013) and is often neglected in geodetic mass balance studies relying on the dDEM technique. We implemented a simple firn densification approach following Sold et al. (2013) to estimate the annual elevation change due to firn densification $dh_{firn}$ integrated over the entire firn column

$$dh_{firn} = \frac{b_n}{\rho firn_u} - \frac{b_n}{\rho firn_l} \qquad (4)$$

where $b_n$ is the net mass balance averaged over the previous mass balance year or period (in kg m$^{-2}$) over the accumulation area [positive, by definition] (Belart et al., 2017). $\rho firn_u$ and $\rho firn_l$ are the density values at the upper and lower ends of the firn column, set to $\rho firn_u$ = 470 kg m$^{-3}$ and $\rho firn_l$ = 857 kg m$^{-3}$, as retrieved from a 41 m-deep ice core drilled at ~7000 m on Muztag Ata (Duan et al., 2015).

Because some of our Pléiades imagery was acquired around the end of the accumulation season (Western Nyainqêntanglha) or contains seasonal snow (Muztag Ata), 'snow' areas do not exactly match "accumulation" areas and can have an overall negative elevation change.

Using the most recent elevation grids available from Bhattacharya et al. (2021) [Muztag Ata: 2013-2019; Western Nyainqêntanglha: 2018-2019) in combination with the "snow" areas derived from the 2019 Sentinel-2 scenes to calculate the net mass balance in the "accumulation" area leads to a negative signal. In consequence, we considered $b_n$ to be the net mass balance of the snow areas as depicted in the September 2021 (Muztag Ata) and October 2020 (Western) Pléiades scenes, which are those showing the least amount of seasonal snow, and thus snow areas should be representative of accumulation areas. Although topoclimatic factors can make firn lines vary spatially and temporally (Guo et al., 2014) through time, for the sake of simplicity we assumed that firn densification did not change over different time periods. Our approach is also based on the accumulation rate retrieved from a single year on each study site, whilst some variability is naturally expected (Duan et al., 2015). We then scaled the firn densification correction linearly according to the length of each time span between consecutive imagery acquisition.

Because of the simplifications and assumptions of the firn correction (equal net accumulation through the study period, constant densification rate through the hydrological year), we considered a 50% uncertainty in the firn correction when quantifying the total mass balance error.

### 3.1.8 Seasonal correction

Ideally, mass balance observations should be made repeatedly at the end of the ablation season to ensure that the full, annual budget of ablation and accumulation has been captured, although data availability means this is rarely the case. Correction for the impact of such seasonal bias is therefore commonly required (Belart et al., 2017 and 2019; Abdel Jaber et al., 2019).

Neglecting such seasonality corrections can introduce potential biases, since the effect of seasonal signals are particularly strong when short time intervals are considered. A simple approach to account for the seasonal shift of acquired remotely–sensed data with respect to the hydrological cycle consists of using the daily mass balance rate for the given period to fill in the deficit of missing days or subtract the input of excess days (Abdel Jaber et al., 2019). The extent of the seasonal bias introduced when using this method therefore depends on the fraction of missing or excess days in relation to the onset of the accumulation and ablation periods. We nevertheless chose





this approach since our study sites lie in transitional to continental semi-arid environments, where contrary to glaciers in maritime regions, large annual mass balance turnover is not to be expected (Duan et al., 2015; Zhu et al.,2015 and 2018a). Moreover, the percentage of missing/excess days relative to the start of the winter (1 October) and summer (1 April) seasons was below 10% for our Pléiades imagery. We considered that any remaining seasonality-related biases should be within the overall uncertainty range.

### 3.1.9 Ice dynamics considerations

As stated in Sect. 1, geodetic mass balance assessments have been often used to calibrate contemporary glaciological measurements. Differences amongst them in relation to vertical surface elevation changes have been attributed to firn densification and ice vertical velocity. Whilst differencing of elevation changes from *dDEM* and glaciological observations should yield representative emergence and submergence velocities (Pelto et al., 2019), glaciological data is unfortunately unavailable as part of this study. Alternatively, Belart et al. (2017) implemented a full-Stokes model by ingesting glacier bedrock and surface DEMs, in situ GPS velocities, coupled with the firn densification model derived realistic emergence and submergence velocities. This, however, was beyond the scope of the present study, as the influence of ice dynamics on overall mass budget is usually very small (<5%) on a year-to-year basis and only significant when calculating mass budget over few decades (e.g. Mukherjee et al. 2022).

### 3.1.10 Uncertainty

In spite of the coregistration and bias-correction procedures applied, some of the *dDEM* grids showed systematic residuals on stable terrain [off-glacier] (Table 2). We addressed these biases in elevation change off-glacier $\sigma_{sys}$ as the mean difference between two DEMs (Koblet et al., 2010):

$$\sigma_{sys} = \frac{\sum (H_{DEMi} - H_{DEMf})}{n} \tag{5}$$

being $H_{DEMi}$ and $H_{DEMf}$ the elevation of the initial and final DEMs and *n* the number of cells on stable terrain.
For the calculation of the random uncertainty of the volumetric mass balance estimation $\sigma\Delta v$, we considered the volumetric uncertainties on mean elevation change, snow and ice areas and firn densification (see Sect. 4.2), which we summed quadratically to propagate the error:

$$\sigma\Delta v = \sum \sqrt{(dh \times \sigma_{Ai})^2 + (A_i \times \sigma_{dh})^2 + (dh_{firn} \times \sigma_{Ai})^2 + (A_i \times \sigma_{dfirn})^2} \tag{6}$$

$\sigma_{Ai}$ being the area uncertainty, $\sigma_{dh}$ the uncertainty on the rate of elevation change, $dh_{firn}$ the annual elevation change owed to firn densification, and $\sigma_{dfirn}$ the uncertainty in firn densification rate. In turn, we calculated $\sigma_{dh}$ using the *patch* method of Berthier et al. (2016), which evaluates the decay of the mean elevation change error on stable terrain (off-glacier) with the averaging area (Wagnon et al., 2021; see also Falaschi et al., 2022). Since we attributed a different density and related uncertainty to the snow and ice area classes (snow: ±60 kg m⁻³; ice: ±10 kg m⁻³), we calculated the uncertainty of each class separately, by adding the density uncertainty $\sigma_{f\Delta v}$ of each surface to the volumetric uncertainty in Eq. (5):

$$\sigma_m = \sum \frac{\sqrt{(f_{\Delta v} \times \sigma\Delta v)^2 + \left(\Delta v \times \sigma_{f_{\Delta v}}\right)^2}}{\bar{A}} \times t^{-1} \tag{7}$$

Finally, to obtain the overall glacier-wide mass balance uncertainty $\sigma_{m.tot}$, we summed the uncertainties of the snow and ice areas quadratically:

$$\sigma_{m.tot} = \sqrt{\sigma_{m.ice}^2 + \sigma_{m.snow}^2} \tag{8}$$

### 3.2 Glacier Index: wet snow and firn area ratios derived from Sentinel-1 and Landsat OLI imagery

We implement the Glacier Index of Huang et al. (2022) to characterize glacier accumulation regimes and validate our geodetic results. To account for different glacier areas between the study sites, we express firn and wet snow areas on each region as a fraction of the total glacier area (hereafter referred to as firn area ratio and wet snow area ratio). This ratio can vary to a great extent across different geographic regions through time, whilst



interannual variations of the firn area ratio remain relatively small. The Glacier Index $I$ is then defined as the difference between the firn and wet snow area ratios


$$I = \frac{A_{firn}}{A_{total}} - \frac{A_{wet.snow}}{A_{total}} \qquad (9)$$

where $A_{firn}$ is the firn zone area, $A_{wet.snow}$ the wet snow zone area and $A_{total}$ the total glacier area. The Index is expected to be positive for winter accumulation–type glaciers, as summer snowfall is rare and hence $A_{wet.snow}$ <
$A_{firn}$. In contrast, the Index is more likely to be negative (i.e. $A_{wet.snow}$ > $A_{firn}$) for summer –glaciers due to recurring summer solid precipitation. We classified regions and glaciers as winter accumulation–type where $I \geq 0.05$ and summer–accumulation type where $I < 0.05$ following Huang et al. (2022).
   To discriminate wet snow areas (in late summer) and firn areas (in winter) over glaciers in our selected study areas, we used Landsat OLI and Sentinel-1 imagery on Google Earth Engine (Gorelick et al., 2017). First, the
Landsat scenes are used to recognize debris-covered and debris-free areas (ice and snow) on glaciers surface applying a threshold to the previously computed Normalized Difference Snow Index [NDSI] (Bruns et al., 2014). Then, since the Sentinel-1 C-band is sensitive to snow wetness and roughness (Huang et al., 2013), the backscattering properties between the different glacier facies are sufficiently different to discriminate the frozen firn in winter (here defined as the period between January and March) and wet snow zone in late summer (defined
here as the time interval from July 20 to September 10) using backscatter coefficients (Fig. 3). For full methodological details we refer to the Supplementary Material in Huang et al. (2022).

### 3.3 Investigating the influence of climate (temperature and precipitation) to govern glacier mass balance

To understand how the precipitation and temperature may have influenced the response of the glaciers in the study sites, we used remotely–sensed daily Global Precipitation Measurement (GPM) IMERG (Hou et al. 2014, Huffman et al. 2015) late run precipitation observations (availability: June 2000 – 2022). The multi-satellite precipitation estimates are obtained using quasi-Lagrangian time interpolation. The algorithm integrates all satellite microwave and infrared precipitation estimates with precipitation gauges and other precipitation
estimators for the entire globe and is expected to provide realistic estimates of precipitation. We opted to use the GPM precipitation dataset over either reanalysis data (e.g. ERA5, HARv2), since they have proven to largely over- or underestimate precipitation over the Tibetan Plateau (Wortmann et al., 2018; Lin et al., 2021). Moreover, Huang et al. (2022) have shown how both reanalysis (APHRODITE, ERA5, HARv2) and instrumental records (or instrumental records (e.g. the Taxikorgan climate station) are difficult to link with glacier accumulation
regimes in several regions throughout High Mountain Asia. We use temperature estimates from ERA5 Land (Hersbach et al. 2020, Munoz-Sabater et al. 2021) (availability: Jan 1950-July 2022), a gridded reanalysis data based on numerical weather prediction models.
   We converted the daily GPM precipitation data to monthly data and calculated the monthly solid precipitation as all the precipitation when the corresponding temperature is less than 0°C. To estimate seasonal temperature and
solid precipitation, we assumed May-October as summer months and November-April as winter months. We then obtained the total amount of solid precipitation for each year by matching with the dates/months of the geodetic data as shown in Table S1.
   To determine the relation between glacier mass balance and climate variability, we first found the mean annual, winter and summer average (temperature and snowfall) over the 2001-2022 time periods and then calculated
annual and seasonal mean temperature and total snowfall anomalies. Finally, we correlated the glacier-wide mass budgets with the annual and seasonal variations of temperature and snowfall. To do this, we used our mass budget estimates at annual scale, added the geodetic mass balance values in Bhattacharya et al. (2021) and performed a correlation analysis using the averaged climate records for analogous sub-periods. Whilst we focus on our shorter 2020-2022 period, our aim was to evaluate climate conditions of the last three years in a longer climate/mass
balance context.

## 4 Results

### 4.1 Geodetic mass balance in Muztag Ata and Western Nyainqêntanglha

The glacier-wide, annual geodetic mass balances in Muztag Ata varied from moderately negative to slightly positive conditions (Fig. 4; Table 3). $B_a$ estimated from Eq. (3) was negative for 2020 (when at its most negative, -0.19 ±0.14 m w.e.) and 2022 glaciological years but rather positive in 2021 (Table 3). In terms of seasonal mass
balance, the 2022 winter mass budget was positive, whereas the summer budget was negative (Table 3).
   On an individual glacier basis, both the annual and seasonal mass balances of the (largest and debris-covered) Kekesayi Glacier matched the overall trends of the broader Muztag Ata massif. Interestingly, the annual mass





balance estimates of Muztag Ata No. 15 Glacier yielded positive values for all three surveyed years, whilst the seasonal winter (positive) and summer (negative) budgets closely followed the general pattern in Muztag Ata, respectively (Table 3).


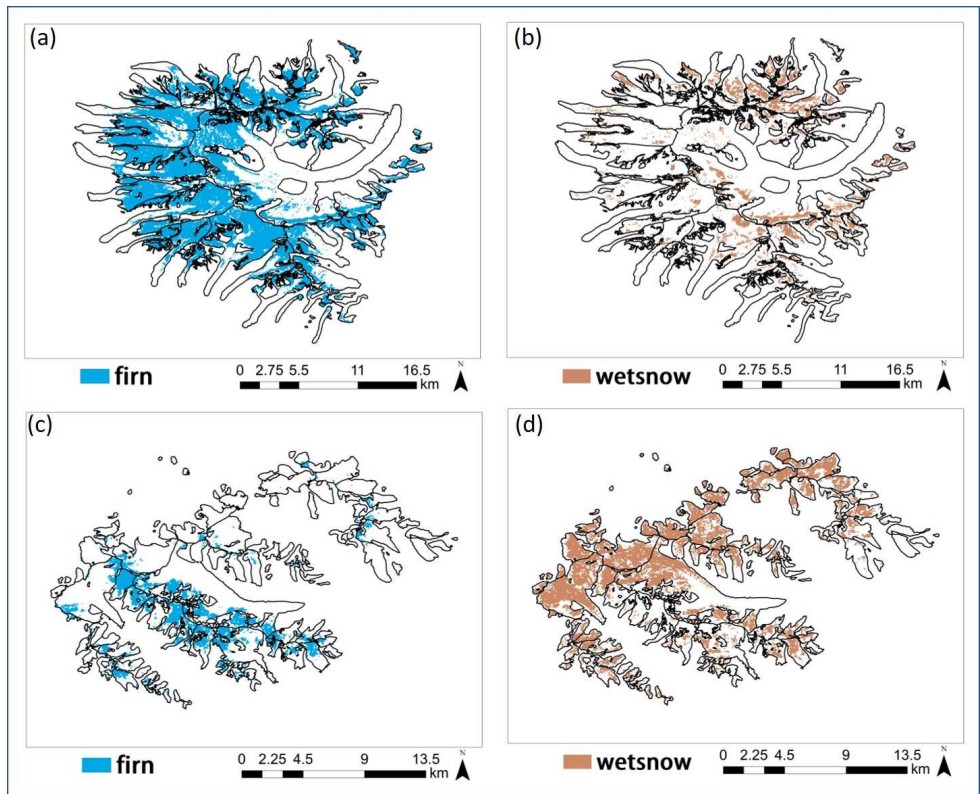

**Figure 3. Examples of firn and wet snow cover retrieval in winter and summer 2021 in Muztag Ata (a, b) and Western Nyainqêntanglha (c, d).**


The hypsometric distribution of elevation changes in Muztag Ata (Fig. 5a) reveals the lack of a clearly defined elevation change (i.e. mass balance) gradient. Periods with a positive mass budget (2021, winter 2022) show elevation gains at different elevation ranges. They occurred from 4800 m up to the uppermost reaches of the glacierized area (7600 m) during 2021, whereas the winter 2022 shows thickening below 5700 m and above 7200 m and elevation loss in between. Elevation gains (accumulation) during the year 2020 and the summer of 2022 were restricted to elevations >6100 m, whilst in the winter of 2022 elevation gains occurred above 7400 m only.

In contrast to Muztag Ata, the annual mass balance of glaciers in Western Nyainqentaglha showed a consistent, highly negative signal, with a negative peak of -0.70 ±0.22 in 2022 (Fig. 6; Table 3). While the winter mass budget was balanced, the summer mass budget was strongly negative (Table 3). It must be noted though, that the interannual and seasonal variability in mass balance are not fully comparable due to the incomplete coverage of the study site in the 2021 Pléiades acquisition. The largest, debris-covered Xibu Glacier had also a highly negative mass loss throughout the study period. Contrary to the overall seasonal budgets in Western Nyainqentaglha, Xibu Glacier lost mass during both the summer and winter seasons. Compared to Xibu Glacier, the mass loss of Zhadang Glacier was even greater on the annual scale (peaking at -1.12 ±0.21 m w.e. in 2021), but interestingly, recovered during the 2022 winter season (Table 3).

In Western Nyainqêntanglha, there are clear annual (and 2022 summer) elevation change gradients, in contrast to Muztag Ata. They depict a steep, positive gradient from lower to higher elevations, with elevation gains starting between 5900-6200 m (Fig. 5b). The 2022 annual hypsometric curve, though similar in shape to the two previous years, shows that all of the sampled glacier area underwent ice thinning over this interval. The 2022 winter curve shows slight elevation gains around 5500 m (coinciding with maximum area per elevation band), yet this does not counterbalance all the elevation losses elsewhere in the ice area.



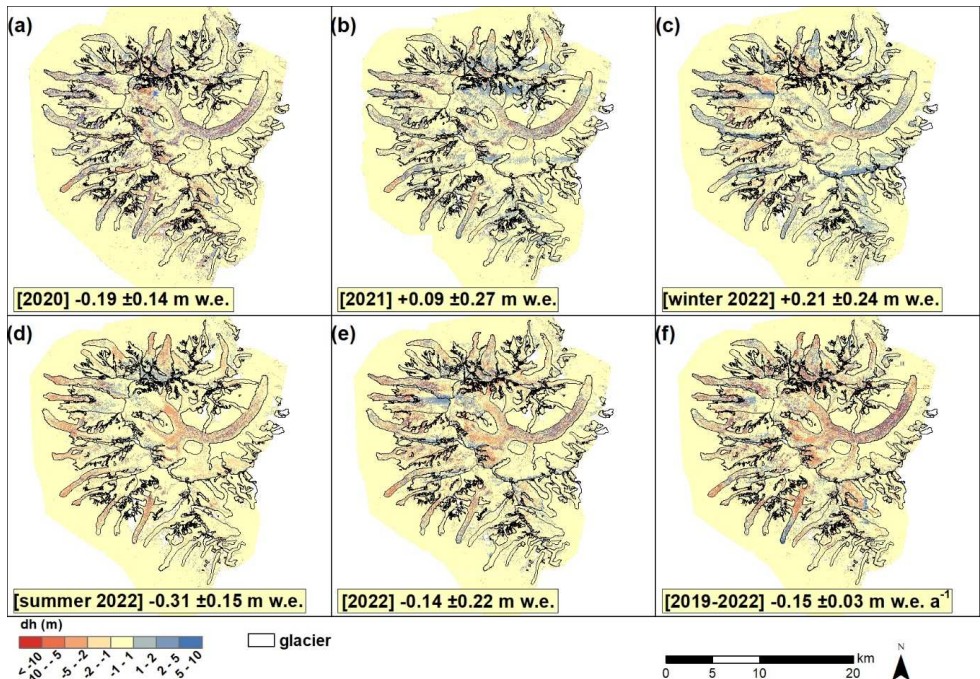

**Figure 4.** Annual (a, b, e, f) and seasonal (c, d) surface elevation change grids over Muztag Ata and associated geodetic
mass balance estimates.

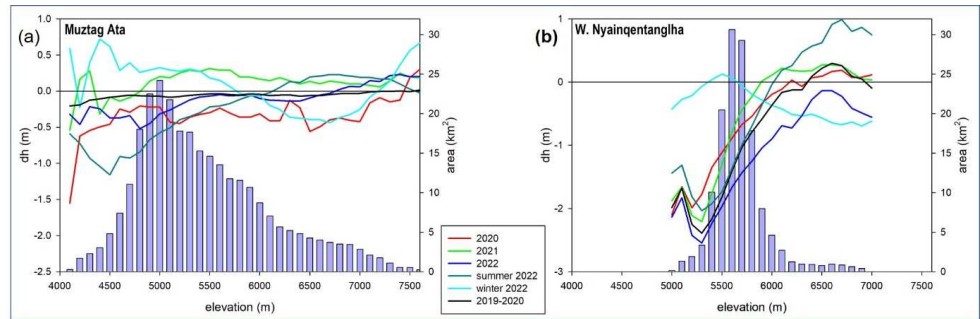

**Figure 5.** Hypsometric curve of glacier surface elevation changes over glacierized area in Muztag Ata (a) and Western
Nyainqêntanglha (b). The light blue bars represent the ice area distribution on 100 m elevation bins for the 2019 glacier
area on the right axis.






**Table 3.** Summary of time-interval volume changes ($10^6$ m$^3$) and mass balance (m w.e.) for the total glacier area and selected glaciers. Mass balance values in the 2020-2022 column are provided in m w.e. a$^{-1}$. Values in parenthesis () are calculated using a density of 850 kg m$^{-3}$ and values in brackets [] with a 3-year weighted density (see Scet. 6.4.1 for details).

| | 2020 | | 2021 | | winter 2022 | | summer 2022 | | 2022 | | 2020-2022 | |
|---|---|---|---|---|---|---|---|---|---|---|---|---|
| | Δv | Δm | Δv | Δm | Δv | Δm | Δv | Δm | Δv | Δm | Δv | Δm |
| **Muztag Ata** | | | | | | | | | | | | |
| **total glacier area** | -90.6 ±36.9 | -0.19 ±0.14 | +46.7 ±71.1 | +0.09 ±0.27 | +76.6 ±64.5 | +0.21 ±0.24 | -106.2 ±78.1 | -0.31 ±0.15 | -56.3 ±58.0 | -0.14 ±0.22 | -52.0 ±10.7 | [-0.15 ±0.03] (-0.14 ±0.03) |
| **No. 15** | 0.1 ±0.2 | +0.08 ±0.35 | 0.1±0.4 | +0.07 ±0.25 | +0.6 ±0.4 | +0.41 ±0.22 | -0.2 ±0.8 | -0.13 ±0.20 | +0.1 ±0.4 | +0.08 ±0.24 | +0.2 ±0.1 | [+0.09 ±0.04] (+0.08 ±0.04) |
| **Kekesayi** | -36.6 ±12.2 | -0.31 ±0.17 | +18.7 ±23.5 | +0.14 ±0.33 | +25.5 ±20.9 | +0.24 ±0.29 | -35.2 ±41.9 | -0.44 ±0.30 | -22.8 ±19.4 | -0.21 ±0.27 | -22.9 ±3.6 | [-0.29 ±0.04] (-0.26 ±0.04) |
| **Western Nyainqêntanglha** | | | | | | | | | | | | |
| **total glacier area** | -103.7 ±16.8 | -0.43 ±0.16 | -83.5 ±8.9 | -0.68 ±0.08 | -10.5 ±63.2 | -0.04 ±0.27 | -152.3 ±13.5 | -0.66 ±0.07 | -155.8 | -0.70 ±0.22 | -168.4 ±4.8 | [-1.04 ±0.11] (-0.93 ±0.10) |
| **Zhadang** | -1.5 ±0.3 | -0.80 ±0.41 | -1.3 ±0.2 | -1.12 ±0.21 | +0.3 ±1.0 | +0.28 ±0.31 | -2.1 ±0.3 | -0.83 ±0.12 | -1.7 ±0.3 | -0.68 ±0.21 | -2.1 ±0.1 | [-1.61 ±0.16] (-1.44 ±0.15) |
| **Xibu** | -17.9 ±3.9 | -0.47 ±0.22 | -14.7 ±2.3 | -0.61 ±0.17 | -3.9 ±13.8 | -0.16 ±0.35 | -20.9 ±3.5 | -0.58 ±0.10 | -28.4 ±3.6 | -0.65 ±0.41 | -27.4 ±1.1 | [-0.99 ±0.22] (-0.89 ±0.20) |





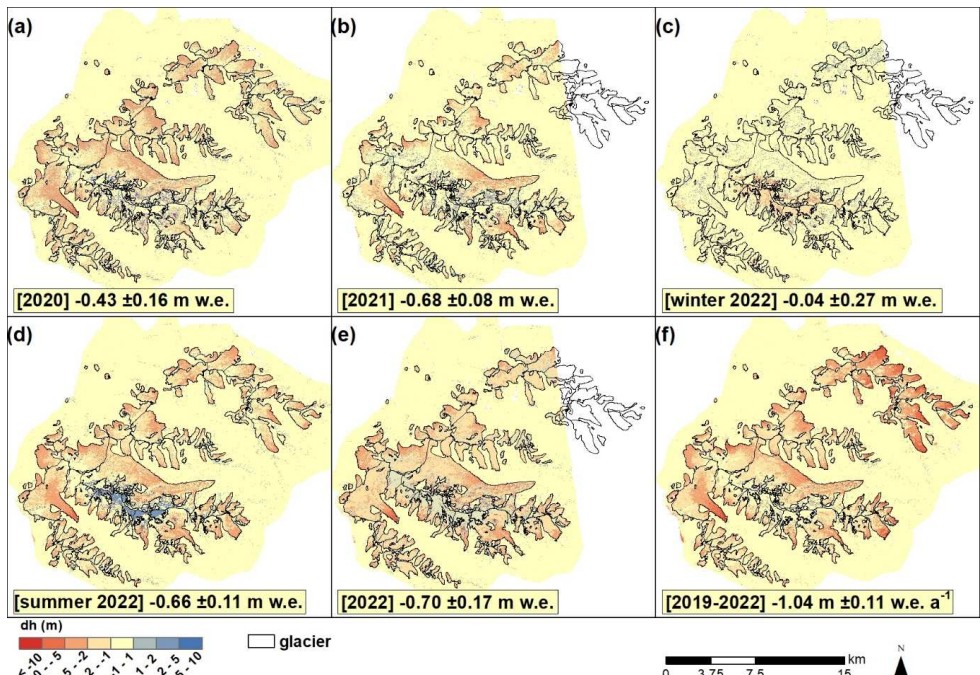

**Figure 6.** Annual (a, b, e, f) and seasonal (c, d) surface elevation change grids over Western **Nyainqêntanglha** and associated geodetic mass balance estimates.


### 4.2 Wet summer snow and winter firn area ratios derived from the Glacier Index

Firn area and wet snow-ratios varied between 0.55 - 0.6 and 0.15 - 0.23 in Muztag Ata and 0.25 - 0.28 and 0.33 - 0.66 in Western Nyainqêntanglha, respectively (Fig. 7 and Table S2). These ratios yielded Glacier Index values

between 0.32 and 0.45 in Muztag Ata and –0.38 and -0.07 in Western Nyainqêntanglha. To provide a measure of the individual glacier variability, the Glacier Index in Kekesayi and Muztag Ata No. 15 glaciers varied between 0.17 to 0.28 and 0.68 to 0.89, whilst in Xibu and Zhadang glaciers the Index ranged between -0.25 to -0.04 and -0.51 to -0.05, respectively. Based on these assessments, we attribute a winter accumulation regime for Muztag Ata and a summer accumulation regime in Western Nyainqêntanglha. Moreover, the evolution of the summer wet

snow area ratio points at 2020 as the most negative mass balance year in Western Nyainqêntanglha and the year 2019 in Muztag Ata.

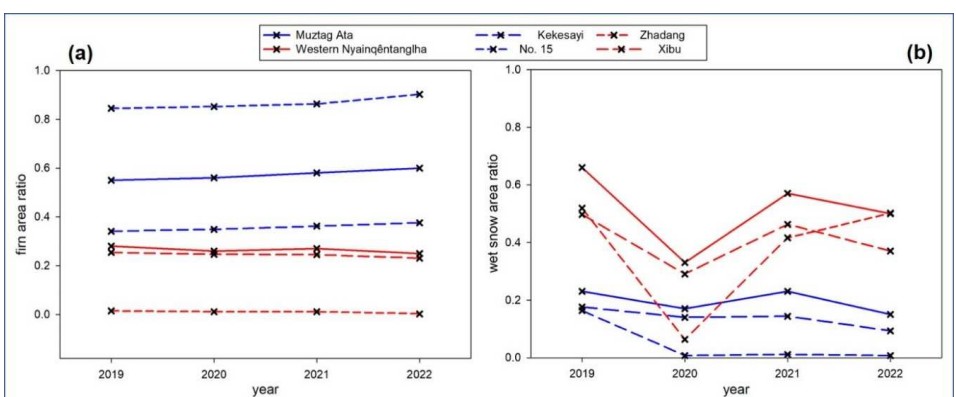

**Figure 7.** Evolution of the firn area– (a) and wet snow area–ratio (b) between 2019 and 2022.

**4.3 Relation between climate (temperature and precipitation) and mass balance**



In Muztag Ata, the summer (+0.5 °C to -1.1 °C) and winter (-0.9 °C and +0.3 °C) temperature anomalies were either positive or negative among the surveyed 2020 to 2022 years, and showed no prominence within the 2001-2022 period (Fig. 8c,e). In contrast, winter snowfall anomalies were all positive between 2020 and 2022, but still

within the 2001-2022 range. It must be also noted that snowfall anomalies (either winter or summer) are of rather small magnitude, especially compared to those in Western Nyainqêntanglha. These results indicate that air temperatures and solid precipitation in the 2020 and 2021 surveyed years were representative of recent climate conditions in Muztag Ata. We were, however, unable to find any significative correlation between climate variability and mass balance in Muztag Ata (Fig. 8a-f). Winter temperatures showed the highest correlation

coefficient (r = 0.57; p = 0.23; α = 0.05) amongst the investigated climate variables in Muztag Ata.

In Western Nyainqêntanglha, the surveyed 2020-2022 years all had positive summer temperature departures (around +0.6 °C), which were at the same time amongst the highest since in the last two decades (summer departure range: -1.2 °C to +0.7 °C). The 2021 and 2022 winter seasons also showed positive temperature anomalies (+1.6 °C and +0.6 °C; winter departure range: -3.3 °C to +3.0 °C; Fig. 8g-j). Of the last 16 years,

summer air temperatures have experienced positive anomalies. Likewise, the winter season has also shown positive temperature anomalies for 8 of the last 10 years. The years 2021 and 2022 were thus among the warmest and driest in Western Nyainqêntanglha in the last two decades approximately. We found a rather strong (though not significant at the 95% confidence interval) correlation between temperature anomalies and mass balance at both annual (r = 0.78, p = 0.08) and seasonal (r = 0.56, p = 0.25 and r = 0.69, p = 0.12 for summer and winter,

respecively) scale. In regard to solid precipitation, our surveyed years showed highly negative summer anomalies (up to ~16 mm/year). This likely had an impact on the overall negative annual snowfall anomalies, which in turn have been negative since 2014. Overall, the correlation between mass balance and solid precipitation was weak (r < 0.39) and non-significative either at annual or seasonal scale.


## 5. Discussion

### 5.1 Methodological constraints

#### 5.1.1 Internal consistency of the geodetic mass balance estimates

To verify the consistency of our high-resolution geodetic mass balance surveys at annual to seasonal mass balance, we tested the internal robustness of our geodetic surveys. We evaluated the residuals between accumulated vs. the sum of individual survey periods encompassing identical time intervals, in so-called 'triangulation' tests. This is

a way to measure the impact of using different inputs (*dDEM* grids, snow and ice distribution maps, and the resulting average material density) on geodetic mass balance values when evaluating similar periods yet at different time steps.

On a glacier-wide scale, the difference between the sum of the Muztag Ata 2022 summer and winter mass balances (-0.10 m w.e.), and accumulated 2022 annual budget (-0.14 ±0.22 m w.e.) yielded a difference of ±0.04 m w.e.

between both measurements. In the case of Western Nyainqêntanglha, this glacier-wide comparison showed no differences between the accumulated and added values (both -0.70 m w.e.). In addition, and on an individual glacier scale, we found a minimal discrepancy of ±0.01 m w.e. for Kekesayi Glacier, whilst a much larger difference was found for the Muztag Ata No. 15 (±0.28 m w.e.) mass budgets. Individually, Zhadang and Xibu glaciers in Western Nyainqêntanglha showed differences between 0.09-0.13 m w.e.a$^{-1}$.

Overall, we find the glacier-wide and (for the most part) individual differences to be well within the uncertainty ranges, and attribute the differences to the overall small differences in average density (which in turn derives from the snow and ice distribution) of the September 2021 (590 kg m$^{-3}$) and April 2022 Sentinel-2 (630 kg m$^{-3}$) snow and ice masks. In this sense, Pelto et al. (2019) showed how the spatial distribution of material densities has a larger impact on seasonal mass balances compared to the assumed density values themselves. In contrast, variable

density distribution has a greater impact when smaller areas are considered, e.g. in the case of Muztag Ata No. 15.

A second consistency test was carried out to test the impact of varying density assumptions in more detail, evaluating the accumulated 2020-2022 mass budget and the sum of all the individual annual periods. We calculated these differences using a single *dDEM* grid and a) an overall density of 850 ±60 kgm$^{-3}$ and b) a 3-year

weighted density (Muztag Ata = 774 ±60 kg m$^{-3}$; Western Nyainqêntanglha = 762 ±60 kg m$^{-3}$) for all the mass involved following Huss (2013) (Table 3). In Muztag Ata, the 2020-2022 glacier-wide estimate (-0.14 ±0.03 m w.e. a$^{-1}$ [scenario a] and -0.14 ±0.03 m w.e. a$^{-1}$ [scenario b]) was similar to the averaged mass budget of the individual periods (-0.08 ±0.21 m w.e. a$^{-1}$). The accumulated 2020-2022 mass balance of Kekesayi (-0.29 ±0.04 m w.e. a$^{-1}$ [a]; (-0.26 ±0.04 m w.e. a$^{-1}$ [b]) differed in ±0.16 and ±0.13 m w.e. a$^{-1}$, respectively, from the averaged

individual periods (-0.13 ±0.26 m w.e. a$^{-1}$). Differences were negligible in the case of Muztag Ata No. 15 Glacier.



In Western Nyainqêntanglha, the differences were much larger (-0.60 ±0.15 m w.e. a$^{-1}$ on average vs. -1.04 ±0.11 m w.e. a$^{-1}$ [scenario a] and -0.93 ±0.10 m w.e. a$^{-1}$ [scenario b]). Expectedly, even larger differences were found for Zhadang and Xibu glaciers, since higher thinning rates will have a greater impact on any given density value. The use of 3-year weighted densities of Huss (2013) tends to reduce the differences between the accumulated vs. added mass budgets, confirming that constant density values should be kept only for time spans >3 years, whereas a more thorough density inspection is preferable for shorter timescales (Belart et al., 2017; Klug et al., 2018).

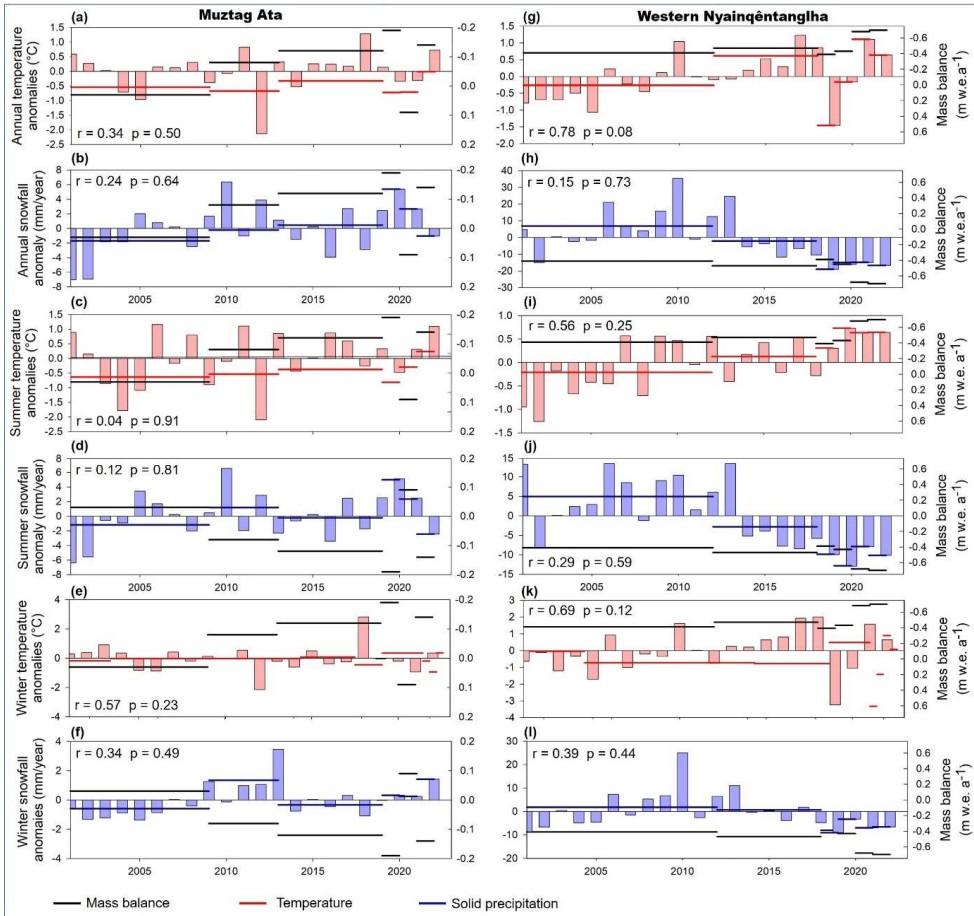

**Figure 8.** Evolution of geodetic glacier mass balance compared to annual and seasonal temperature and snowfall anomalies in Muztag Ata (a-f) and Western Nyainqêntanglha (g-l). In the temperature panels, the mass-balance values on the right axis have been reversed for a better interpretation.

### 5.1.2 Potential and limitations of Pléiades DEMs for assessment of short-term geodetic mass balance

Among the various uncertainty sources introduced in geodetic mass balance assessments at high temporal and spatial resolution, the ability to consistently map glacier surfaces, the density assumptions and especially DEM precision and overall quality have probably the highest impact on the final uncertainty estimate (Beraud et al., 2022). Here we discuss the suitability of our Pléiades DEM dataset to quantify both annual and seasonal mass balances over areas with contrasting glacier mass change rates.

Because 90% of the annual (seasonal) glacier elevation changes in Muztag Ata and Western Nyainqêntanglha range between -4.7 m to +3.3 m, highly accurate and precise DEMs are needed to account for the resultant small magnitude of glacier mass fluctuations. In Sect. 3.1.4 we reported that our DEMs errors are within previously reported biases of Pleiades DEMs used in geodetic mass balance assessments (e.g. Berthier et al., 2014; Denzinger et al., 2021; Falaschi et al., 2022). A fine coregistration between DEMs allows the minimization of vertical and



horizonal biases. This is especially important when utilizing DEMs acquired when seasonal snow is present off-glacier, but difficult to circumvent when surveying mass balance during accumulation periods. Under these conditions, vertical shifts ~1 m measured over stable terrain have been reported (Beraud et al., 2022). Glaciers close to balanced conditions (such as Muztag Ata) are particularly sensitive to an adequate quantification of systematic biases compared to glaciers with a much more negative mass budget, as a given small vertical DEM

adjustment may shift from slightly negative to slightly positive elevation changes and vice-versa.
Around our reported biases, the vertical precision (standard deviation on stable terrain) of our Pleiades time series in Muztag Ata and Western Nyainqêntanglha Pleiades were ±2.1 m and ±1.2 m on average, respectively (Table 2). DEM precision can be also described using the normalized median absolution deviation (NMAD), which is less sensitive to outliers compared to standard deviation (Dehecq et al., 2016). Independently from seasonal snow

conditions, the NMAD over off-glacier terrain was consistently around ±1.3 m in Muztag Ata, and varied between ±0.6 m and 1.2 m in Western Nyainqêntanglha. These standard deviations and NMAD values are in the same order of magnitude as other very high resolution DEMs used in glaciological applications (Berthier et al., 2014; Belart et al., 2017; Pelto et al., 2019; Beraud et al., 2022). Comparatively, the lower precision in the Muztag Ata DEMs result in relatively larger uncertainties in particular cases (Table 3). This becomes more problematic due

to the short temporal baseline, and when glaciers are close to in-balance conditions. With these caveats in mind, our annual and seasonal geodetic mass balance estimates were robust and consistent with the results retrieved from the Glacier Index (see Sect. 5.2) and hence can provide valuable insight into glacier accumulation regimes in High Mountain Asia.

**5.2 Discerning accumulation regimes**

The heterogeneity of regional climates displayed in High Mountain Asia, from monsoon-dominated regions with abundant precipitation occurring in the warm season to areas where westerly-induced, winter precipitation prevails, results in diverse glacier behaviour and sensitivity to climate drivers (Bolch et al., 2012; Sakai and Fujita,

2017). This, added to the sparsity of glaciological in situ observations, poses a fundamental challenge for understanding the response of the region's glaciers to climatic change. More so, the inadequate number of meteorological stations placed at glacier altitude often means that only distant, lower elevation instrumental records are available for glacier studies of local extent. Climatic conditions at low-lying valleys, however, can deviate substantially or even be entirely unrepresentative of those met by glaciers at high elevation (Wortmann et

al., 2018). Consequently, a major gap for a better understanding of glacier mass changes in response to climatic change and their frozen water storage lies in the sparse knowledge of climate conditions at glacier elevations in general, precipitation and snow accumulation in particular (Miles at el., 2021; Vishwakarma et al., 2022).
Huang et al. (2022) showed important discrepancies in accumulation regimes for some regions across High Mountain Asia between 2015 and 2018, as derived from gridded climate datasets (including APHRODITE, ERA5

and HARv2 products) on the one hand, and the SAR–derived Glacier Index on the other hand. In the specific case of the Muztagh Ata area, there is conflicting evidence about whether glaciers are of winter- or summer-accumulation type. Data from the nearest Taxikorgan station indicate that precipitation occurs mainly during summer time [April-September] (Zhu et al., 2018a), so that a summer accumulation regime may be assumed for the Muztag Ata glaciers. Indeed, Zhu et al. (2018a, b) pointed out as summer precipitation as the main driver of

the Muztag Ata No. 15 Glacier mass balance, and that higher amounts of solid precipitation in summer compared to the cold season were mostly responsible for the overall positive budget between 1998 and 2012. However, these authors also showed that between 1980 and 1997, and in spite of heavier precipitation in summer compared to winter, other processes affecting glacier mass balance (e.g. snowmelt, sublimation) resulted in a more negative budget in the summer season.

In contrast to the Taxikorgan instrumental records in Muztag Ata, which show higher amounts of precipitation during the summer months, Maussion et al. (2014) used HAR data and found that overall, the Pamir region experiences mostly winter precipitation. Huang et al. (2022) found a conspicuous mismatch between the (transitional) accumulation regime derived from a) gridded climate datasets and climate stations off-glacier and b) the winter accumulation–type according to the SAR analyses. The authors noted the substantial differences in

snow accumulation measured at ~7000 asl (605 mm a[-1]) and those recorded at Taxikorgan station between 1960 and 2002 [60-70 mm; Duan et al. (2015)], suggesting that the upper part of Muztag Ata [(an anomalously high peak in the region, Seong et al., 2009)] may be affected by a different atmospheric circulation system compared to the valley bottoms (i.e. at Taxikorgan station elevation). Sakai et al. (2015) used a summer- to annual-precipitation ratio from APHRODITE data and Temperature-Precipitation plots as indicators of glacier sensitivity

to climate changes and put the glaciers in Eastern Pamir within the winter accumulation (and less sensitive) envelope. Although our Pléiades datasets have allowed us to retrieve mass changes for two seasonal intervals only, the geodetic estimates of the glacier-wide Muztag Ata 2022 winter (+0.21 ±0.24 m w.e.) and summer (-0.31 ±0.15 m w.e.) mass balance favor a winter accumulation–type scenario for the Muztag Ata glaciers.



In stark contrast to Muztag Ata and Eastern Pamir, the summer Monsoon-dominated glaciers in Western Nyainqêntanglha have a low annual temperature range, are highly sensitive to temperature and precipitation changes, and are therefore prone to strong mass losses (Sakai et al., 2015). There is good consensus in characterizing most glaciers in the area as summer accumulation–type (Fujita and Ageta, 2000; Maussion et al., 2014; Sakai et al., 2015; Huang et al., 2022). The mass balance of Zhadang Glacier was found to be particularly sensitive to the onset of the Monsoon period (Kang et al., 2009; Mölg et al., 2012), though mid-latitude westerlies too drive its mass balance (Mölg et al., 2014). Our geodetic results for the Western Nyainqêntanglha glaciers showed high ablation rates prevailing over accumulation in the 2022 summer season (-0.66 ±0.07 m w.e.a$^{-1}$) and little mass recovery during the winter season (-0.04 ±0.27 m w.e.a$^{-1}$). The 2009-2011 and 2008-2013 mass balance of Zhadang Glacier modelled by Zhang et al. (2013) and Zhu et al. (2018a) support our findings, indicating high mass loss during the ablation (summer) season, but minor losses during the cold (winter) season.
.

### 5.2.1 Insight from climate records and the Glacier Index

Establishing a clear link between the investigated climate datasets and variables proved to be a challenging task. Several of the stronger correlations between glacier mass balance and temperature and solid precipitation were not significant, which is most probably due to the small number of mass balance observations (n max = 7 in the 1967-2022 correlation tests). In Muztag Ata, our results showed a stronger correlation between glacier mass budget and solid precipitation compared to air temperature (though we stress here that neither correlations were significant). Consequently, snowfall appears to be a stronger mass balance driver in relation to air temperature. This is a relevant development, as mass balance in colder and drier environments are influenced by solid precipitation to a greater extent (Zhu et al. 2018a, b).
We observed a different scenario in Western Nyainqêntanglha. Correlations were much stronger (and significant) between mass balance and temperature anomalies in comparison to solid precipitation. This is not surprising, since summer temperatures have a greater impact on glacier mass balance in more humid climatic regions such as Western Nyainqêntanglha (Zhang et al., 2013, Bhattacharya et al., 2021). Mölg et al., (2014) showed how summer precipitation determines the annual mass balance of summer accumulation glaciers in the area. Summer temperature, however, modulates the solid/liquid precipitation ratio during the summer season, when most of the solid precipitation occurs in the region.
The intricacies and limitations of determining accumulation regimes over our investigated glaciers based on the available climate records reveal the need for additional complementary approaches. To support our geodetic and climate data findings, we implemented the Glacier Index, which bypasses the need for climate data altogether.
The wet snow area– and firn area–ratio showed different patterns and trends in Muztag Ata compared to Western Nyainqêntanglha during the 2019-2022 period (Figure 7). In Western Nyainqêntanglha, the annual wet snow area are higher than the firn area–ratios, meaning that there is more accumulation in summer than in winter. The opposite was found in Muztag Ata, where the firn area–ratios are higher than the late summer wet snow area ratio. This is in turn indicative of higher accumulation during winter in Muztag Ata. In addition, the firn area–ratio is much lower in Western Nyainqêntanglha compared to Muztag Ata, and it is therefore not surprising that glacier mass balance has been much more negative for at least the last six decades (Bhattacharya et al., 2021 and references therein, see Table 4). These results agree well with our geodetic mass balance results for the annual and seasonal intervals in the 2020-2022 sampled period (Table 3), which support winter and summer accumulation–type regimes in Muztag Ata and Western Nyainqêntanglha, respectively.
Despite the overall good agreement between the proposed accumulation regime types based on both the geodetic and Glacier Index approaches, we report a minor caveat. The wet snow area–ratio in Western Nyainqêntanglha indicates that the year 2020 should be the most negative mass balance year in the study period (-0.43 ±0.16 m w.e.a$^{-1}$ as per our geodetic estimate), whereas the geodetic results suggest 2022 (-0.70 ±0.22 m w.e.) as the most negative year. A possible explanation lies in the differences in the acquisition dates between the Pléiades and Sentinel-1 satellite images used in the geodetic and Glacier Index approaches. The Glacier Index utilizes SAR imagery acquired from July 20 to September 10 each year to guarantee wet snow conditions over accumulation areas (whereas in late September and October, snow may be dry for areas above 5,000 m asl due to low temperatures). On the contrary, the 2020 Pléiades scenes employed in the geodetic method were acquired on 8 October. Our climate data, however, shows that in Western Nyainqentanlha, more snow fell during September 2020 compared to August in 2020, which will lead to a lower wet snow area–ratio.

### 5.3 Mass balance differences between both study sites and in the long-term perspective

Our mass balance estimates between 2020 and 2022 for the Muztag Ata massif suggest a greater variability in mass budget than previously acknowledged in the region (Fig. 9a). During the past six decades, Bhattacharya et al. (2021) found mass balance rates between +0.03 ±0.10 m w.e. a$^{-1}$ and -0.14 ±0.10 m w.e. a$^{-1}$. On annual time



steps, our values ranged between +0.09 ±0.27 m w.e. a$^{-1}$ and -0.19 ±0.14 m w.e. a$^{-1}$, suggesting a greater variability
(Table 3). Averaged between 2020 and 2022, the glacier-wide mass balance of Muztag Ata (-0.08 ± 0.20 m w.e.
a$^{-1}$) is similar to the 2013-2019 mean [-0.12 ±0.09 m w.e. a$^{-1}$; Bhattacharya et al. (2021)]. This suggests an ongoing
trend of slight mass loss in the Muztagh Ata massif for the last three years, meaning a continuation of the very
slight mass loss rate observed for most of approximately the last decade (Bhattacharya et al., 2021). A number of
studies (Holzer et al., 2015; Lv et al., 2020) report slight, but insignificant positive values between 1999 and
760 2015/2016, whilst Bhattacharya et al. (2021) found in-balance conditions since 1973.
On an individual glacier basis, glaciological measurements exist for the small (~1.1 km$^2$) Muztag Ata No. 15
glacier (Fig. 1) between the years 2002 and 2014 (Yao et al., 2012 and unpublished data; Holzer et al., 2015). The
mean 2005-2014 mass balance of this glacier was +0.11 m w.e. a$^{-1}$, which is similar to our +0.08 ±0.20 m w.e. a$^{-1}$ 2020-2022 average.
In contrast to Muztag Ata, glaciers in the Western Nyainqêntanglha range show much higher mass loss rates. The
2022 mass balance of -0.70 ±0.22 m w.e. is not only much more negative than the -0.08 ± 0.20 w.e. a$^{-1}$ 2020-2022
average found in Muztag Ata, but also represents a negative maximum for the Western Nyainqêntanglha area
itself since the late 1960's (Fig. 9b). Indeed, the mass budget has not been more negative than -0.65 ±0.08 m w.e.
a$^{-1}$ approximately the last five decades (Hugonnet et al., 2021 and other sources; Table 4). It must be noted,
however, that the sampled glaciers vary among these studies, and the mass budget could have differed if data from
different temporal scales were considered.
Zhadang Glacier, which had a glaciological mass balance program running between 2006 and 2017 (Yao T, pers.
com.), was one of the glaciers with the most negative mass budget in the Western Nyainqêntanglha range. The
2020-2022 mean mass balance was -0.87 ±0.28 m w.e. a$^{-1}$), depicting a high mass loss rate consistent with previous
modelled (–1.0 m w.e. a$^{-1}$ [2001-2011], Huintjes et al., 2015; -0.91 w.e. a$^{-1}$ [2010-2012] approx.; Zhu et al., 2015)
and in situ (-1.2 m w.e. a$^{-1}$ to -1.6 m w.e. a$^{-1}$ [2009-2014] estimates; Zhang et al., 2013 and 2016). Surprisingly,
though, remote sensing–based estimates have yielded significantly less negative mass balances encompassing
longer but overlapping periods (-0.50 ±0.17 m w.e. a$^{-1}$ [2000-2014], Li and Lin. (2017); -0.60 ±0.19 m w.e. a$^{-1}$
[2000-2017], Ren et al., (2020)). The average, 2006-2017 in situ mass balance of Zhadang Glacier (-1.35 m w.e.
a$^{-1}$; Yao et al., 2012 and unpublished data) was found to be closer to our estimate, which may suggest that
penetration correction for mass balance estimates based on DEMs originating from SAR data are inadequate in
the area.





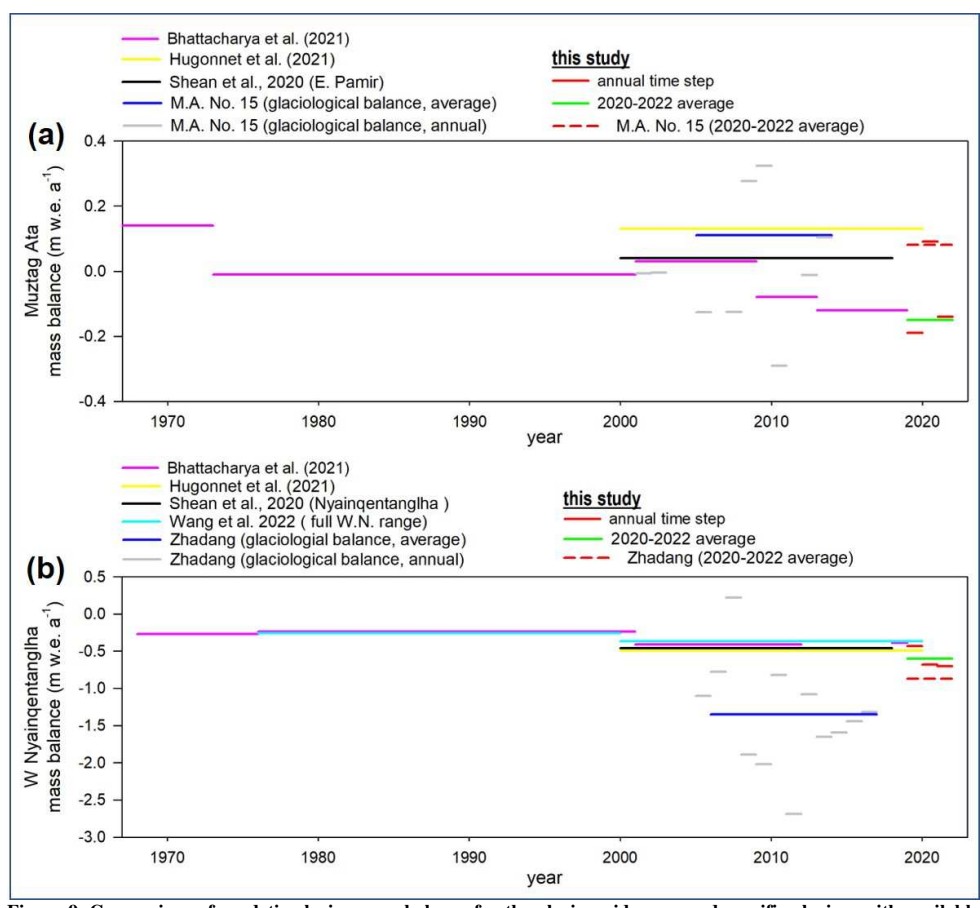

**Figure 9. Comparison of geodetic glacier mass balance for the glacier-wide area and specific glaciers with available glaciological mass balance records in Muztag Ata (a) and Western Nyainqêntanglha (b).**



**Table 4.** Mass balance estimates in Muztag Ata and Western Nyainqêntanglha based on optical and SAR–derived DEM differencing.

| | Time period | Mass balance (m w.e.a$^{-1}$) | source |
|---|---|---|---|
| **Muztag Ata** | | | |
| | 1967-1973 | -0.14 ±0.10 | Bhattacharya et al. (2021) |
| | 1973-2001 | -0.01 ±0.06 | |
| | 2001-2009 | +0.03 ±0.10 | |
| | 2009-2013 | -0.08 ±0.12 | |
| | 2013-2019 | -0.12 ±0.09 | |
| | 1967-2019 | -0.06 ±0.07 | |
| | 1973-1999 | −0.04 ±0.42 | Holzer et al. (2015) |
| | 1999-2009 | +0.04 ± 0.45 | |
| | 2009-2013 | −0.07 ±0.53 | |
| | 1973-2009 | −0.03 ±0.33 | |
| | 2000-2004 | +0.19 ±0.08 | Hugonnet et al. (2021) |
| | 2005-2009 | +0.13 ±0.06 | |
| | 2010-2014 | +0.09 ±0.07 | |
| | 2014-2019 | +0.07 ±0.08 | |
| | 2000-2019 | +0.13 ±0.10 | |
| | 2000-2015/2016 | +0.16 ±0.03 | Lv et al. (2020) |
| | 2000-2018 | +0.21 ±0.06 | Shean et al. (2020) |
| **W. Nyainqêntanglha** | | | |
| | 1968-1976 | -0.27 ±0.11 | Bhattacharya et al. (2021) |
| | 1976-2001 | -0.24 ±0.13 | |
| | 2001-2012 | -0.41 ±0.11 | |
| | 2012-2018 | -0.47 ±0.15 | |
| | 2018-2019 | -0.39 ±0.18 | |
| | 1968-2019 | -0.32 ±0.09 | |
| | 2000-2004 | -0.31 ±0.08 | Hugonnet et al. (2021) |
| | 2005-2009 | -0.44 ±0.06 | |
| | 2010-2014 | -0.58 ±0.07 | |
| | 2014-2019 | -0.65 ±0.08 | |
| | 2000-2019 | -0.49 ±0.17 | |
| | 2000-2018 | -0.51 ±0.11 | Shean et al. (2020) |
| (total W. Nyainqêntanglha) | 2000-2020 | -0.37 ±0.12 | Wang et al. (2022) |
| (total W. Nyainqêntanglha) | 2000-2017 | −0.30 ±0.19 | Ren et al. (2020) |
| (total W. Nyainqêntanglha) | 1975-2000 | −0.25 ±0.15 | Zhou et al. (2018) |
| (total W. Nyainqêntanglha) | 2000-2013/2014 | −0.23 ±0.13 | Li and Lin (2017) |
| (total W. Nyainqêntanglha) | 2000-2013/2014 | −0.30 ±0.07 | Zhang and Zhang (2017) |

## Conclusions

In this study we have assessed the capability of very high-resolution Pleiades DEMs to quantify glacier mass
balance over short (annual to seasonal) time steps over regions in High Mountain Asia that have shown contrasting
mass balance trends in the last few decades. We find intrinsic DEM quality to be the main source of uncertainty
in geodetic mass balance estimates, rather than varying seasonal snow conditions among the two study sites.
Indeed, the mean vertical precision (NMAD) of Muztag Ata and Western Nyainqêntanglha DEMs were ±1.3 m
and ±0.6 m to ±1.2 m, respectively. This, in turn, is along the lines of previously reported values of other very-
800 high resolution DEMs used in similar geodetic mass balance estimations worldwide.
Two main conclusions can be drawn from the internal consistency tests (utilized to evaluate the differences
between accumulated vs. the sum of individual periods). On one side, the tests confirm that the usage of a semi-
automated approach (such as ASMAG) for the identification of different glacier surfaces (ice, snow/firn) (and
thus the distribution of assumed material densities) is a robust approach when quantifying glacier mass balance at
805 a multi-glacier scale. Greater differences on individual glaciers, however, suggest that manual mapping by visual
interpretation might be preferable, especially on small glaciers. Concurrently, the tests reaffirm that using time-
weighted densities reduced the residuals between accumulated vs. added mass budgets, stressing the necessity of
implementing detailed density distributions over constant values for intervals longer than 3 years.



Mean annual (-0.11 ±0.21 m w.e. a⁻¹) mass balance estimates in Muztagh Ata between 2020 and 2022 point at the continuation of the slight mass loss trend after a period of apparently balanced conditions. On the contrary, the glacier mass balance in Western Nyainqêntanglha reached a negative peak of -0.70 ±0.22 m w.e. a⁻¹ in 2022, which represents a new maximum over the last six decades. The analysis of ERA5-land 1950-2021 temperature and GPM 2001-2021 solid precipitation anomalies confirm that the years 2020 and 2021 as a) average years in terms of air temperature and snowfall in Muztag Ata and b) particularly warm and dry years in Western Nyainqêntanglha.

The 2022 winter and summer mass balance estimates (+0.21 ±0.24 m w.e. and -0.31 ±0.15 m w.e., respectively) in Muztag Ata suggest a winter accumulation type, whilst mass losses of -0.04 ±0.27 m w.e and -0.66 ±0.07 m w.e. in the winter and summer seasons, respectively, confirm a summer accumulation–type regime in Western Nyainqêntanglha, with ablation prevailing over accumulation in the summer (ablation) season. With the SAR–based Glacier Index, we indirectly validated our geodetic mass balance estimates and the derived inferences that were made in relation to glacier accumulation types in Muztag Ata and Western Nyainqêntanglha. Whilst the Index does not provide a specific mass balance estimate per se that can be directly compared against geodetic (or glaciological) results, it can provide further insight into accumulation regimes in poorly known regions. Moreover, it fully bypasses the need for instrumental or reanalysis records, which are often unavailable or unrepresentative of climate conditions at glacier locations. Further geodetic and glaciological mass balance measurements in combination with such an Index will open new possibilities in glaciological research.

Based on the above, we conclude that our DEM time series and mass budget estimates proved to be consistent for making reliable, short-term estimations of glacier mass balance using a remote sensing–based approach. The ever-increasing number and availability of very-high resolution optical satellites (with stereo capability and relatively short revisit time) will allow for increasing the number of glaciers in isolated regions that can be readily monitored.

**Code availability**

We are grateful to Etienne Berthier for kindly providing the along-track bias correction and the elevation change uncertainty "patch" tools.

The DEM generation code from Shean et al. (2016) is available from the corresponding github repository at https://github.com/NeoGeographyToolkit/StereoPipeline. The DEM coregistration code (Nuth and Kääb, 2011) is available at https:// github.com/GeoUtils/geoutils.

**Data availability**

The Sentinel-2 scenes were obtained from the USGS EarthExplorer data poll (https://earthexplorer.usgs.gov/) and the ESA Copernicus Open Access Hub (https://scihub.copernicus.eu/). The Sentinel-1 and Landsat OLI scenes used in the Glacier Index are available online on Google Earth Engine through the Google Cloud Storage and the Google Cloud public data program. The elevation change grids from Bhattacharya et al. (2021) are available from PANGAEA at https://www. pangaea.de and https://www.mountcryo.org/datasets/. The elevation change maps from Hugonnet et al., 2021 are publicly available at https://doi.org/10.6096/13. The ERA5-Land data was downloaded from the Copernicus Climate Data Store at https://cds.climate.copernicus.eu/.

**Author contribution**

DF and TB designed the study. OK contributed to the study design. DF led the study, processed and analyzed data and wrote the manuscript. AB, KM and LH, PR processed and analyzed data. AB, KM, LH, OK and TB contributed to the writing of the manuscript. TB and TY secured the funding. The order of authors from third to seventh is in alphabetical order.

**Competing interests**

Some authors are members of the editorial board of The Cryosphere. The peer-review process was guided by an independent editor, and the authors have also no other competing interests to declare.

**Acknowledgements**

We are grateful to CNES/Airbus DS for the provision of the Pléiades satellite data within the ISIS program for reduced costs. Pleiades © CNES 2020/2021/2022 and AIRBUS DS. We would like to thank Christine Baron, Sylvie Boureausseau, and the whole Airbus Intelligence Team for their assistance during the acquisition of the



Pléiades imagery. AB acknowledges funding by the Science & Engineering Research Board (SERB), Department
of Science & Technology (DST), India (grant no. CRG/2021/002450).

**Funding**

This study was supported by the Strategic Priority Research Program of Chinese Academy of Sciences
(XDA20100300) and the Swiss National Science Foundation (200021E_177652/1) within the framework of the
DFG Research Unit GlobalCDA (FOR2630) and benefited from the research cooperation within the Dragon 5
program supported by ESA and NRSCC (4000136930/22/I-NB).

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
