# Peer review of "Annual to seasonal glacier mass balance in High Mountain Asia derived from Pléiades stereo images: examples from the Pamir and the Tibetan Plateau"

_The Cryosphere, 2022_

## Referee Comment (RC2)

**General comments**

This manuscript discusses the potential and limitations of annual and seasonal geodetic mass balance estimates retrieved from Pleiades stereo images, over two regions in Muztagh Ata in eastern Pamir and Western Nyainqêntanglha. Both glacier regions are in High Mountain Asia representing different melting regimes. The topic is of high importance to further understand how the geodetic approach of measuring glacier mass balance using very high resolution optical imagery like Pleiades can be used to estimate seasonal glacier mass budgets without the need for in situ observations in a region with high elevations and difficult topography. In addition, the authors use other remote sensing data in their study. Sentinel-1 SAR and Landsat OLI was used for defining firn area and wet snow zones, both included in a Glacier Index for defining accumulation regimes. Sentinel-2 was used for retrieving glacier surfaces. In addition, the results are compared with climatic data and other geodetic mass balance estimate studies.

The manuscript is cited well and with a clear language. The datasets and method chapter is well described. The study is well compared with other studies and the authors illustrate this in tables and figures. The result chapter gives in general a good overview of the findings, and in the discussion chapter the authors compare their results with other relevant studies. This is of high importance especially for the Mustag Ata region due to the indication of a change in glacier mass balance regime to a slight mass loss.

**Spesific comments**

Belart et al.,2017 states that the bulk snow density is most likely the largest contributor to uncertainty in winter geodetic mass balance. In the chapter "3.1.5 Bulk density" in the manuscript, the authors refer to other studies and assumes uncertainty values, e.g. that the density values referred to are from snow pits. It is limited data of snow density in the study regions. The authors should discuss this uncertainty more and be clearer on the consequences it might have for the results.

The use of the surface classification of snow, ice and firn from Sentinel-2 can be clarified in chapter "3.1.2 Classification of snow and ice using Sentinel-2 scenes". The chapter gives a good description of how the analysis is done, but it can be elaborated in the start of the chapter what use these data has for the geodetic glacier mass balance and why it is important.

Some of the sentences and text includes to many parentheses with additional information or clarifications. A suggestion is to go through the whole manuscript text in general and write shorter sentences that are clearer and easier to read. Here is an example to illustrate: L: 595: "Overall, we find the glacier-wide and (for the most part) individual differences to be well within the uncertainty ranges, and attribute the differences to the overall small differences in average density (which in turn derives from the snow and ice distribution) of the September 2021 (590 kg m-3) and April 2022 Sentinel-2 (630 kg m-3) snow and ice masks.". The parentheses are sometimes randomly placed, e.g., "(630 kg m-3)" should maybe be placed after "April 2022"?

The authors give a good overview of how they used the Glacier Index of Huang et al. (2022), to find glacier accumulation regimes. However, it is no error estimations of the retrieval of firn and wet snow areas from the remote sensing data, and this should be elaborated.

**Technical comments:**

L: 91: "The major aims of this paper are therefore to investigate the potential and limitations of geodetic mass balance estimates derived from VHR Pleiades satellite data (using 5 DEMs over the 3-year period 2020-2022)." Should it be data between 2019 and 2022? (Ref. table 1).

L:425-429: "To account for different glacier areas between the study sites, we express firn and wet snow areas on each region as a fraction of the total glacier area (hereafter referred to as firn area ratio and wet snow area ratio). This ratio can vary to a great extent across different geographic regions

through time, whilst interannual variations of the firn area ratio remain relatively small." Which ratio do the authors refer to when compared to the firn area ratio? Glacier index, I? A suggestion to rewrite sentences.

L: 440: "First, the Landsat scenes are used to recognize debris-covered and debris-free areas (ice and snow) on glaciers surface applying a threshold to the previously computed Normalized Difference Snow Index [NDSI] (Bruns et al., 2014).". The authors describe the use of Landsat-data for glacier surface characteristics. Why was not Sentinel-2 data used for this purpose as this satellite sensor has higher spatial resolution?

L: 830: "The ever increasing and availability of very-high resolution optical satellites (with stereo capability and relatively short revisit time) will allow for increasing the number of glaciers in isolated regions that can be readily monitored.". Can the authors clarify which satellite sensors they are referring to in the last sentence of the conclusion? It is not planned many optical missions with stereo capability in the future. Consider to be more specific and give examples of missions you refer to.

L: 1015: Wrong year in Huang et al., 2022 in reference list.

Figure 1: It is not clear to me which glaciers are "investigated glacier" in the figures. Is it all of them? Consider changing color or outline and rewrite to "investigated glaciers".

Figure 4 and 6: Cannot really see the dh variation in the figures. A suggestion is to make the figures larger, and subsets of the individual glaciers discussed in the text can also be included.

Figure 8: Improve the representation and better the resolution of the plots.

Figure 9: It is hard to see the difference between the lines indication "this study" in the plots. Consider changing color on either "annual time step" or the individual glaciers.

---

## Referee Comment (RC3)

**Annual to seasonal glacier mass balance in High Mountain Asia derived from Pléiades stereo images: examples from the Pamir and the Tibetan Plateau**

Falaschi et al. TC Discussion
Review by César Deschamps-Berger

This article presents a time series of three annual and two seasonal mass balances for two glacerized massifs in High Mountain Asia. In line with previous studies, it is found that the Pamir glaciers have a mass balance close to equilibrium between 2019 and 2022 while the glaciers from the Tibetan Plateau have a negative mass balance. Various satellite products (Pléiades, Sentinel 1 and 2) were combined to identify the surface elevation change and the accumulation and ablation areas. The snow/firn density and the firn densification are taken into account based on field measurements. The authors put the measured mass balance in longer time scale context and improve the description of the glacier accumulation regimes. It is a detailed work which combines advanced methodologies with good knowledge of the region. I think this article will be a valuable contribution after the following main concerns are addressed.

1. The winter elevation changes show disturbing patterns (Figure 4, 6 and blue lines in Figure 5). In Western Nyainqêntanglha, elevation loss in winter are stronger at the highest elevation. On the contrary, elevation loss in winter are measured in the Muztag Ata between 6000 m asl and 7000 m asl with areas of elevation gain below and above.
Are these elevation change significant or within the calculated uncertainties?
What process could explain such altitudinal distribution of elevation change?
I wonder if it could be related to remaining errors in the elevation change map (jitter correction, gap filling, shaded areas, see areas highlighted below). Jitter correction might not be perfect due to the lack of stable terrain, especially for Muztag Ata. Better highlighting and explaining these errors might impact the conclusions on the accumulation regimes (e.g. L496, 5.2) and on the sources of uncertainty of the mass balance estimation (e.g. L595-601 and 5.1.2).

[Figure]

[Figure]

Left: Figure 4. c. Muztag Ata; Right: Figure 6.c. Western Nyainqêntanglha,

2. I estimate that this manuscript would result in an article of more than 20 pages. The article readability would benefit from being more concise. I would advise the authors to revise the manuscript with that in mind. Some of my minors comment should help to gain space (e.g. L89, L108, L125, L194, L283...).
If a radical choice was to be made, moving to supplement or removing the parts about the correlation of mass balance with climatological variables might not alter the value of the article. It is almost a distinct topic from the core of the article (see title). The data used partly come from other articles and finally, few (or no) significant correlation are found.

3. The quality of the figures should be improved. Almost all the plots with lines are hard to read due to the style and colour of the lines. For instance, it is very hard to distinguish several lines of Figure 5, the

individual glaciers in Figure 7, Mass balance from Solid precipitation in Figure 8. Select better colour and line style.

**Minor comments and suggestions**

L26. Pl**é**iades. Throughout the text.

L31. delete « previously observed »

L32-33. delete « *on average* ». « *mean*» is already stated at the beginning of the sentence.

L33. « *increased* » compared to what?

L33. Why is Western Nyainqêntanglha qualified here as summer accumulation type when summer mass balance (-0.66 m w.e.) is more negative than the winter one (-0.04 m w.e.)? Besides, this conclusion (Western Nyainqêntanglha being summer accumulation type) seems based on other studies in the dedicated Discussion paragraph (L696-698).

« *The 2022 winter (+0.21 ±0.24 m w.e.) and summer (-0.31 ±0.15 m w.e.) mass budgets in Muztag Ata and Western Nyainqêntanglha (-0.04 ±0.27 m w.e. [winter]; -0.66 ±0.07 m w.e. [summer]) suggest winter and summer accumulation-type regimes, respectively.* » I suggest rephrasing as: « The seasonal mass balance in Muztag Ata (winter: XX m w.e., summer : XX m w.e.) and Western Nyainqêntanglha (winter: XX m w.e., summer : XX m w.e.) suggest... ».

L80. Not only WorldView-2 (see Shean et al., 2020). Simply put « WorldView ».

L86. Des**c**hamps-Berger.:)

L89. «*(e.g. Ice, Cloud and land Elevation Satellite-2 -ICESat-2)*» ICESat-2 is not further used. Give only the acronym.

L92. Quit the brackets.

L96. «*displayed dissimilar mass change rates.*» precise over which epoch, otherwise it sounds like an article's result is given away.

L97. « *mass balance for longer period* » Longer than what?

L104. In 2.1., provide the max elevation as in 2.2. Tell if the whole massif is covered.

L108. Delete «*(along with the nearby Kongur Shan mountains).*» It is never mentioned again.

L124. «*glacier *»

L125. «*Glaciers in the arid NW Tibetan Plateau are predominantly continental-type, cold–based (with their basal part entirely below the pressure melting point) and receive little precipitation.*» Might be deleted? All these informations are repeated in the next lines.

L131-134. Give periods for each mass balance epoch, confusing otherwise.

L147. « *~230 km in length reaches* » => « extend over ~230 km in length and reaches... »

L163. « *in their model* »? An energy balance model is not a source of information about precipitation.

L165. « > » => « more than ». Delete « here ».

L168. Why use []? And why « accelerated »? compared to what?

L170. Keep giving MB with two digits precision « (–1.0X m) ».

L172. « *Zhadang glacier*» first time it is mentioned. Introduce shortly the glaciers of interest (size, specificities).

L192. Delete «*relatively*».

194. « *separated in time on most occasions (Table 1). In all cases, partial acquisition dates were no more than 2 weeks apart.* » => « separated by two weeks in the worst case (Table 1).»

L195. Was the DEM produced in one run from the raw stereo images to a high-resolution DEM? A common practice to reduce errors is to first project the images on a low-resolution DEM (idea for future work).

L196. « *implementing* » => « using »

L197. Why «although»? The first part of the sentence refers to a method of this work, the second refers to results from other studies. Grammatically hard to understand.

L200. « *Pléiades DEMs are currently amongst the most common very high resolution DEMs used in geodetic mass balance assessments*». Maybe not necessary as there are anyway few high resolution photogrammetric satellite and studies exists with WorldView (Shean et al., 2020).

L203. « *beyond the higher spatial resolution,* » not true for WorldView satellites, could be deleted.

L205. «*saturated areas*» => « areas prone to saturation »

L209. Did you request that the images were acquired with reduced Time Domain Integration (TDI)? It can help preventing saturation (Deschamps-Berger et al., 2020).

L223. I understand that the lower resolution saves computational time but less the number of images. Is the opening and closing of the images really a bottleneck in the treatment?

L232. delete « *see also* »

L235. «*To this ends, it implements an automatic threshold to the near-Infrared*» => « It determines automatically a threshold for the NIR band values ».

L241. Move the « Muñoz » citation after « daily data ».

L257. Delete « see e.g. ».

L260. Move the UTM info somewhere else more generic.

L265. In future work, you might want to first co-register your reference DEM to an external reference (e.g. Copernicus DEM) to ensure a better absolute co-registration.

L269. «*reprocessed 2019 Pléiades DEM from Bhattacharya et al. (2021)*» is confusing. Was the 2019 DEM eventually calculated like the others of this study? Then, the Bhattacharya reference could be deleted. Maybe, the link between this study and Bhattacharya et al. (2021) should be better explained in introduction. At least better introduce the Bhattacharya et al. (2021) study as some products are used here (L 472).

L270. Which metric is used to correct the vertical biases? The mean, the median?

L271. Delete « *(reference)* »?

L275. «*implemented*» to be clarified. It sounds like you implemented the code (i.e. wrote the code). However in the acknowledgement  the tools of Etienne Berthier are mentioned. Also note that it is a different method than the one used in Deschamps-Berger et al. (2020). The first one fits polynomial functions to the residual while the second calculates and modifies the Fourrier transform spectrum of the residual. Make sure which one was used or implemented.

L277. « *on- and* » missing blank

L283. « *We chose a 3-cell buffer so as not to remove valid cells from the original dDEM grids.* » This kind of sentence could be deleted to make the article more concise.

L284. « *mosaiced*»? merged? How are managed areas where there is overlap between tiles (concisely)?

L305. « *in the energy balance model comprising Muztag Ata N15 and Zhadang glaciers (in Muztag Ata and Western Nyainqêntanglha districts, respectively) by Zhu et al. (2018a). According to the authors, this density value was retrieved from snow pits.*» Maybe no need to repeat the districts if the glaciers are introduced before. Why mention the energy balance model if the density actually come from pits measurements? Merge sentences concisely.

L318. « time interval » dt might be more clear than *t*?

L321. Eqn (2) « k » is missing. Replace «i»?

L326. It is not clear what the « i » of Ai refers to? Ice? The previous sentence mentions ice and snow areas.

L330. I cannot find easily in this paragraph if the correction is applied on a pixel scale or at the firn area scale? Please clarify.

L346-347 () [] to homogenise

L346. «*most recent elevation grids available* » => « most recent elevation **change** grids »? Since period are provided in brackets (2013-2019, 2018-2019).

L374. « *is not to be expected* » => « is not expected ».

L395. what does « *addressed* » mean? Calculated, defined?

L397. By construction, the mean elevation difference over stable terrain is zero or close to it. Depending whether the mean or median elevation residual was used for vertical coregistration (to be added in 3.1.4). This underestimate potential systematic error. Besides, I do not find sigma_sys further in the uncertainty calculations.

L410. Delete « *see also* »

L415. Eqn 7. What is « f »?

L424. Huang et al. (2022) missing in the bibliography.

L433. The **i**ndex

L433. Please repeat that firn area is measured at the end of the winter and the wet snow area at the end of the summer. Or clarify this point if I misunderstood.

L449. « the influence of XX to YY » Is this grammatically correct? Otherwise replace «*to govern*» by «on».

L452. Put the citations at the end of the sentence. Why mentioning the period of availability of the data? Idem L463.

L456. Delete « *either* »?

L458. Cite APHRODITE along with ERA5, HARv2 in the previous sentence and delete « (APHRODITE, ERA5, HARv2) » in this one.

L459. «*(or instrumental records*» to delete

L461. «*Mu**ñ**oz-Sabater*»

L463. « *to monthly time step* » Was ERA5 initially at a daily or monthly time step? Mention it in the previous sentence.

L468. «*found*» => « calculated »?

L468. Redundancy with « mean » and « average » in the same sentence.

L472. « *added the geodetic mass balance values in Bhattacharya et al. (2021)* » see comment about L269.

I understand that data from Bhattacharya et al. (2021) are not at a yearly or seasonal resolution. Do you think that mixing periods of different durations could have an impact on the correlation calculated?

L481. Cite « Table 3 » only once in this paragraph.

L483-484. Please provide value in brackets for each year. Or alternatively do not provide any.

L486. « *the (largest and debris-covered) Kekesayi Glacier* » hard to read. Rephrase without ().

L502. « > » => «above»

L510. « *but interestingly, recovered during the 2022 winter season* » why is it interesting? It slightly implies that there is a causal relationship between the summer and winter mass balance.

L546. 2019 does not seem to be an extrema for wet snow area ration in Muztag Ata in Figure 7. Why would it be the most negative mass balance year?

L561 « *departures* » => anomaly

L565 « *Of the last 16 years, summer air temperatures have experienced positive anomalies.* » To be deleted? This is expected from a series of anomalies. Maybe the number of summer with positive anomalies is missing.

L567. «*rather strong (though not significant...*» It is hard to interpret this results.

L570. « *respectively* ». Move « *scale* » before the brackets.

L571. « *likely had an impact* » => « contributed to »

L571. Is « *in turn* » necessary?

L587. Cite a study which used this method. One that comes to my mind is Nuth et al. (2013, 10.3189/2012JoG11J036) but there must be others.

L588. «*glacier-wide*» sounds like a single glacier. Maybe find another term like massif-wide? Check throughout the manuscript.

L594. Give respectively, the value for each glacier.

L607. How can two scenarios with the same dh grid but different densities result in the same mass balance for Muztag Ata?

L615. I would rephrase in: « variable density should be used for time spans of 3 years or less. » No conclusion can be drawn on longer time span periods since solely a 3 years period is studied here.

L630. Too long phrase.

L633 « *bias* » => errors. Bias often refers to systematic error.

L641 « Around our reported biases, » rephrase. Does bias means error?

L642. Delete +-, the standard deviation is a single value, not a range of uncertainty.

L644. Cite Höhle and Höhle (2009, https://dx.doi.org/10.1016/j.isprsjprs.2009.02.003) along with Dehecq et al. (2016).

L656. First paragraph of 5.2. It sounds like an introduction paragraph not a discussion one. Maybe only keep the first sentence.

L666. Too long sentence.

L674. « *as* » => that.

L686. Avoid intricating () and [], please rephrase.

L688. No undercatch of the weather station is expected ?

L691. « *have allowed us* » => allows

L700. « *too* »=> also

L711. «*Several of the stronger correlations between glacier mass balance and temperature and solid precipitation were not significant,*» Which one were significant ?

L715. What is a « *relevant development* »? Not clear.

L757.« *trend* » sounds like  there is a continuous decrease or increase of the mass balance while only two periods are compared. Replace with « period »?

L810. Provide the « period ». Idem as previous comment about the term «trend»

Figure 1. Top panels : Legend for « investigated glaciers » does not match with the figure. The bottom panels should be merged to show both study sites on the same map. Consider changing the geographical features (topographical map ? countries borders ?), the colours and texts. It is blurry and very hard to read.

Figure 3. Which map is in winter and which one is in summer?

Figure 4. f is not annual but multi-annual. I would put all annual mass balance on the upper row (move e map to c position).

Figure 5. Improve readability, change line colours, increase line width.

Figure 7. Hardly readable. Change the line style and/or the marker style.

Figure 8. Highlight which mass balances comes from this study.

Figure 9. Zadang line style is too similar to annual time step of this study.

Table 1. Consider adding a « Difference (days) » column?

Table 2. What are these values of SE 10-4? Cite Höhle and Höhle (2009) for the NMAD. Would be nice to have the same number of significant digits. SD and SE in full letter in the first line. Caption says on-glacier too but not found.

---

## Author Comment (AC1)

**Annual to seasonal glacier mass balance in High Mountain Asia derived from Pléiades stereo images: examples from the Pamir and the Tibetan Plateau (tc-2022-264)**

**By Daniel Falaschi et al.**

Reply to anonymous referee#1

**The paper deals with mass inter- and intra- annual ballance of glaciers in High Asia fociusing at two sites one in Eastern Pamirs and the second in the central Tibetan Plateau. It uses Pleiades stereo data for derivation of DSMs which are compared by differencing. A number of corections are applied prior to the differencing. Findings for the two specific sites are presented. Several auxiliary methods are utlized such as classification of snow type from Sentinel-2 data for detection of the accumulation type of gaciers or Glacier Index. The potential of Pleiades data to monitor mountain glaciers is dicussed.**
**The manuscript has high scientific quality and is recomended for publication with minor changes.**
We thank the reviewer for the positive review of the manuscript. Below we provide an inline response to each comment.

**Minor comments:**
**35: Consider using the name "Nyenchen Tanglha Mountains" instead of "Nyainqêntanglha" as it is widely used in the literature. The Chinese version has more than variants for instance "Nyainqentanglha" in Bolch et al. 2010. The English version is unambiguous and appropriate.**
We thank the reviewer for the suggestion. The name "Nyainqêntanglha" is well established in the international literature, but agree that it would be beneficial to add also the English name. Were therefore will add "Nyenchen Tanglha" in the study site section for clarification purposes.

**140: Use larger font for the coordinates around on the map frame and remove left and bottom coordinates as they are duplicates.**
Many thanks for the suggestion. We will remove duplicate coordinates and increase the font size of the grid labels. We will also thicken the glacier outlines and reduce the transparency for better visibility of the figure.

**190: What is the meaning of the "oversampled GSD"? Does this mean that the real spatial resolution is different? For how much?**
With "oversampling" we meant that Pléiades panchromatic scenes are acquired at 0.7 m pixel resolution, yet they are delivered at an (increased) ground sampling distance (GSD) of 0.5 m. Whilst we had originally explained this in the main manuscript, we will omit the word "oversampled" for simplicity.

**220: "it can be reduced to a few decimeters after DEM coregistration" This needs a citation.**
The citations for this statement are actually in the previous line. We will amend the text to clarify this.

**205: The sentence starting with "Such is the case of…" is somehow abrupt.**

We will reword this for improved reading.

**210: Table 1: add column Δt for each site to show what are the tine differences. Otherwise, the reader is forced to do an awkward calculation.**
We thank the reviewer for the suggestion. We will include an additional column in the table, showing the elapsed time interval in years between consecutive Pleiades acquisitions over each study site.

**220: You should expand "AMSAG" if it is an abbreviation.**
We will add the full name (automated snow mapping on glaciers) of the ASMAG tool.

**240: three sentences starting with "According to ERA…" are not well understandable. Please reword.**
On the grounds of excessive manuscript length, we plan to reduce the length of the manuscript by a fair amount, and hence changes can be expected. We will keep this paragraph in mind when amending the draft.

**250: What was the source of the SLA?**
The snow line altitude is derived from ASMAG itself. We will clarify this in the text.

**280, 285: This para id cryptic. Please reword it.**
On the grounds of excessive manuscript length, we plan to reduce the length of the manuscript by a fair amount, and hence changes can be expected. We will keep this paragraph in mind when amending the draft.

**295: "appreciated" to "seen"**
We will change this to "observed".

**295: In Figure 2 the (c) and (d) and (e) and (f) are not localized. Are these subsets at terminus or in the summit part of the glacier? Pleas make this clear.**
The c-f panels depict glacier details that are too small to be annotated in panel a-b. We will clarify the general location of these sites in the figure caption.

**295: Figure 2 caption: "in the in the Muztag.."**
Thanks for noting this typing error. We will correct this in the text.

**305: the sentence "According to the authors" should be merged with the previous sentence.**
We will rephrase and merge these two sentences for better reading.

**385: You can remove the sentence "Alternatively Belart et al…." as it is not needed.**
We plan to keep this sentence, as it gives the necessary background on why we have neglected ice dynamics on the firn densification approach and the mass balance calculation overall.

**425: Why not using the classification of glaciers on the Tibetan Plateau based on the accumulation regime by Maussion et al. 2014?**

**Maussion, F., Scherer, D., Mölg, T., Collier, E., Curio, J., & Finkelnburg, R. (2014). Precipitation seasonality and variability over the Tibetan Plateau as resolved by the High Asia Reanalysis. Journal of Climate, 27(5), 1910-1927.**

We thank the reviewer for the suggestion and agree that in principle, the glacier accumulation regime classification of Maussion et al (2014) would be another dataset that could provide further insight to our findings. In the first place, however, this approach is based on the High Asia Reanalysis, which covers the period 2001-2011. On one hand, Zhu et al., 2018a showed that accumulation on Muztag Ata N15 Glacier varied greatly on a yearly basis, with some years showing either summer or winter accumulation. On the other hand, Huang et al. (2022) found relevant discrepancies in accumulation regimes for some regions across High Mountain Asia (being Eastern Pamir-Muztag Ata a prime example of this mismatch) as derived from gridded reanalysis data (e.g. HARv2) on one side, and the SAR-derived glacier Index (see section 5.2; lines 668-688). With this in mind, we chose the Glacier Index of Huang et al (2022) to follow an approach which would allow us to derive an accumulation regime for our own surveyed years (2019-2022). More so, with the Glacier Index we use a third validation method fully independent from reanalysis data to validate our geodetic estimates.

**650: "in-balance", would it be better to use "balanced" instead?**

Thanks for the suggestion, we will change this to "balanced".

**References cited in this reply:**

Huang, L., Hock, R., Li, X., Bolch, T., Yang, K., Wang, N., Yao, T., Zhou, J., Dou, C., & Li, Z. (2022). Winter accumulation drives the spatial variations in glacier mass balance in High Mountain Asia. Science Bulletin, S2095927322003644. https://doi.org/10.1016/j.scib.2022.08.019

Maussion, F., Scherer, D., Mölg, T., Collier, E., Curio, J., & Finkelnburg, R. (2014). Precipitation Seasonality and Variability over the Tibetan Plateau as Resolved by the High Asia Reanalysis*. *Journal of Climate*, *27*(5), 1910–1927. https://doi.org/10.1175/JCLI-D-13-00282.1

Zhu, M., Yao, T., Yang, W., Xu, B., Wu, G., Wang, X., & Xie, Y. (2018). Reconstruction of the mass balance of Muztag Ata No. 15 glacier, eastern Pamir, and its climatic drivers. Journal of Glaciology, 64(244), 259–274. https://doi.org/10.1017/jog.2018.16

---

## Author Comment (AC2)

**Annual to seasonal glacier mass balance in High Mountain Asia derived from Pléiades stereo images: examples from the Pamir and the Tibetan Plateau (tc-2022-264)**

**By Daniel Falaschi et al.**

Reply to anonymous referee#2

**Specific comments:**

**Belart et al.,2017 states that the bulk snow density is most likely the largest contributor to uncertainty in winter geodetic mass balance. In the chapter "3.1.5 Bulk density" in the manuscript, the authors refer to other studies and assumes uncertainty values, e.g. that the density values referred to are from snow pits. It is limited data of snow density in the study regions. The authors should discuss this uncertainty more and be clearer on the consequences it might have for the results.**

Thank you for the comment. It is indeed unfortunate that no further in situ densities are available to us for the selected study sites. In these conditions, we have investigated the possible effects and in section 5.1.1 of the discussion, we elaborate on the so-called triangulation tests, which illustrate the varying mass balance values when using different material densities. We provide results for two types of tests.

A first test evaluates differences in mass balance as retrieved from the accumulated (total hydrological year) vs. the sum of individual (seasonal) survey periods in the 2022 hydrological year. We illustrate the variability in mass balance residuals on both glacier-wide and glacier specific scale. This test provides confirmation of the findings of Pelto et al. (2019), in that the distribution of glacier surface classes is actually more important that assumed densities.

In the second triangulation test, we elaborate on the influence of varying material density further, and evaluate the accumulated 2020-2022 mass budget vs the sum of all the individual annual periods in two possible scenarios following Huss, 2013: a) a density of 850 ±60 kgm$^{-3}$ and b) a 3-year weighted density (Muztag Ata = 774 ±60 kg m$^{-3}$). Our results agree in fact with those of Huss et al. (2013), and confirm that using period-weighted densities reduces the differences between the accumulated vs. added mass budgets.

In addition to the above, we will expand the discussion taking into account the findings of Pelto et al. (2019) regarding how the variability of field surveyed densities in comparison with assumed densities can affect glacier mass balance.

**The use of the surface classification of snow, ice and firn from Sentinel-2 can be clarified in chapter "3.1.2 Classification of snow and ice using Sentinel-2 scenes". The chapter gives a good description of how the analysis is done, but it can be elaborated in the start of the chapter what use these data has for the geodetic glacier mass balance and why it is important.**

We will add a brief sentence about the use of the surface classification maps in the geodetic method, so as not to repeat methodological details later found in section 3.1.2. we will expand on the relevance and consequences of the snow and ice distribution as requested.

**Some of the sentences and text includes to many parentheses with additional information or clarifications. A suggestion is to go through the whole manuscript text in general and write shorter sentences that are clearer and easier to read. Here is an example to illustrate: L: 595: "Overall, we find the glacier-wide and (for the most part) individual differences to be well within the uncertainty ranges, and attribute the differences to the overall small differences in average density (which in turn derives from the snow and ice distribution) of the September 2021 (590 kg m-3) and April 2022**

**Sentinel-2 (630 kg m-3) snow and ice masks.".** The parentheses are sometimes randomly placed, e.g., "(630 kg m-3)" should maybe be placed after "April 2022"?

We appreciate the reviewer's concern for overall clarity and reading fluidity of the manuscript. As suggested, we will go throughout the manuscript and make a conscious effort to remove unnecessary or redundant parenthesis and brackets, and split long sentences into shorter ones for improved reading. Among the amends, we will include the specific lines indicated by the reviewer above.

**The authors give a good overview of how they used the Glacier Index of Huang et al. (2022), to find glacier accumulation regimes. However, it is no error estimations of the retrieval of firn and wet snow areas from the remote sensing data, and this should be elaborated.**

We thank the reviewer for this suggestion. We will include a new paragraph on the error assessment and correction. Citing the supplementary material of Huang et al. (2022), *"…the firn area ratio can be overestimated for two surface types: (1) debris covered by very thin snow which may be taken as clean ice/snow on optical satellite image, but the snow can be penetrated by SAR; (2) ice crevasses, which may form a corner reflector effect to the SAR satellite. Both cases will cause high SAR backscatter coefficients regardless of the season, and lead to misclassification as firn instead of debris and ice. To account for these effects, we assume that the pixels higher than -6 dB on both winter and late summer SAR images are misclassified as firn, and the pixels that are lower than -14 dB on both winter and summer SAR images are misclassified as wet snow pixels, and correct these the surface type accordingly."* We will rewrite this information in the manuscript but still refer to the supplementary material for full details.

**Technical comments:**

**L: 91: "The major aims of this paper are therefore to investigate the potential and limitations of geodetic mass balance estimates derived from VHR Pleiades satellite data (using 5 DEMs over the 3-year period 2020-2022)." Should it be data between 2019 and 2022? (Ref. table 1).**

Whilst we reckon that our earliest dataset stem from 2019, in the study we actually assess the geodetic mass balance of three hydrological years (2020, 2021 and 2022). The hydrological year 2020 starts in 2019, so this issue is a bit tricky. We will change to geodetic mass balance between 2019 and 2022, but will clarify the issue with the hydrological year in the text.

**L:425-429: "To account for different glacier areas between the study sites, we express firn and wet snow areas on each region as a fraction of the total glacier area (hereafter referred to as firn area ratio and wet snow area ratio). This ratio can vary to a great extent across different geographic regions through time, whilst interannual variations of the firn area ratio remain relatively small." Which ratio do the authors refer to when compared to the firn area ratio? Glacier index, I? A suggestion to rewrite sentences.**

Many thanks for the observation. Here we simply compare the interannual variability of the wet snow area ratio and the firn area ratio. We will amend this for clarification.

**L: 440: "First, the Landsat scenes are used to recognize debris-covered and debris-free areas (ice and snow) on glaciers surface applying a threshold to the previously computed Normalized Difference Snow Index [NDSI] (Bruns et al., 2014).". The authors describe the use of Landsat-data for glacier higher spatial resolution?**

This is a valid question. In principle, Sentinel-2 scenes would also be a good choice to identify debris-covered and debris-free areas on glaciers. On Google Earth Engine, the highest resolution of the Sentinel-1 SAR images is 10 m. However, in the methodology developed by Huang et al. (2022), the spatial resolution of the Sentinel-1 scenes is resampled to 30 m to reduce the speckle effect on SAR images, which may affect the firn and snow identification. The method thus uses 30 m resolution Landsat images, matching the resampled SAR images, to identify and remove debris.

**L: 830: "The ever increasing and availability of very-high resolution optical satellites (with stereo capability and relatively short revisit time) will allow for increasing the number of glaciers in isolated regions that can be readily monitored.". Can the authors clarify which satellite sensors they are referring to in the last sentence of the conclusion? It is not planned many optical missions with stereo capability in the future. Consider to be more specific and give examples of missions you refer to.**

Good point. We will add a brief list of VHR satellites with stereo capability (Pleiades, WorldView 1-2, SPOT6-7) that are currently operating, whilst we also listed ALOS-3, which was recently launched. We will also remove the "ever increasing" statement in the text.

**L: 1015: Wrong year in Huang et al., 2022 in reference list.**

Many thanks for noting this typo. We will amend this accordingly.

**Figure 1: It is not clear to me which glaciers are "investigated glacier" in the figures. Is it all of them? Consider changing color or outline and rewrite to "investigated glaciers".**

We will thicken the glacier outlines and reduce their transparency for a better visualization. We will also eliminate duplicated coordinates and increase the size of the grid labels in this figure.

**Figure 4 and 6: Cannot really see the dh variation in the figures. A suggestion is to make the figures larger, and subsets of the individual glaciers discussed in the text can also be included.**
**Figure 8: Improve the representation and better the resolution of the plots.**
**Figure 9: It is hard to see the difference between the lines indication "this study" in the plots. Consider changing color on either "annual time step" or the individual glaciers.**

Many thanks for the suggestion. We will modify all figures by using thicker lines, different line types to separate our results from previous studies, a color-blind friendly palette, increased font size and larger scales when possible. In Figures 4 and 6 we will incorporate hillshade images as background on all panels, to better illustrate the relation between the glacier elevation changes and topography.

**References cited in this reply:**

Huang, L., Hock, R., Li, X., Bolch, T., Yang, K., Wang, N., Yao, T., Zhou, J., Dou, C., & Li, Z. (2022). Winter accumulation drives the spatial variations in glacier mass balance in High Mountain Asia. Science Bulletin, S2095927322003644. https://doi.org/10.1016/j.scib.2022.08.019

Huss, M., Sold, L., Hoelzle, M., Stokvis, M., Salzmann, N., Farinotti, D., & Zemp, M. (2013). Towards remote monitoring of sub-seasonal glacier mass balance. Annals of Glaciology, 54(63), 75–83. https://doi.org/10.3189/2013AoG63A427

Pelto, B. M., Menounos, B., & Marshall, S. J. (2019). Multi-year evaluation of airborne geodetic surveys to estimate seasonal mass balance, Columbia and Rocky Mountains, Canada. *The Cryosphere*, *13*(6), 1709–1727. https://doi.org/10.5194/tc-13-1709-2019

---

## Author Comment (AC3)

**Annual to seasonal glacier mass balance in High Mountain Asia derived from Pléiades stereo images: examples from the Pamir and the Tibetan Plateau**

**Falaschi et al. TC Discussion**
**Review by César Deschamps-Berger**

**This article presents a time series of three annual and two seasonal mass balances for two glacerized massifs in High Mountain Asia. In line with previous studies, it is found that the Pamir glaciers have a mass balance close to equilibrium between 2019 and 2022 while the glaciers from the Tibetan Plateau have a negative mass balance. Various satellite products (Pléiades, Sentinel 1 and 2) were combined to identify the surface elevation change and the accumulation and ablation areas. The snow/firn density and the firn densification are taken into account based on field measurements. The authors put the measured mass balance in longer time scale context and improve the description of the glacier accumulation regimes. It is a detailed work which combines advanced methodologies with good knowledge of the region. I think this article will be a valuable contribution after the following main concerns are addressed.**

Thank you for the overall positive assessment of the manuscript and for the critical but supportive comments.

**1. The winter elevation changes show disturbing patterns (Figure 4, 6 and blue lines in Figure 5). In Western Nyainqêntanglha, elevation loss in winter are stronger at the highest elevation. On the contrary, elevation loss in winter are measured in the Muztag Ata between 6000 m asl and 7000 m asl with areas of elevation gain below and above. Are these elevation change significant or within the calculated uncertainties? What process could explain such altitudinal distribution of elevation change? I wonder if it could be related to remaining errors in the elevation change map (jitter correction, gap filling, shaded areas, see areas highlighted below). Jitter correction might not be perfect due to the lack of stable terrain, especially for Muztag Ata. Better highlighting and explaining these errors might impact the conclusions on the accumulation regimes (e.g. L496, 5.2) and on the sources of uncertainty of the mass balance estimation (e.g. L595-601 and 5.1.2). Left: Figure 4. c. Muztag Ata; Right: Figure 6.c. Western Nyainqêntanglha,**
We are grateful to the reviewer for dwelling in the detailed analysis of the elevation change grids. In response we have carefully inspected all elevation change tiles and actually detected more problematic maps than those in the review Figure. To our knowledge, the effect of satellite (Pléiades, ASTER) jitter varies on a region- and year-specific basis. We are in possession of Pléiades data acquired over other mountain regions, which do not show such strong jitter effect compared to the dataset used in the present study. We agree that limited stable terrain might be responsible for a non perfect jitter correction (and vertical coregistration, for that matter). However, upon examining our grids, we nevertheless came to the conclusion that the artefacts in our elevation change grids were not necessarily the consequence of remaining errors in the jitter correction, but probably more related to the void filling procedure. We are currently exploring different possibilities of void filling on all the problematic grids.

In Muztag Ata, we inspected and identified the elevation bins that showed artefacts (paying particular attention to the bins showing very high elevation gains). We will amend the dh grids by removing cells exceeding the mean value ±3σ. This basically means applying the method of GArdelle et al. (2013) for the problematic bins.

In the case of the Western Nyainqentnaglha grids, we examined the problematic winter 2022 elevation change grid in detail and noticed that at the uppermost elevation, there are negative elevation change values of local extent that are most probably related to crevasses and maybe wind-blown snow to a lesser degree. These are valid cells in the "raw" elevation change grids (with no corrections applied) and in our opinion they should not be treated as outliers. They do not look like random or high slope-related artefacts, nor do the Pleiades scenes look especially saturated in those areas. In addition, these negative change areas are not placed in shadowed areas. The problem in the void-filled grids arises as there are relatively few valid cells at similar elevations (these are the highermost reaches of the entire

Western Nyainquetanglha study area). The glacier-wide void-filling polynomial fitting thus gives too much weight to these negative values, and fills in the large voids in the saturated areas with rather negative values. This gives the wrong impression that all of the upper part of the glaciers has gone through significant thinning at high elevation during that period. To solve this, we will remove the few highly negative elevation change values in the uppermost reaches and adjust a new function. In doing so, we will resample the elevation bins to 25 m so that sufficient bins are available for a representative fit.

Whilst we acknowledge that some of the mass balance estimates might change (final calculations need to be redone at this point), we doubt that this will have an overly large impact on them, and will most probably not alter any conclusion on the proposed accumulation regimes.

**2. I estimate that this manuscript would result in an article of more than 20 pages. The article readability would benefit from being more concise. I would advise the authors to revise the manuscript with that in mind. Some of my minors comment should help to gain space (e.g. L89, L108, L125, L194, L283...). If a radical choice was to be made, moving to supplement or removing the parts about the correlation of mass balance with climatological variables might not alter the value of the article. It is almost a distinct topic from the core of the article (see title). The data used partly come from other articles and finally, few (or no) significant correlation are found.**

We appreciate the reviewer's concern for the overall readability of the manuscript. We will do a conscious effort to reduce the paper length by keeping in mind the suggestions made by the reviewer, and will put an additional effort to shorten each section wherever possible, writing in a more concise way. We think that this approach is preferable over moving entire sections of the manuscript regarding climate analyses to the supplementary material.

**3. The quality of the figures should be improved. Almost all the plots with lines are hard to read due to the style and colour of the lines. For instance, it is very hard to distinguish several lines of Figure 5, the individual glaciers in Figure 7, Mass balance from Solid precipitation in Figure 8. Select better colour and line style.**

Based on these comments and the critical review by the other reviewers, we will introduce changes to all figures to improve visibility. Specifically, in Fig 5 we will increase the font size and increase the line thickness. We will incorporate color-blind friendly palettes for plots. In Fig 7 we will increase the font size and line thickness. In Fig 8 we will change the color from solid precipitation and separate the results from this study against previous ones. In Fig 9 we will modify the figure by changing the color of the annual time step lines for an improved visualization. Please see the specific replies to Figure suggestions toward the end of this document. Since we do not know the final sizes of the figures at this stage, we are happy to introduce further modifications to the figures in the production stage shall this manuscript be accepted for publication.

**Minor comments and suggestions**
**L26. Pléiades. Throughout the text.**
We will correct this accordingly.

**L31. delete « previously observed »**
If this clarification is removed, then it would seem that the mass balance records from the last 6 decades was generated in this study, which is clearly not. We chose to keep this in the text.

**L32-33. delete « *on average* ». « *mean*» is already stated at the beginning of the sentence.**
We will correct this accordingly.

**L33. « *increased* » compared to what?**
Good point. We will add "to the previous ~6 decades" (which is the period covered in Battacharya et al., 2021))

**L33. Why is Western Nyainqêntanglha qualified here as summer accumulation type when summer mass balance (-0.66 m w.e.) is more negative than the winter one (-0.04 m w.e.)? Besides,**

**this conclusion (Western Nyainqêntanglha being summer accumulation type) seems based on other studies in the dedicated Discussion paragraph (L696-698). «** *The 2022 winter (+0.21 ±0.24 m w.e.) and summer (-0.31 ±0.15 m w.e.) mass budgets in Muztag Ata and Western Nyainqêntanglha (-0.04 ±0.27 m w.e. [winter]; -0.66 ±0.07 m w.e. [summer]) suggest winter and summer accumulation-type regimes, respectively.* **» I suggest rephrasing as: «** **The seasonal mass balance in Muztag Ata (winter: XX m w.e., summer : XX m w.e.) and Western Nyainqêntanglha (winter: XX m w.e., summer : XX m w.e.) suggest... ».**

We acknowledge that W. Nyainqentanglha is actually located in a transition area between the monsoon-dominated glaciers in the SE Tibetan Plateau and Himalaya and the westerlies-dominated glaciers to the Northwest (see e.g. Bolch et al. 2010, TC or the recent paper by Zhu et al., 2023 in GPC). Whilst we reckon that the winter mass balance estimate (-0.04 m w.e.) in W. Nyainqentnglha must yet be corrected (due to the artefacts in the elevation change maps), we did not observe elevation gains in the uppermost reaches of the glaciers. Yet, the summer elevation change panel in Figure 6 (see also Figure 5) does indeed show elevation change that can be attributed to accumulation, which is not the case in the winter panel. Zhang et al (2013) clearly show that in W Nyainqentnglha, accumulation trough precipitation is higher in summer than in winter, but contemporary mass losses though runoff and evaporation can exceed mass gains and lead to a strongly negative mass balance. We interpret this as the reasoning for the observed mass balance in our study. We will clarify the information about the accumulation-type in the revised manuscript.

**L80. Not only WorldView-2 (see Shean et al., 2020). Simply put « WorldView ».**
Changed

**L86. Deschamps-Berger.:)**
Many thanks for noting; we will correct this accordingly.

**L89. «***(e.g. Ice, Cloud and land Elevation Satellite-2 -ICESat-2)***» ICESat-2 is not further used. Give only the acronym.**
We will change this accordingly

**L92. Quit the brackets.**
In general, and following the suggestion by another reviewer, we will go throughout the manuscript and will remove redundant and unnecessary brackets for an improved readability.

**L96. «***displayed dissimilar mass change rates.***» precise over which epoch, otherwise it sounds like an article's result is given away.**
We will add the six decade period

**L97. «** *mass balance for longer period* **» Longer than what?**
We will rephrase this part of the text for clarity.

**L104. In 2.1., provide the max elevation as in 2.2. Tell if the whole massif is covered.**
We will rephrase this to incorporate the maximum elevation of Muztag Ata Mountain.

**L108. Delete «***(along with the nearby Kongur Shan mountains).***» It is never mentioned again.**
We will remove this accordingly

**L124. «***glacier ice***»**
We are unsure about what this particular correction is, Can you please clarify?

**L125. «***Glaciers in the arid NW Tibetan Plateau are predominantly continental-type, cold–based (with their basal part entirely below the pressure melting point) and receive little precipitation.***» Might be deleted? All these informations are repeated in the next lines.**
Good point. We will delete this accordingly

**L131-134. Give periods for each mass balance epoch, confusing otherwise.**
We will add the year 2009 and 2019 as the start and end (according to the literature) of the slight mass loss period in Muztag Ata.

**L147. « *~230 km in length reaches* » => « extend over ~230 km in length and reaches... »**
We will correct this accordingly.

**L163. « *in their model* »? An energy balance model is not a source of information about precipitation.**
We will correct this accordingly.

**L165. « > » => « more than ». Delete « here ».**
We will correct this accordingly.

**L168. Why use []? And why « accelerated »? compared to what?**
We will rephrase the sentence and include time intervals to clarify the accelerating rate of glacier mass loss.

**L170. Keep giving MB with two digits precision « (–1.0X m) ».**
We thank the reviewer for the suggestion. We will amend the manuscript in this regard based on the provided citations.

**L172. « *Zhadang glacier*» first time it is mentioned. Introduce shortly the glaciers of interest (size, specificities).**
Many thanks for the suggestion. We will introduce Zhadang Glacier and includ it in Figure 1.

**L192. Delete «*relatively*».**
We will delete this accordingly

**194. « *separated in time on most occasions (Table 1). In all cases, partial acquisition dates were no more than 2 weeks apart.* » => « separated by two weeks in the worst case (Table 1).»**
We will correct this accordingly.

**L195. Was the DEM produced in one run from the raw stereo images to a high-resolution DEM? A common practice to reduce errors is to first project the images on a low-resolution DEM (idea for future work).**
Yes, the DEMs were generated in one run from the raw stereo images as described. Whilst generating the DEMs using the suggested approach again might be an excessive amount of work for this study, we thank the reviewer very much for the methodological tip for future work!

**L196. « *implementing* » => « using »**
We will correct this accordingly.

**L197. Why «although»? The first part of the sentence refers to a method of this work, the second refers to results from other studies. Grammatically hard to understand.**
This is correct. We will remove the word although.

**L200. « *Pléiades DEMs are currently amongst the most common very high resolution DEMs used in geodetic mass balance assessments*». Maybe not necessary as there are anyway few high resolution photogrammetric satellite and studies exists with WorldView (Shean et al., 2020).**
We will remove this accordingly

**L203. « *beyond the higher spatial resolution,* » not true for WorldView satellites, could be deleted.**
We will delete this accordingly

**L205.** *«saturated areas» => « areas prone to saturation »*
We will correct this accordingly.

**L209. Did you request that the images were acquired with reduced Time Domain Integration (TDI)? It can help preventing saturation (Deschamps-Berger et al., 2020).**
No, we did not require this option to be active during our acquisitions, but many thanks for the suggestion. We have now two sources that recommend this specific setting for the next acquisitions. Upon reviewing the AIRBUS image acquisition request form, however, we noticed that this is not explicitly shown as an available option. We understand that only by adding it as an additional comment may work.

**L223. I understand that the lower resolution saves computational time but less the number of images. Is the opening and closing of the images really a bottleneck in the treatment?**
This is an interesting question. We used the ASMAG algorithm, whose current version can incorporate Landsat and Sentinel-2 scenes. It has not been adapted for Pleiades images yet. We will clarify this in the manuscript. Moreover, we had to modify the existing code so that it could open and read Sentinel scenes from the two sources we used (EarthExplorer and the Copernicus Hub). Since the Sentinel image files are structured differently depending on the source, this required independent processing for each image source.

**L232. delete « *see also* »**
We will delete this accordingly

**L235.** *«To this ends, it implements an automatic threshold to the near-Infrared»* **=> « It determines automatically a threshold for the NIR band values ».**
We will correct this accordingly.

**L241. Move the « Muñoz » citation after « daily data ».**
We will change this accordingly.

**L257. Delete « see e.g. ».**
We will delete this accordingly.

**L260. Move the UTM info somewhere else more generic.**
We will delete this part of the sentence, since it is of little relevance really.

**L265. In future work, you might want to first co-register your reference DEM to an external reference (e.g. Copernicus DEM) to ensure a better absolute co-registration.**
Good point. Many thanks for the tip for future work!

**L269.** *«reprocessed 2019 Pléiades DEM from Bhattacharya et al. (2021)»* **is confusing. Was the 2019 DEM eventually calculated like the others of this study? Then, the Bhattacharya reference could be deleted. Maybe, the link between this study and Bhattacharya et al. (2021) should be better explained in introduction. At least better introduce the Bhattacharya et al. (2021) study as some products are used here (L 472).**
This is a valid question and the writing was indeed confusing. The 2019 DEM from Bhattacharya et al were originally processed using commercial software. Our 2019 DEMs (to which all later DEMs were coregistered) were newly generated using the Ames Stereo Pipeline as written in the manuscript. We will reword this paragraph to avoid any confusion.

**L270. Which metric is used to correct the vertical biases? The mean, the median?**
As per Nuth and Kääb (2011), the metric used to correct biases is the mean value on stable terrain. During the preparation of the study, we also tried DEM coregistration using the algorithm of Berthier et al. (2007, RSE), which we run using the median value of the elevation changes on stable terrain. This provided coregistration results of lower quality compared to Nuth and Kääb (2011).

**L271. Delete « *(reference)* »?**
We will delete this accordingly

**L275. «*implemented*» to be clarified. It sounds like you implemented the code (i.e. wrote the code). However in the acknowledgement the tools of Etienne Berthier are mentioned. Also note that it is a different method than the one used in Deschamps-Berger et al. (2020). The first one fits polynomial functions to the residual while the second calculates and modifies the Fourrier transform spectrum of the residual. Make sure which one was used or implemented.**
We greatly appreciate the reviewer's insight into these methodological differences. Indeed, we used the tools of Etienne Berthier, which f, it a spline function for correcting the systematic bias, and included the reference of Falaschi et al. (2023).

**L277. « *on- and* » missing blank**
We will correct this accordingly.

**L283. « *We chose a 3-cell buffer so as not to remove valid cells from the original dDEM grids.* » This kind of sentence could be deleted to make the article more concise.**
We will remove this accordingly

**L284. « *mosaiced*»? merged? How are managed areas where there is overlap between tiles (concisely)?**
From our understanding, the correct term should be *mosaiced* as these are raster files and this is the GIS operation that we have used. For the overlapping areas, we opted to use the average value of each grid elevation change. Void filling and firn densification correction were carried out on each elevation change individually before final mosaicking. Finally, since we applied a seasonality correction to adjust each grid to the hydrological year, using the mean elevation change on overlapping areas seems a technically sounding approach. We consider that any uncertainty owed to this should be within the overall elevation change uncertainty. We will include these clarifications in the text.

**L305. « *in the energy balance model comprising Muztag Ata N15 and Zhadang glaciers (in Muztag Ata and Western Nyainqêntanglha districts, respectively) by Zhu et al. (2018a). According to the authors, this density value was retrieved from snow pits.*» Maybe no need to repeat the districts if the glaciers are introduced before. Why mention the energy balance model if the density actually come from pits measurements? Merge sentences concisely.**
We will remove the district reference. We will amend this part of the text to avoid mentioning energy mass balance, but still pointing at the in-situ surveys mentioned in Zhu et al (2018a).

**L318. « time interval » dt might be more clear than *t*?**
We will change  to dt

**L321. Eqn (2) « k » is missing. Replace «i»?**
Thanks for noting. We will discard k and keep i in the equation instead.

**L326. It is not clear what the « i » of Ai refers to? Ice? The previous sentence mentions ice and snow areas.**
We will remove the "i" in Eqn3. By using the i, we originally intended to make it clear that we considered the accumulation and ablation areas independently for the mass balance calculation.

**L330. I cannot find easily in this paragraph if the correction is applied on a pixel scale or at the firn area scale? Please clarify.**
Good question. We applied the correction on a pixel scale and included this information in the text.

**L346-347 () [] to homogenise**
We will correct this accordingly.

**L346.** *«most recent elevation grids available »* => « **most recent elevation change grids »? Since period are provided in brackets (2013-2019, 2018-2019).**
Yes, we will correct this accordingly..

**L374. «** *is not to be expected* **» => « is not expected ».**
We will correct this accordingly.

**L395. what does «** *addressed* **» mean? Calculated, defined?**
We will change this to defined

**L397. By construction, the mean elevation difference over stable terrain is zero or close to it. Depending whether the mean or median elevation residual was used for vertical coregistration (to be added in 3.1.4). This underestimate potential systematic error. Besides, I do not find sigma_sys further in the uncertainty calculations.**
In our study we have assessed both systematic and random uncertainties. Systematic and random uncertainties need to be determined separately (see the original reference of Koblet et al, 2010) in the manuscript. Our systematic uncertainty estimates are displayed in Table 2, whilst the corresponding calculation is given in Eqn 5. As per the Nuth and Kääb (2011), the mean elevation residual on stable terrain is used to perform vertical coregistration. It is our opinion that the relatively small stable terrain areas around our glaciers causes a systematic uncertainty in some of the elevation change grids. In Table 3, the ±values refer to the random uncertainty, which are calculated from Eqn 7.

**L410. Delete «** *see also* **»**
We will delete this accordingly

**L415. Eqn 7. What is « f »?**
Thanks for noting the absence of the definition for $f_{\Delta v}$. $f_{\Delta v}$. **is** the volume to mass conversion factor of ice and snow. We will add this to the text.

**L424. Huang et al. (2022) missing in the bibliography.**
The citation is correct. We simply had written the wrong year in the reference list. We will correct this accordingly.

**L433. The index**
We will correct this accordingly.

**L433. Please repeat that firn area is measured at the end of the winter and the wet snow area at the end of the summer. Or clarify this point if I misunderstood.**
We will add this accordingly.

L449. « the influence of XX to YY » Is this grammatically correct? Otherwise replace *«to govern»* by «on».
We will correct this accordingly.

**L452. Put the citations at the end of the sentence. Why mentioning the period of availability of the data? Idem L463.**
We will move the citation and remove the reference to the data availability

**L456. Delete «** *either* **»?**
We will delete this accordingly

**L458. Cite APHRODITE along with ERA5, HARv2 in the previous sentence and delete « (APHRODITE, ERA5, HARv2) » in this one.**

We will add APHRODITE along with ERA5, HARv2 in the first sentence and removed all the datasets in the second one.

**L459. «*(or instrumental records»* to delete**
We will delete this accordingly

**L461. «*Muñoz-Sabater*»**
We will correct this accordingly.

**L463. «** *to monthly time step* **» Was ERA5 initially at a daily or monthly time step? Mention it in the previous sentence.**
This information is stated in line 455 of the preprint.

**L468. «***found***» => « calculated »?**
We will correct this accordingly.

**L468. Redundancy with « mean » and « average » in the same sentence.**
We will correct this accordingly.

**L472. «** *added the geodetic mass balance values in Bhattacharya et al. (2021)* **» see comment about L269. I understand that data from Bhattacharya et al. (2021) are not at a yearly or seasonal resolution. Do you think that mixing periods of different durations could have an impact on the correlation calculated?**
Mixing periods of different durations might certainly affect the correlations. However, we think that the small number of observations overall available (considering both Bhattacharya et al (2021) and our study) is the main limiting factor that does not allow to obtain correlation coefficient values that are statistically significative. We will comment on this in the manuscript.

**L481. Cite « Table 3 » only once in this paragraph.**
We will correct this accordingly.

**L483-484. Please provide value in brackets for each year. Or alternatively do not provide any.**
We will add the missing values in brackets.

**L486. «** *the (largest and debris-covered) Kekesayi Glacier* **» hard to read. Rephrase without ().**
We will remove this, as is it was anyway introduced in the study area section

**L502. « > » => «above»**
We will change this accordingly

**L510. «** *but interestingly, recovered during the 2022 winter season* **» why is it interesting? It slightly implies that there is a causal relationship between the summer and winter mass balance.**
The reviewer is correct. We therefore removed this statement.

**L546. 2019 does not seem to be an extrema for wet snow area ration in Muztag Ata in Figure 7. Why would it be the most negative mass balance year?**
We did not actually produce geodetic results for the hydrological year 2019. Whilst we reckon that our earliest dataset stem from 2019, in the study we actually assess the geodetic mass balance of three hydrological years (2020, 2021 and 2022). The hydrological year 2020 starts in 2019, so this issue is a bit tricky. We will clarify the issue with the hydrological year in the text.
It is actually the hydrological year 2020 which has the most negative mass balance (-0.19 m we), and has also the smallest wet snow area ratio except for the hydrological year 2022. Both years have rather similar Glacier Index values, so we do not necessarily see a contradiction between the Index and the mass balance data.

**L561 «** *departures* **» => anomaly**
To our knowledge, it is perfectly acceptable the use of departures as a surrogate for anomalies. We originally tried to use both terms as not to constantly repeat the same word, but in the end decided to use anomaly in the manuscript only.

**L565 «** *Of the last 16 years, summer air temperatures have experienced positive anomalies.* **» To be deleted? This is expected from a series of anomalies. Maybe the number of summer with positive anomalies is missing.**
Yes! Many thanks for noting. We were missing the number of years with positive anomalies. We will correct this accordingly..

**L567. «***rather strong (though not significant...***» It is hard to interpret this results.**
We will delete "rather".

**L570. «** *respectively* **». Move «** *scale* **» before the brackets.**
We will move this accordingly

**L571. «** *likely had an impact* **» => « contributed to »**
We will correct this accordingly.

**L571. Is «** *in turn* **» necessary?**
We will delete this accordingly
**L587. Cite a study which used this method. One that comes to my mind is Nuth et al. (2013,10.3189/2012JoG11J036) but there must be others.**
Many thanks for the reference. We will cite it as suggested.

**L588. «glacier-wide» sounds like a single glacier. Maybe find another term like massif-wide? Check throughout the manuscript.**
To our knowledge, the term "glacier-wide" is fully accepted in the literature to refer to all glaciers or all glacier ice area. See for example McNabb et al (2019), a study that we reference in the context of the usage of the term glacier wide.

**L594. Give respectively, the value for each glacier.**
Do you mean the triangulation residuals for each glacier? It would seem to us an overkill (and of little added value) to provide the residuals for 117 glaciers in W. Nyainqentanglha and 86 glaciers in Muztag Ata, even in a supplementary Table. We have provided the residuals for the full glacier area and illustrate the variability with other 4 glaciers.

**L607. How can two scenarios with the same dh grid but different densities result in the same mass balance for Muztag Ata?**
This was a simple typo since values were very similar but still different. We will correct this accordingly.

**L615. I would rephrase in: « variable density should be used for time spans of 3 years or less. » No conclusion can be drawn on longer time span periods since solely a 3 years period is studied here.**
Agreed, we will correct this accordingly.

**L630. Too long phrase.**
We will split the sentence in two shorter ones.

**L633 «** *bias* **» => errors. Bias often refers to systematic error.**
We will correct this accordingly.

**L641 « Around our reported biases, » rephrase. Does bias means error?**

Yes, we will change to errors.

**L642. Delete +-, the standard deviation is a single value, not a range of uncertainty.**
That's correct, we will delete this accordingly

**L644. Cite Höhle and Höhle (2009, https://dx.doi.org/10.1016/j.isprsjprs.2009.02.003) along with Dehecq et al. (2016).**
Many thanks for the reference, we will add it in the manuscript.

**L656. First paragraph of 5.2. It sounds like an introduction paragraph not a discussion one. Maybe only keep the first sentence.**
This is a good point, and a simple mean to shorten the manuscript as pointed out earlier in the review. We will keep the first sentence of the paragraph only.

**L666. Too long sentence.**
We will rephrase and shorten the sentence.

**L674. « *as* » => that.**
This actually needs to be changed to "at"

**L686. Avoid intricating () and [], please rephrase.**
We have gone throughout the manuscript and will make a conscious effort to remove unnecessary or redundant parenthesis and brackets, and split long sentences into shorter ones for better reading. Among these amends are the specific lines indicated by the reviewer above.

**L688. No undercatch of the weather station is expected ?**
There is a possibility of undercatch like for many other weather stations. The very low precipitation is however, consistent with gridded climate data and also the vegetation in the region. Even if the precipitation would be twice as much there would still be a huge difference to the precipitation at the top of the mountain.

**L691. « *have allowed us* » => allows**
We will change this to allowed

**L700. « *too* »=> also**
We will correct this accordingly.

**L711. «*Several of the stronger correlations between glacier mass balance and temperature and solid precipitation were not significant,*» Which one were significant ?**
Actually, only the correlation between annual temperature anomalies in W. Nyainqentanglha and mass balance was significant at the 95% confidence interval. We will correct this accordingly.

**L715. What is a « *relevant development* »? Not clear.**
We will rephrase for clarity.

**L757.« *trend* » sounds like there is a continuous decrease or increase of the mass balance while only two periods are compared. Replace with « period »?**
Good observation, we will change to period.

**L810. Provide the « period ». Idem as previous comment about the term «trend»**
Not fully clear what is meant here. The periods are included in the manuscript: (-0.50 ±0.17 m w.e. a-1 between 2000 and 2014, Li and Lin., 2017; -0.60 ±0.19 m w.e. a-1 between 2000 and 2017, Ren et al., 2020).

**Figure 1. Top panels : Legend for « investigated glaciers » does not match with the figure. The bottom panels should be merged to show both study sites on the same map. Consider changing the geographical features (topographical map ? countries borders ?), the colours and texts. It is blurry and very hard to read.**

We will modify the figure, correcting the legend for investigated glaciers. We have will also merge both location maps into a single one to better illustrate the location of Muztag Ata and W. Nyainqentanglha in High Mountain Asia. The figure inset will feature a topographic map. We will also add the highest elevations on each study site. We have chosen not to include country borders based on the fact that this is a region where a number of border conflicts exist between several countries. We will still provide an overview of the relative position of the countries.

**Figure 3. Which map is in winter and which one is in summer?**

We will add winter and summer labels to the figure

**Figure 4. f is not annual but multi-annual. I would put all annual mass balance on the upper row (move e map to c position).**

Many thanks for the suggestion. We will move the panels as indicated.

**Figure 5. Improve readability, change line colours, increase line width.**

We will increase the font size and line thickness.

**Figure 7. Hardly readable. Change the line style and/or the marker style.**

We will modify the figure to make the plots larger in size, increase font size and line thickness, and use different line types for improved visibility.

**Figure 8. Highlight which mass balances comes from this study.**

We will distinguish the mass balance and solid precipitation/temperature anomalies stemming from Bhattacharya et al. (2021) from our own results in the figure using different line styles and colors.

**Figure 9. Zhadang line style is too similar to annual time step of this study.**

Many thanks for the suggestion. We will revamp the figure, using a color-blind friendly palette, and separating our results from previous studies using different line styles We will also increase line thickness and font size.

**Table 1. Consider adding a « Difference (days) » column?**

This was suggested by another reviewer too. We will add a column with the time interval (in years) between consecutive Pleiades acquisitions for each portion of the study sites covered by the Pleiades scenes.

**Table 2. What are these values of SE 10-4? Cite Höhle and Höhle (2009) for the NMAD. Would be nice to have the same number of significant digits. SD and SE in full letter in the first line. Caption says on-glacier too but not found.**

We will check the SE values. As suggested in one of the comments, we will cite Höhle and Höhle (2009) in line 650 regarding NMAD, and will write standard deviation and standard error in full letters. We will correct the figure caption, as "on-glacier" values were not included.

**References cited in this reply:**

Berthier, E., Arnaud, Y., Kumar, R., Ahmad, S., Wagnon, P., & Chevallier, P. (2007). Remote sensing estimates of glacier mass balances in the Himachal Pradesh (Western Himalaya, India). Remote Sensing of Environment, 108(3), 327–338. https://doi.org/10.1016/j.rse.2006.11.017

Bhattacharya, A., Bolch, T., Mukherjee, K., King, O., Menounos, B., Kapitsa, V., Neckel, N., Yang, W., & Yao, T. (2021). High Mountain Asian glacier response to climate revealed by multi-temporal

satellite observations since the 1960s. Nature Communications, 12(1), 4133. https://doi.org/10.1038/s41467-021-24180-y

Bolch, T., Yao, T., Kang, S., Buchroithner, M. F., Scherer, D., Maussion, F., Huintjes, E., & Schneider, C. (2010). A glacier inventory for the western Nyainqentanglha Range and the Nam Co Basin, Tibet, and glacier changes 1976–2009. The Cryosphere, 4(3), 419–433. https://doi.org/10.5194/tc-4-419-2010

Falaschi, D., Berthier, E., Belart, J. M. C., Bravo, C., Castro, M., Durand, M., & Villalba, R. (2023). Increased mass loss of glaciers in Volcán Domuyo (Argentinian Andes) between 1962 and 2020, revealed by aerial photos and satellite stereo imagery. Journal of Glaciology, 69(273), 40–56. https://doi.org/10.1017/jog.2022.43

Höhle, J., & Höhle, M. (2009). Accuracy assessment of digital elevation models by means of robust statistical methods. ISPRS Journal of Photogrammetry and Remote Sensing, 64(4), 398–406. https://doi.org/10.1016/j.isprsjprs.2009.02.003

Koblet, T., Gärtner-Roer, I., Zemp, M., Jansson, P., Thee, P., Haeberli, W., & Holmlund, P. (2010). Reanalysis of multi-temporal aerial images of Storglaciären, Sweden (1959–99) – Part 1: Determination of length, area, and volume changes. The Cryosphere, 4(3), 333–343. https://doi.org/10.5194/tc-4-333-2010

McNabb, R., Nuth, C., Kääb, A., & Girod, L. (2019). Sensitivity of glacier volume change estimation to DEM void interpolation. The Cryosphere, 13(3), 895–910. https://doi.org/10.5194/tc-13-895-2019

Nuth, C., & Kääb, A. (2011). Co-registration and bias corrections of satellite elevation data sets for quantifying glacier thickness change. The Cryosphere, 5(1), 271–290. https://doi.org/10.5194/tc-5-271-2011

Zhang, G., Kang, S., Fujita, K., Huintjes, E., Xu, J., Yamazaki, T., Haginoya, S., Wei, Y., Scherer, D., Schneider, C., & Yao, T. (2013). Energy and mass balance of Zhadang glacier surface, central Tibetan Plateau. Journal of Glaciology, 59(213), 137–148. https://doi.org/10.3189/2013JoG12J152

Zhu, M., Thompson, L. G., Yao, T., Jin, S., Yang, W., Xiang, Y., & Zhao, H. (2023). Opposite mass balance variations between glaciers in western Tibet and the western Tien Shan. Global and Planetary Change, 220, 103997. https://doi.org/10.1016/j.gloplacha.2022.103997

Zhu, M., Yao, T., Yang, W., Xu, B., Wu, G., Wang, X., & Xie, Y. (2018). Reconstruction of the mass balance of Muztag Ata No. 15 glacier, eastern Pamir, and its climatic drivers. Journal of Glaciology, 64(244), 259–274. https://doi.org/10.1017/jog.2018.16

---

## Author Response (AR1)

**Instituto Argentino de Nivología,**

**Glaciología y Ciencias Ambientales**

[Figure]

Av. Dr. Adrián Ruiz Leal s/n - Parque Gral. San Martín - Mendoza – Argentina
Domicilio Postal: C.C. 330, (5500) Mendoza, Argentina
Tel. 54 261 524 4200 – Fax: 54 261 524 4201
E-mail: ianigla@mendoza-conicet.gob.ar   http://www.mendoza-
conicet.gob.ar/institutos/ianigla

Dear Regula Frauenfelder
Handling Editor
The Cryosphere
Manuscript #TC-2022-264

Please find enclosed a revised version of the manuscript entitled "Annual to seasonal glacier mass balance in High Mountain Asia derived from Pléiades stereo images: examples from the Pamir and the Tibetan Plateau" by Daniel Falaschi, Atanu Bhattacharya, Gregoire Guillet, Lei Huang, Owen King, Kiti Mukherjee, Philipp Rastner, Tandong Yao and Tobias Bolch. The revised file is a marked-up Word document showing all changes requested by the three reviewers and our own additional amends. Below we summarize the most relevant changes that were made to the manuscript in response to the reviewer's major concerns. We further attach to this document our updated point by point response to the three reviewers.

1.      Crucially, the revised manuscript is ~10 % shorter than the original submission. To do this, we followed a reviewer's suggestion and moved the climate analysis to the Supplementary material. We further did a conscious effort to write the manuscript in a more concise way removing unnecessary information and statements. We stress, however, that further reduction of the manuscript length was nevertheless hampered by the inclusion of additional clarifications and discussions requested by the reviewers.

2.      Upon reviewing the artefact-affected elevation change grids, we came to the conclusion that the artefacts were not the consequence of remaining errors in the jitter correction, but were related to the void filling procedure and the fitting of a function to calculate the elevation change gradient. We addressed the issue by following the approach of Gardelle et al. (2013), excluding cells where absolute elevation differences differed by more than three standard deviations from the mean elevation change within each altitude band and calculating a new function. As a result, some mass balance values have changed with respect to the original submission (see Table 3 of the revised document), yet the overall signal of the mass changes and our main conclusions on the accumulation regimes of the investigated glacier remains unchanged.

3.      In the specific case of the Western Nyainqêntanglha 2022 winter elevation change grid, the artefacts were caused by the small number of available elevation change cells in the uppermost glacier area (10 % of the total area only). The original polynomic fit gave too much weight to these negative values, which gave the wrong impression that all of the upper part of the glaciers had significantly thinned at high elevation. When removing the problematic points in the elevation change plot and fitting the new polynomic function, the mass balance value differed by 0.01 m w.e. only when compared to the mass balance estimate using the "standard" approach used in other grids. We thus did not include any additional uncertainty source and considered that the void-filling related uncertainty is contained within the overall one.

4.      As requested by the reviewers, we improved the overall readability and visibility of plot figures using a color-blind friendly palette, thickening lines and enlarging font sizes. We added DEM hillshade images to the elevation change figures, providing some topographical reference to the observed elevation changes.

Sincerely,

Dr. Daniel Falaschi
IANIGLA - CCT CONICET Mendoza
C.C. 330 - 5500 Mendoza, Argentina
E-mail: dfalaschi@mendoza-conicet.gob.ar
Tel: +54 - 261 - 5244262

**Annual to seasonal glacier mass balance in High Mountain Asia derived from Pléiades stereo images: examples from the Pamir and the Tibetan Plateau (tc-2022-264)**

**By Daniel Falaschi et al.**

Reply to anonymous referee#1

**The paper deals with mass inter- and intra- annual ballance of glaciers in High Asia fociusing at two sites one in Eastern Pamirs and the second in the central Tibetan Plateau. It uses Pleiades stereo data for derivation of DSMs which are compared by differencing. A number of corections are applied prior to the differencing. Findings for the two specific sites are presented. Several auxiliary methods are utlized such as classification of snow type from Sentinel-2 data for detection of the accumulation type of gaciers or Glacier Index. The potential of Pleiades data to monitor mountain glaciers is dicussed.**
**The manuscript has high scientific quality and is recomended for publication with minor changes.**
We thank the reviewer for the positive review of the manuscript. Below we provide an inline response to each comment.

**Minor comments:**
**35: Consider using the name "Nyenchen Tanglha Mountains" instead of "Nyainqêntanglha" as it is widely used in the literature. The Chinese version has more than variants for instance "Nyainqentanglha" in Bolch et al. 2010. The English version is unambiguous and appropriate.**
We thank the reviewer for the suggestion. Whilst we understand that the name "Nyainqêntanglha" is well established in the international literature, we have added the English name "Nyenchen Tanglha Mountains" in the study site section for clarification purposes.

**140: Use larger font for the coordinates around on the map frame and remove left and bottom coordinates as they are duplicates.**
Many thanks for the suggestion. We have now removed duplicate coordinates and increased the font size of the grid labels. We have also thickened the glacier outlines and reduced the transparency for better visibility of the figure.

**190: What is the meaning of the "oversampled GSD"? Does this mean that the real spatial resolution is different? For how much?**
With "oversampling" we meant that Pleiades panchromatic scenes are acquired at 0.7 m pixel resolution, yet they are delivered at an (increased) ground sampling distance (GSD) of 0.5 m. Whilst we had originally explained this in the main manuscript, we have now omitted the word "oversampled" for simplicity.

**220: "it can be reduced to a few decimeters after DEM coregistration" This needs a citation.**
The citations for this statement are actually in the previous line. We have amended the text to clarify this.

**205: The sentence starting with "Such is the case of…" is somehow abrupt.**

We have reworded this for improved reading:
"The voids in the September 2021, March 2022 Western Nyainqêntanglha and April 2022 Muztag Ata Pléiades DEMs, account for 20-23% of the glacier area, whilst the remaining DEMs contain less than 9% data voids. "

**210: Table 1: add column Δt for each site to show what are the tine differences. Otherwise, the reader is forced to do an awkward calculation.**
We thank the reviewer for the suggestion. We have included an additional column in the table, showing the elapsed time interval in years between consecutive Pleiades acquisitions over each study site.

**220: You should expand "AMSAG" if it is an abbreviation.**
We have now added the full name (automated snow mapping on glaciers) of the ASMAG tool

**240: three sentences starting with "According to ERA…" are not well understandable. Please reword.**
The manuscript was revised by Dr. Owen King for correct use of English and found no grammatical issues. We stand by our original writing.

**250: What was the source of the SLA?**
The snow line altitude is derived from ASMAG itself. We have now clarified this in the text:
"To remove these misclassifications, we masked out pixels located above the ASMAG-derived mean snowline altitude (SLA) plus 2 standard deviations and reassigned them to the snow class and implemented a low-pass filter. "

**280, 285: This para id cryptic. Please reword it.**
We changed this section of the text:
"To filter this noise, we first implemented a 3-cell buffer around the data gaps and removed the cells within the buffered areas. We then followed the approach of Gardelle et al. (2013), excluding cells where absolute elevation differences differed by more than three standard deviations from the mean elevation change within each altitude band. Finally, we filled the resulting data gaps using the glacier-wide hypsometric approach of McNabb et al. (2019), fitting a fifth-degree polynomial function to the mean elevation change on 50 m elevation bins (Fig. 2c-f)."

**295: "appreciated" to "seen"**
We changed this to "observed".

**295: In Figure 2 the (c) and (d) and (e) and (f) are not localized. Are these subsets at terminus or in the summit part of the glacier? Pleas make this clear.**
The c-f panels depict glacier details that are too small to be annotated in panel a-b. We have nevertheless clarified the general location of these sites in the figure caption.

**295: Figure 2 caption: "in the in the Muztag.."**
Thanks for noting this typing error. We have now corrected the text.

**305: the sentence "According to the authors" should be merged with the previous sentence.**

We have rephrased and merged these two sentences for better reading:

"With no field-surveyed snow density measurements contemporary to our surveyed time periods available, we used the 410 ±60 kg m-3 snow density value retrieved from snow pits in Muztag Ata N15 (Zhu et al., 2018a)."

**385: You can remove the sentence "Alternatively Belart et al…." as it is not needed.**

Although the dynamic considerations was moved to the supplementary material, we opted to keep this sentence, as it gives the necessary background on why we have neglected ice dynamics on the firn densification approach and the mass balance calculation overall.

**425: Why not using the classification of glaciers on the Tibetan Plateau based on the accumulation regime by Maussion et al. 2014? Maussion, F., Scherer, D., Mölg, T., Collier, E., Curio, J., & Finkelnburg, R. (2014). Precipitation seasonality and variability over the Tibetan Plateau as resolved by the High Asia Reanalysis. Journal of Climate, 27(5), 1910-1927.**

We thank the reviewer for the suggestion and agree that in principle, the glacier accumulation regime classification of Maussion et al (2014) would be another dataset that could provide further insight to our findings. In the first place, however, this approach is based on the High Asia Reanalysis, which covers the period 2001-2011. On the one hand, Zhu et al., 2018a showed that accumulation on Muztag Ata N15 Glacier varied greatly on a yearly basis, with some years showing either summer or winter accumulation. On the other hand, Huang et al. (2022) found relevant discrepancies in accumulation regimes for some regions across High Mountain Asia (being Eastern Pamir-Muztag Ata a prime example of this mismatch) as derived from gridded reanalysis data (e.g. HARv2) on one side, and the SAR-derived glacier Index (see section 5.2; lines 668-688). With this in mind, we chose the Glacier Index of Huang et al (2022) to follow an approach which would allow us to derive an accumulation regime for our own surveyed years (2019-2022). More so, with the Glacier Index we use a third validation method fully independent from reanalysis data to validate our geodetic estimates.

**650: "in-balance", would it be better to use "balanced" instead?**

Thanks for the suggestion, we have changed this to "balanced".

[Figure]

[Figure]

Annual to seasonal glacier mass balance in High Mountain Asia derived from

Pléiades stereo images: examples from the Pamir and the Tibetan Plateau (tc-2022-264)

By Daniel Falaschi et al.

Reply to anonymous referee#2

 **Specific comments:**

**Belart et al.,2017 states that the bulk snow density is most likely the largest contributor to uncertainty in winter geodetic mass balance. In the chapter "3.1.5 Bulk density" in the manuscript, the authors refer to other studies and assumes uncertainty values, e.g. that the density values referred to are from snow pits. It is limited data of snow density in the study regions. The authors should discuss this uncertainty more and be clearer on the consequences it might have for the results.**

We briefly expanded on the use of a higher snow density using density values fond in the literature. We found that this approach reduces the differences between the accumulated vs. added mass budgets in the triangulation tests:

"In comparison with end of summer snow densities retrieved from glaciers in other mountain regions (e.g. Pelto et al., 2019; Beraud et al., 2023), our snow density of 410 kg m-3, derived from in-situ surveys (Zhu et al., 2018a) is somewhat lower. Using a snow density of 570 kg m-3 (Pelto et al., 2019), the glacier-wide annual mass balance shifts to -0.21 w.e., +0.24 w.e. and -0.21 w.e. for the years 2020, 2021 and 2022 in Muztag Ata, and -0.89 w.e., -0.71 w.e. and -0.85 w.e. in Western Nyainqêntanglha. This approach, or the use of a 3-year weighted density as in Huss (2013) tend to reduce the differences between the accumulated vs. added mass budgets."

**The use of the surface classification of snow, ice and firn from Sentinel-2 can be clarified in chapter "3.1.2 Classification of snow and ice using Sentinel-2 scenes". The chapter gives a good description of how the analysis is done, but it can be elaborated in the start of the chapter what use these data has for the geodetic glacier mass balance and why it is important.**

We have added a brief sentence (L225-226) about the use of the surface classification maps in the geodetic method, so as not to repeat methodological details later found in section 3.1.2.:

"We generated masks that represent the distribution and density of these components on glacier surfaces to convert volume to mass changes (see Sect. 3.1.6)."

**Some of the sentences and text includes to many parentheses with additional information or clarifications. A suggestion is to go through the whole manuscript text in general and write shorter sentences that are clearer and easier to read. Here is an example to illustrate: L: 595: "Overall, we find the glacier-wide and (for the most part) individual differences to be well within the uncertainty ranges, and attribute the differences to the overall small differences in average density (which in turn derives from the snow and ice distribution) of the September 2021 (590 kg m-3) and April 2022 Sentinel-2 (630 kg m-3) snow and ice masks.". The parentheses are sometimes randomly placed, e.g., "(630 kg m-3)" should maybe be placed after "April 2022"?**

We appreciate the reviewer's concern for overall clarity and reading fluidity of the manuscript. As

suggested, we have gone throughout the manuscript and have made a conscious effort to remove unnecessary or redundant parenthesis and brackets, and split long sentences into shorter ones for improved reading. Among these amends are the specific lines indicated by the reviewer above.

**The authors give a good overview of how they used the Glacier Index of Huang et al. (2022), to find glacier accumulation regimes. However, it is no error estimations of the retrieval of firn and wet snow areas from the remote sensing data, and this should be elaborated.**

We thank the reviewer for this suggestion. We have now included a new paragraph on the error assessment and correction. Citing the supplementary material of Huang et al. (2022), *"…the firn area ratio can be overestimated for two surface types: (1) debris covered by very thin snow which may be taken as clean ice/snow on optical satellite image, but the snow can be penetrated by SAR; (2) ice crevasses, which may form a corner reflector effect to the SAR satellite. Both cases will cause high SAR backscatter coefficients regardless of the season, and lead to misclassification as firn instead of debris and ice. To account for these effects we assume that the pixels higher than -6 dB on both winter and late summer SAR images are misclassified as firn, and the pixels that are lower than -14 dB on both winter and summer SAR images are misclassified as wet snow pixels, and correct these the surface type accordingly."* We have rewritten this information in the manuscript but still refer to the supplementary material for full details.

**Technical comments:**

**L: 91: "The major aims of this paper are therefore to investigate the potential and limitations of geodetic mass balance estimates derived from VHR Pleiades satellite data (using 5 DEMs over the 3-year period 2020-2022)." Should it be data between 2019 and 2022? (Ref. table 1).**

Whilst we reckon that our earliest dataset stem from 2019, in the study we actually assess the geodetic mass balance of three hydrological years (2020, 2021 and 2022). The hydrological year 2020 starts in 2019, so this issue is a bit tricky. We have opted to leave all references to the study period as 2020-2022 for simplicity.

**L:425-429: "To account for different glacier areas between the study sites, we express firn and wet snow areas on each region as a fraction of the total glacier area (hereafter referred to as firn area ratio and wet snow area ratio). This ratio can vary to a great extent across different geographic regions through time, whilst interannual variations of the firn area ratio remain relatively small." Which ratio do the authors refer to when compared to the firn area ratio? Glacier index, I? A suggestion to rewrite sentences.**

Many thanks for the observation. Here we simply compare the interannual variability of the wet snow area ratio and the firn area ratio. We have now amended this for clarification.

**L: 440: "First, the Landsat scenes are used to recognize debris-covered and debris-free areas (ice and snow) on glaciers surface applying a threshold to the previously computed Normalized Difference Snow Index [NDSI] (Bruns et al., 2014).". The authors describe the use of Landsat-data for glacier higher spatial resolution?**

This is a valid question. In principle, Sentinel-2 scenes would also be a good choice to identify debris-covered and debris-free areas on glaciers. On Google Earth Engine, the highest resolution of the Sentinel-1 SAR images is 10 m. However, in the methodology developed by Huang et al (2022), the spatial resolution of the Sentinel-1 scenes is resampled to 30 m to reduce the speckle effect on SAR images, which may affect the firn and snow identification. The method thus uses 30 m resolution Landsat images, matching the resampled SAR images, to identify remove debris.

**L: 830: "The ever increasing and availability of very-high resolution optical satellites (with stereo capability and relatively short revisit time) will allow for increasing the number of glaciers in isolated regions that can be readily monitored.". Can the authors clarify which satellite sensors they are referring to in the last sentence of the conclusion? It is not planned many optical missions with stereo capability in the future. Consider to be more specific and give examples of missions you refer to.**

Good point. We have now added a brief list of VHR satellites with stereo capability (Pleiades, WorldView 1-2, SPOT6-7) that are currently operating, whilst we also listed ALOS-3, which was recently launched.

**L: 1015: Wrong year in Huang et al., 2022 in reference list.**

Many thanks for noting this typo. We have now amended this.

**Figure 1: It is not clear to me which glaciers are "investigated glacier" in the figures. Is it all of them? Consider changing color or outline and rewrite to "investigated glaciers".**

We have thickened the glacier outlines and reduced their transparency for a better visualization. We have also eliminated duplicated coordinates and increased the size of the grid labels in this figure.

**Figure 4 and 6: Cannot really see the dh variation in the figures. A suggestion is to make the figures larger, and subsets of the individual glaciers discussed in the text can also be included.**
**Figure 8: Improve the representation and better the resolution of the plots.**
**Figure 9: It is hard to see the difference between the lines indication "this study" in the plots. Consider changing color on either "annual time step" or the individual glaciers.**

Many thanks for the suggestion. We have now modified all figures, using thicker lines, different line types to separate our results from previous studies, a color-blind friendly palette, increased font size and larger scales when possible. In Figures 4 and 6 we also have incorporated hillshade images as background on all panels, to better illustrate the relation between the glacier elevation changes and topography.

**Annual to seasonal glacier mass balance in High Mountain Asia derived from Pléiades stereo images: examples from the Pamir and the Tibetan Plateau**

**Falaschi et al. TC Discussion**
**Review by César Deschamps-Berger**

This article presents a time series of three annual and two seasonal mass balances for two glacerized massifs in High Mountain Asia. In line with previous studies, it is found that the Pamir glaciers have a mass balance close to equilibrium between 2019 and 2022 while the glaciers from the Tibetan Plateau have a negative mass balance. Various satellite products (Pléiades, Sentinel 1 and 2) were combined to identify the surface elevation change and the accumulation and ablation areas. The snow/firn density and the firn densification are taken into account based on field measurements. The authors put the measured mass balance in longer time scale context and improve the description of the glacier accumulation regimes. It is a detailed work which combines advanced methodologies with good knowledge of the region. I think this article will be a valuable contribution after the following main concerns are addressed.

**1. The winter elevation changes show disturbing patterns (Figure 4, 6 and blue lines in Figure 5). In Western Nyainqêntanglha, elevation loss in winter are stronger at the highest elevation. On the contrary, elevation loss in winter are measured in the Muztag Ata between 6000 m asl and 7000 m asl with areas of elevation gain below and above. Are these elevation change significant or within the calculated uncertainties? What process could explain such altitudinal distribution of elevation change? I wonder if it could be related to remaining errors in the elevation change map (jitter correction, gap filling, shaded areas, see areas highlighted below). Jitter correction might not be perfect due to the lack of stable terrain, especially for Muztag Ata. Better highlighting and explaining these errors might impact the conclusions on the accumulation regimes (e.g. L496, 5.2) and on the sources of uncertainty of the mass balance estimation (e.g. L595-601 and 5.1.2). Left: Figure 4. c. Muztag Ata; Right: Figure 6.c. Western Nyainqêntanglha,**

5.      We are grateful to the reviewer for dwelling in the detailed analysis of the elevation change grids. We have carefully inspected all elevation change tiles and actually detected more problematic maps than those in the review Figure. We introduced further methodological considerations to processing of the elevation change grids. As a result, some mass balance estimates have changed, though the overall signal of the mass changes and our main conclusions on the accumulation regimes of the investigated glacier remains unchanged.

"We then followed the approach of Gardelle et al. (2013), excluding cells where absolute elevation differences differed by more than three standard deviations from the mean elevation change within each altitude band. Finally, we filled the resulting data gaps using the glacier-wide hypsometric approach of McNabb et al. (2019), fitting a fifth-degree polynomial function to the mean elevation change on 50 m elevation bins (Fig. 2c-f). In the specific case of the Western Nyainqêntanglha 2022 winter elevation change grid, data voids concentrated almost exclusively in the uppermost reaches of the sampled glaciers. At similar elevations, the relatively few valid cells available correspond to crevasse movement and show highly negative elevation changes. The glacier-wide void-filling polynomial fitting thus gives too much weight to these negative values and gives the wrong impression that all of the upper part of the glaciers has significantly thinned at high elevation. Therefore, we removed the few highly negative elevation change values in the uppermost reaches of the elevation change plot and adjusted a new function. In doing so, we resampled the elevation bins to 25 m, so that sufficient bins are available for a representative fit".

**2. I estimate that this manuscript would result in an article of more than 20 pages. The article readability would benefit from being more concise. I would advise the authors to revise the manuscript with that in mind. Some of my minors comment should help to gain space (e.g. L89, L108, L125, L194, L283…). If a radical choice was to be made, moving to supplement or removing the parts about the correlation of mass balance with climatological variables might not alter the value of the article. It is almost a distinct topic from the core of the article (see title). The data used partly come from other articles and finally, few (or no) significant correlation are found.**

6.      We appreciate the reviewer's concern for the overall readability of the manuscript. The revised manuscript is now ~10 % shorter than the original submission. In the end, we decided to move the climate analysis to the Supplementary material. We further did a conscious effort to write the manuscript in a more concise way removing unnecessary information and statements, including specific sections pointed out by the reviewer. We stress, however, that further reduction of the manuscript length was nevertheless hampered by the inclusion of additional clarifications and discussions requested by the reviewers.

**3. The quality of the figures should be improved. Almost all the plots with lines are hard to read due to the style and colour of the lines. For instance, it is very hard to distinguish several lines of Figure 5, the individual glaciers in Figure 7, Mass balance from Solid precipitation in Figure 8. Select better colour and line style.**

Based on these comments and the critical review of additional reviewers, we have introduced changes to all figures to improve visibility. Specifically, in Fig 5 we have increased the font size, increased the line thickness and changed one of the colours. We have incorporated colo-blond palettes for plots. In Fig 7 we have increased the font size and line thickness. In Fig 8 we have changed the color from solid precipitation and separated the results from this study against previous ones. In Fig 9 we have modified the figure by changing the color of the annual time step lines for an improved visualization. Please see the specific replies to Figure suggestions toward the end of this document. Since we do not know the final sizes of the figures at this stage, we are happy to introduce further modifications to the figures in the production stage shall this manuscript be accepted for publication.

**Minor comments and suggestions**
**L26. Pléiades. Throughout the text.**
Corrected accordingly

**L31. delete « previously observed »**
If this clarification is removed, then it would seem that the mass balance records from the last 6 decades was generated in this study, which is clearly not. We opted to keep this in the text.

**L32-33. delete « *on average* ». « *mean*» is already stated at the beginning of the sentence.**
Corrected accordingly

**L33. « *increased* » compared to what?**
Good point. We added "to the previous ~6 decades" (which is the period covered in Battacharya et al., 2021))

**L33. Why is Western Nyainqêntanglha qualified here as summer accumulation type when summer mass balance (-0.66 m w.e.) is more negative than the winter one (-0.04 m w.e.)? Besides, this conclusion (Western Nyainqêntanglha being summer accumulation type) seems based on other studies in the dedicated Discussion paragraph (L696-698). « *The 2022 winter (+0.21 ±0.24 m w.e.) and summer (-0.31 ±0.15 m w.e.) mass budgets in Muztag Ata and Western Nyainqêntanglha (-0.04 ±0.27 m w.e. [winter]; -0.66 ±0.07 m w.e. [summer]) suggest winter and summer accumulation-type regimes, respectively.* » I**

**suggest rephrasing as: « The seasonal mass balance in Muztag Ata (winter: XX m w.e., summer : XX m w.e.) and Western Nyainqêntanglha (winter: XX m w.e., summer : XX m w.e.) suggest... ».**

We acknowledge that W.Nyainqentanglha is actually located in a transition area between the monsoon-dominated glaciers in the SE Tibetan Plateau and Himalaya and the westerlies-dominated glaciers to the Northwest (see e.g. the recent paper by Zhu et al., 2023 in GPC). The newly calculated winter mass balance estimate (-0.03 m w.e.) in W. Nyainqentnglha does not show elevation gains in the uppermost reaches of the glaciers. Yet, the summer elevation change panel in Figure 6 (see also Figure 5) does indeed show elevation change that can be attributed to accumulation, which is not the case in the winter panel. Zhang et al (2013) clearly show that in W Nyainqentnglha, accumulation trough precipitation is higher in summer than in winter, but contemporary mass losses though runoff and evaporation can exceed mass gains and lead to a strongly negative mass balance. We interpret this as the reasoning for the observed mass balance in our study.

**L80. Not only WorldView-2 (see Shean et al., 2020). Simply put « WorldView ».**
Changed accordingly

**L86. Deschamps-Berger.:)**
Many thanks for noting; corrected accordingly.

**L89. «*(e.g. Ice, Cloud and land Elevation Satellite-2 -ICESat-2)*» ICESat-2 is not further used. Give only the acronym.**
Changed accordingly

**L92. Quit the brackets.**
In general, and following the suggestion by another reviewer, we have gone throughout the manuscript and removed redundant and unnecessary brackets for an improved readability.

**L96. «*displayed dissimilar mass change rates.*» precise over which epoch, otherwise it sounds like an article's result is given away.**
We added the 6 decade period

**L97. « *mass balance for longer period* » Longer than what?**
We have rephrased this part of the text for clarity.

**L104. In 2.1., provide the max elevation as in 2.2. Tell if the whole massif is covered.**
We have rephrased this to incorporate the maximum elevation of Mount Muztag Ata.

**L108. Delete «*(along with the nearby Kongur Shan mountains).*» It is never mentioned again.**
Removed accordingly

**L124. «*glacier ice*»**
We are unsure about what this particular correction is, Can you please clarify?

**L125. «*Glaciers in the arid NW Tibetan Plateau are predominantly continental-type, cold–based (with their basal part entirely below the pressure melting point) and receive little precipitation.*» Might be deleted? All these informations are repeated in the next lines.**
Good point. Deleted accordingly

**L131-134. Give periods for each mass balance epoch, confusing otherwise.**

We added the year 2009 and 2019 as the start and end (according to the literature) of the slight mass loss period in Muztag Ata.

**L147. « *~230 km in length reaches* » => « extend over ~230 km in length and reaches... »**
Corrected accordingly

**L163. « *in their model* »? An energy balance model is not a source of information about precipitation.**
Corrected accordingly

**L165. « > » => « more than ». Delete « here ».**
Corrected accordingly

**L168. Why use []? And why « accelerated »? compared to what?**
We rephrased the sentence and included time intervals to clarify the accelerating rate of glacier mass loss:
"In contrast to the Muztag Ata massif, glaciers in the Western Nyainqêntanglha region have been rapidly shrinking (27% between 1970-2014; Wu et al., 2016) and losing mass at an accelerating pace (from -0.27 ±0.11 m w.e a-1 between 1968 and 1976 to -0.47 m w.e a-1 between 2012 and 2018; Zhang and Zhang, 2017; Luo et al., 2020; Ren et al., 2020; Bhattacharya et al., 2021)."

**L170. Keep giving MB with two digits precision « (–1.0X m) ».**
We thank the reviewer for the suggestion. We amended the manuscript using two digits precision.

**L172. « *Zhadang glacier*» first time it is mentioned. Introduce shortly the glaciers of interest (size, specificities).**
Many thanks for the suggestion. We have introduced Zhadang glacier and included it in Figure 1.

**L192. Delete «*relatively*».**
Deleted accordingly

**194. « *separated in time on most occasions (Table 1). In all cases, partial acquisition dates were no more than 2 weeks apart.* » => « separated by two weeks in the worst case (Table 1).»**
Corrected accordingly

**L195. Was the DEM produced in one run from the raw stereo images to a high-resolution DEM? A common practice to reduce errors is to first project the images on a low-resolution DEM (idea for future work).**
Yes, the DEMs were generated in one run from the raw stereo images as described. Whilst generating the DEMs using the suggested approach again might be an excessive amount of work for this study, we thank the reviewer very much for the methodological tip for future work!

**L196. « *implementing* » => « using »**
Corrected accordingly

**L197. Why «although»? The first part of the sentence refers to a method of this work, the second refers to results from other studies. Grammatically hard to understand.**
Corrected for clarity

**L200. « *Pléiades DEMs are currently amongst the most common very high resolution DEMs used in***

*geodetic mass balance assessments***». Maybe not necessary as there are anyway few high resolution
photogrammetric satellite and studies exists with WorldView (Shean et al., 2020).**
Removed accordingly

**L203. «** *beyond the higher spatial resolution,* **» not true for WorldView satellites, could be deleted.**
Deleted accordingly

**L205. «***saturated areas»* **=> « areas prone to saturation »**
Corrected accordingly

**L209. Did you request that the images were acquired with reduced Time Domain Integration (TDI)? It can
help preventing saturation (Deschamps-Berger et al., 2020).**
No, we did not require this option to be active during our acquisitions, but many thanks for the suggestion.
We have now two sources that recommend this specific setting for the next acquisitions. Upon reviewing
the AIRBUS image acquisition request form, however, we noticed that this is not explicitly shown as an
available option. We understand that only by adding it as an additional comment may work.

**L223. I understand that the lower resolution saves computational time but less the number of images. Is
the opening and closing of the images really a bottleneck in the treatment?**
We removed the statement as the number of images is not the actual bottleneck in the ASMAG processing,
but stress here that the current version can incorporate Landsat and Sentinel-2 scenes and has not been
adapted for Pleiades images yet.

**L232. delete «** *see also* **»**
Deleted accordingly

**L235. «***To this ends, it implements an automatic threshold to the near-Infrared»* **=> « It determines
automatically a threshold for the NIR band values ».**
Corrected accordingly

**L241. Move the « Muñoz » citation after « daily data ».**
Changed accordingly

**L257. Delete « see e.g. ».**
Deleted accordingly

**L260. Move the UTM info somewhere else more generic.**
We fully deleted this part of the sentence, since it is of little relevance really.

**L265. In future work, you might want to first co-register your reference DEM to an external reference
(e.g. Copernicus DEM) to ensure a better absolute co-registration.**
Good point. Many thanks for the tip for future work!

**L269. «***reprocessed 2019 Pléiades DEM from Bhattacharya et al. (2021)»* **is confusing. Was the 2019 DEM
eventually calculated like the others of this study? Then, the Bhattacharya reference could be deleted.
Maybe, the link between this study and Bhattacharya et al. (2021) should be better explained in
introduction. At least better introduce the Bhattacharya et al. (2021) study as some products are used
here (L 472).**

This is a valid question and the writing was indeed confusing. The 2019 DEM from Bhattacharya et al were originally processed using commercial software. Our 2019 DEMs (to which all later DEMs were coregistered) were newly generated using the Ames Stereo Pipeline as written in the manuscript. We have removed this statement to avoid any confusion.

**L270. Which metric is used to correct the vertical biases? The mean, the median?**
As per Nuth and Kääb (2011), the metric used to correct biases is the mean value on stable terrain. During the preparation of the study, we also tried DEM coregistration using the algorithm of Berthier et al. (2007, RSE), which we run using the median value of the elevation changes on stable terrain. This provided coregistration results of lower quality compared to Nuth and Kääb (2011).

**L271. Delete «** *(reference)* **»?**
Deleted accordingly

**L275. «***implemented***» to be clarified. It sounds like you implemented the code (i.e. wrote the code). However in the acknowledgement the tools of Etienne Berthier are mentioned. Also note that it is a different method than the one used in Deschamps-Berger et al. (2020). The first one fits polynomial functions to the residual while the second calculates and modifies the Fourrier transform spectrum of the residual. Make sure which one was used or implemented.**
We greatly appreciate the reviewer's insight into these methodological differences. Indeed, we used the tools of Dr. Berthier, which uses a spline fit to remove the bias in the along-track direction as in Falaschi et al 2023)

**L277. «** *on- and* **» missing blank**
Corrected accordingly

**L283. «** *We chose a 3-cell buffer so as not to remove valid cells from the original dDEM grids.* **» This kind of sentence could be deleted to make the article more concise.**
Removed accordingly

**L284. «** *mosaiced***»? merged? How are managed areas where there is overlap between tiles (concisely)?**
From our understanding, the correct term should be *mosaiced* as these are raster files and this is the GIS operation that we have used. For the overlapping areas, we opted to use the average value of each grid elevation change. Void filling and firn densification correction were carried out on each elevation change individually, before final mosaicking. Finally, since we applied a seasonality correction to adjust each grid to the hydrological year, using the mean elevation change on overlapping areas seems a technically sounding approach. We consider that any uncertainty owed to this should be within the overall elevation change uncertainty.

**L305. «** *in the energy balance model comprising Muztag Ata N15 and Zhadang glaciers (in Muztag Ata and Western Nyainqêntanglha districts, respectively) by Zhu et al. (2018a). According to the authors, this density value was retrieved from snow pits.***» Maybe no need to repeat the districts if the glaciers are introduced before. Why mention the energy balance model if the density actually come from pits measurements? Merge sentences concisely.**
We removed the district reference. We amended this part of the text, to avoid mentioning energy mass balance, but still pointing at the in-situ surveys mentioned in Zhu et al (2018a)

**L318. « time interval » dt might be more clear than *t*?**
Changed to dt

**L321. Eqn (2) « k » is missing. Replace «i»?**
Thanks for noting. We discarded k and kept i in the equation instead.

**L326. It is not clear what the « i » of Ai refers to? Ice? The previous sentence mentions ice and snow areas.**
We removed the "i" in Eqn3. . By using the i, we originally intended to make it clear that we considered the accumulation and ablation areas independently for the mass balance calculation.

**L330. I cannot find easily in this paragraph if the correction is applied on a pixel scale or at the firn area scale? Please clarify.**
Good question. We applied the correction on a pixel scale.

**L346-347 () [] to homogenise**
Corrected accordingly

**L346. «*most recent elevation grids available* » => « most recent elevation change grids »? Since period are provided in brackets (2013-2019, 2018-2019).**
Yes, we corrected this accordingly

**L374. « *is not to be expected* » => « is not expected ».**
Corrected accordingly

**L395. what does « *addressed* » mean? Calculated, defined?**
Changed to defined

**L397. By construction, the mean elevation difference over stable terrain is zero or close to it. Depending whether the mean or median elevation residual was used for vertical coregistration (to be added in 3.1.4). This underestimate potential systematic error. Besides, I do not find sigma_sys further in the uncertainty calculations.**
In our study we have assessed both systematic and random uncertainties. Systematic and random uncertainties need to be determined separately (see the original reference of Koblet et al, 2010) in the manuscript. Our systematic uncertainty estimates are displayed in Table 2, whilst the corresponding calculation is given in Eqn 5.  As per the Nuth and Kääb (2011), the mean elevation residual on stable terrain is used to perform vertical coregistration. It is our opinion that the relatively small stable terrain areas around our glaciers causes a systematic uncertainty in some of the elevation change grids. In Table 3, the ±values refer to the random uncertainty, which are calculated from Eqn 7.

**L410. Delete « *see also* »**
Deleted accordingly

**L415. Eqn 7. What is « f »?**
Thanks for noting the absence of the definition for $f_{\Delta v}$. $f_{\Delta v}$ **is** the volume to mass conversion factor of ice and snow. We have added this to the text.

**L424. Huang et al. (2022) missing in the bibliography.**
The citation is correct. We simply had written the wrong year in the reference list. Now corrected accordingly.

[Figure]

[Figure]

**L433. The index**
Corrected accordingly

**L433. Please repeat that firn area is measured at the end of the winter and the wet snow area at the end of the summer. Or clarify this point if I misunderstood.**
Added accordingly.

**L449. « the influence of XX to YY » Is this grammatically correct? Otherwise replace «*to govern*» by «on».**
Corrected accordingly

**L452. Put the citations at the end of the sentence. Why mentioning the period of availability of the data? Idem L463.**
We moved the citation and removed the reference to the data availability

**L456. Delete « *either* »?**
Deleted accordingly

**L458. Cite APHRODITE along with ERA5, HARv2 in the previous sentence and delete « (APHRODITE, ERA5, HARv2) » in this one.**
Corrected accordingly

**L459. «*(or instrumental records*» to delete**
Deleted accordingly

**L461. «*Muñoz-Sabater*»**
Corrected accordingly

**L463. « *to monthly time step* » Was ERA5 initially at a daily or monthly time step? Mention it in the previous sentence.**
This is stated in line 19 of the Supplementary material

**L468. «*found*» => « calculated »?**
Corrected accordingly

**L468. Redundancy with « mean » and « average » in the same sentence.**
Corrected accordingly

**L472. « *added the geodetic mass balance values in Bhattacharya et al. (2021)* » see comment about L269. I understand that data from Bhattacharya et al. (2021) are not at a yearly or seasonal resolution. Do you think that mixing periods of different durations could have an impact on the correlation calculated?**
Mixing periods of different durations might certainly affect the correlations. However, we think that the small number of observations overall is the main limiting factor that does not allow to obtain correlation coefficient values that are statistically significative.

**L481. Cite « Table 3 » only once in this paragraph.**
Corrected accordingly

**L483-484. Please provide value in brackets for each year. Or alternatively do not provide any.**
We added the missing values in brackets.

**L486. «** *the (largest and debris-covered) Kekesayi Glacier* **» hard to read. Rephrase without ().**
We removed this, as is it was anyway introduced in the study area sections

**L502. « > » => «above»**
Changed to above

**L510. «** *but interestingly, recovered during the 2022 winter season* **» why is it interesting? It slightly implies that there is a causal relationship between the summer and winter mass balance.**
Removed accordingly

**L546. 2019 does not seem to be an extrema for wet snow area ration in Muztag Ata in Figure 7. Why would it be the most negative mass balance year?**
We did not actually produce geodetic results for the hydrological year 2019. It is actually the hydrological year 2020 which has the most negative mass balance (-0.19 m we), and has also the smallest wet snow area ratio except for the hydrological year 2022. Both years have rather similar Glacier Index values, so we do not necessarily see a contradiction between the Index and the mass balance data.

**L561 «** *departures* **» => anomaly**
To our knowledge, it is perfectly acceptable the use of departures as a surrogate for anomalies. We originally tried to use both terms as not to constantly repeat the same word, but in the end decided to replace with anomaly in the manuscript

**L565 «** *Of the last 16 years, summer air temperatures have experienced positive anomalies.* **» To be deleted? This is expected from a series of anomalies. Maybe the number of summer with positive anomalies is missing.**
Yes! Many thanks for noting. We were missing the number of years with positive anomalies. Now corrected.

**L567. «***rather strong (though not significant***...» It is hard to interpret this results.**
Deleted rather

**L570. «** *respectively* **». Move «** *scale* **» before the brackets.**
Moved accordingly

**L571. «** *likely had an impact* **» => «** *contributed to* **»**
Corrected accordingly

**L571. Is «** *in turn* **» necessary?**
Deleted accordingly

**L587. Cite a study which used this method. One that comes to my mind is Nuth et al. (2013,10.3189/2012JoG11J036) but there must be others.**
Many thanks for the reference.

*L588. «glacier-wide» sounds like a single glacier. Maybe find another term like massif-wide? Check throughout the manuscript.*
To our knowledge, the term "glacier-wide" is fully accepted in the literature to refer to all glaciers or all

glacier ice area. See for example McNabb et al (2019), a study that we reference in the context of usage of the term glacier wide.

**L594. Give respectively, the value for each glacier.**
Do you mean the triangulation residuals for each glacier? It would seem to us an overkill (and of little added value) to provide the residuals for 117 glaciers in W. Nyainqentanglha and 86 glaciers in Muztag Ata, even in a supplementary Table. We have provided the residuals for the full glacier area and illustrate the variability with other 4 glaciers.

**L607. How can two scenarios with the same dh grid but different densities result in the same mass balance for Muztag Ata?**
This was a simple typo since values were very similar but still different. We have corrected this.

**L615. I would rephrase in: « variable density should be used for time spans of 3 years or less. » No conclusion can be drawn on longer time span periods since solely a 3 years period is studied here.**
Agreed, corrected accordingly

**L630. Too long phrase.**
We split the sentence in two shorter ones.

**L633 « *bias* » => errors. Bias often refers to systematic error.**
Corrected accordingly

**L641 « Around our reported biases, » rephrase. Does bias means error?**
Yes, changed to errors.

**L642. Delete +-, the standard deviation is a single value, not a range of uncertainty.**
That's correct, now deleted

**L644. Cite Höhle and Höhle (2009, https://dx.doi.org/10.1016/j.isprsjprs.2009.02.003) along with Dehecq et al. (2016).**
Reference added

**L656. First paragraph of 5.2. It sounds like an introduction paragraph not a discussion one. Maybe only keep the first sentence.**
This is a good point, and a simple mean to shorten the manuscript as pointed out earlier in the review. We kept the first sentence of the paragraph only.

**L666. Too long sentence.**
We rephrased and shortened the sentence.

**L674. « *as* » => that.**
This actually needs to be changed to "at"

**L686. Avoid intricating () and [], please rephrase.**
We have gone throughout the manuscript and have made a conscious effort to remove unnecessary or redundant parenthesis and brackets, and split long sentences into shorter ones for better reading. Among these amends are the specific lines indicated by the reviewer above.

**L688. No undercatch of the weather station is expected ?**
There is a possibility of undercatch like for many other weather stations. The very low precipitation is however, consistent with gridded climate data and also the vegetation in the region. Even if the precipitation would be twice as much there would still be a huge difference to the precipitation at the top of the mountain.

**L691. «** *have allowed us* **» => allows**
Changed to allowed

**L700. «** *too* **»=> also**
Corrected accordingly

**L711. «***Several of the stronger correlations between glacier mass balance and temperature and solid precipitation were not significant,***» Which one were significant ?** Only annual temperatures in Western Nyainqentanglha were significant at the 95% confidence interval. Now corrected in the text.

**L715. What is a «** *relevant development* **»? Not clear.**
Now rephrased for clarity.

**L757.«** *trend* **» sounds like there is a continuous decrease or increase of the mass balance while only two periods are compared. Replace with « period »?**
Good observation, we changed this to period.

**L810. Provide the « period ». Idem as previous comment about the term «trend»**
Not fully clear what is meant here. The periods are included in the manuscript: (-0.50 ±0.17 m w.e. a-1 between 2000 and 2014, Li and Lin., 2017; -0.60 ±0.19 m w.e. a-1 between 2000 and 2017, Ren et al., 2020).

**Figure 1. Top panels: Legend for « investigated glaciers » does not match with the figure. The bottom panels should be merged to show both study sites on the same map. Consider changing the geographical features (topographical map? countries borders ?), the colours and texts. It is blurry and very hard to read.**
We modified the figure, correcting the legend for investigated glaciers. We also merged both location maps into a single one to better illustrate the location of Muztag Ata and W. Nyainqentanglha in High Mountain Asia. The figure inset now features a topographic map. We also added the highest elevations on each study site. We chos not to include country borders based on the fact that this is a region where a number of border conflicts exist between several countries. We still provide an overview of the relative position of the countries.

**Figure 3. Which map is in winter and which one is in summer?**
We added winter and summer labels to the figure

**Figure 4. f is not annual but multi-annual. I would put all annual mass balance on the upper row (move e map to c position).**
Many thanks for the suggestion. We moved the panels as indicated.

**Figure 5. Improve readability, change line colours, increase line width.**
We increased the font size and line thickness.

**Figure 7. Hardly readable. Change the line style and/or the marker style.**
We modified the figure to make the plots larger in size, increased also font size and line thickness, and used different line types for improved visibility.

**Figure 8. Highlight which mass balances comes from this study.**
We distinguished the mass balance and solid precipitation/temperature anomalies stemming from Bhattacharya et al 201 from our own results in the figure using different line styles and colors

**Figure 9. Zhadang line style is too similar to annual time step of this study.**
Many thanks for the suggestion. We used a color-blind friendly palette, and separated our results from previous studies using different line styles We have also increased line thickness and font size.

**Table 1. Consider adding a « Difference (days) » column?**
This was suggested by another reviewer too. We added a column with the time interval (in years) between consecutive Pleiades acquisitions for each portion of the study sites covered by the Pleiades scenes.

**Table 2. What are these values of SE 10-4? Cite Höhle and Höhle (2009) for the NMAD. Would be nice to have the same number of significant digits. SD and SE in full letter in the first line. Caption says on-glacier too but not found.**
We removed the SE column from the Table, as the NMAD and SD metrics are sufficient to describe the statistics of *dDEM* grids off-glacier after coregistration as per the most recent literature on the subject. As suggested in one of the comments, we cited Höhle and Höhle (2009) in line 650 regarding NMAD, and wrote standard deviation and standard error in full letters. We corrected the figure caption, as "on-glacier" values were not included.

---

## Author Response (AR2)

**Instituto Argentino de Nivología,**

**Glaciología y Ciencias Ambientales**

[Figure]

Av. Dr. Adrián Ruiz Leal s/n - Parque Gral. San Martín - Mendoza – Argentina
Domicilio Postal: C.C. 330, (5500) Mendoza, Argentina
Tel. 54 261 524 4200 – Fax: 54 261 524 4201
E-mail: ianigla@mendoza-conicet.gob.ar   http://www.mendoza-conicet.gob.ar/institutos/ianigla

[Figure]

Dear Regula Frauenfelder
Handling Editor
The Cryosphere
Manuscript #TC-2022-264

Thank you very much for your decision. Please find enclosed the second revision of the manuscript entitled "Annual to seasonal glacier mass balance in High Mountain Asia derived from Pléiades stereo images: examples from the Pamir and the Tibetan Plateau" by Daniel Falaschi, Atanu Bhattacharya, Gregoire Guillet, Lei Huang, Owen King, Kiti Mukherjee, Philipp Rastner, Tandong Yao and Tobias Bolch. The revised file is a marked-up document showing all changes, including your editorial comments which we took carefully into account and a number of minor amends to the main text. Please find our response to your specific comments below.
We further provide missing ORCID numbers below for some authors that are missing in the MS record.
Thank you very much for your assistance and we hope to hear back from you soon.

1.  Following up on the respective comment by R#3 and your response, I would still like to encourage you to reconsider your usage of this term and whether it might increase comprehensibility if you would replace it – at least in ambiguous cases – with e.g. the term "mountain-wide".

    **L30 and throughout the manuscript: we are the opinion that the term "mountain-wide" is misleading. As we want to refer to the whole study area in contrast to individual glaciers, we either used the term ¨global mean¨, omitted the term or clarified that we refer to the study regions.**

2.  Following up on the comment by R#3: please rephrase: glacier ice in the context of sqkm is not precise. Either you have xx cubic km of ice (volume) or you have xxx sqkm of glacierized area.

    **L126: We replaced glacier ice with glacierised area**

3.  Formatting in figure panel: - please move "Muztag Ata" to the right, to have the same space between the word and the left coordinate line as the title has in "Western N..."-panel.- In the "WN..."-panel, please move the (b) to the left, in order to have the same space between (b) and the right coordinate line as the (a) has in the "MA"-panel.

    **Figure1: We uploaded a new version with the requested title alignment**

4.  I agree with the comment by R#1 that this passage is not super clear. Even though the sentences are - as you rightly state - grammatically correct, I still suggest some amendments in order to increase comprehensibility.

    **L238-242: The use of the term snowbound was indeed problematic. To clarify this paragraph and the seasonal snow conditions on the 30 September and 6 October 2021 Pléiades image in the Western Nyainqêntanglha, we now state: ¨According to ERA5-Land daily data (Muñoz-Sabater et al., 2021), 3 cm of fresh snow fell between the acquisition of the 30 September and 6 October 2021 Pléiades and Sentinel-2 images in the Western Nyainqêntanglha range. Because seasonal snow is present in the 30 September Pléiades scene, the overall seasonal snow conditions appear nevertheless consistent among the two scenes¨.**

5.  acquisition of the

[Figure]

**L239 Corrected accordingly**

6. This contradicts your response to this point in your author response, where you state that the metric used by Nuth and Kääb is the MEAN. You also state in your response that you have tried the approach by Berthier et al., who use the MEDIAN.

   **L267: You are correct, Nuth and Kääb (2011) use the mean difference in elevation change over stable terrain for coregistration purposes. We have now replaced median with mean in the manuscript.**

7. If this method is originally by Berthier, you should either reference the original paper, or at least rephrase. For example, "... (by E. Berthier, as described e.g. in Falaschi et al., 2023)".

   **L273: After double-checking with Etienne Berthier, we confirm that the bias correction tool using the spline fit was used (or published) for the first time in Falaschi et al. (2023)**

8. Applies to entire table: please center all the numbers in all columns except in the time period column.

   **In Tables 2 and 4 we used a centred alignment as requested.**

9. Even though this will enlarge paper size: please "blow up" this figure to the full-page width, in order to enhance its readibility.

   **The size of Figure 2 was increased to full page width.**

10. Along the line of your response to this point by R#1, please add 1-2 sentences justifying your choice of this method, as opposed to other potentially applicable methods like the one by Maussion et al (2014)

    **L431: We included a brief text in which we justify the selection of the Glacier Index over other available methods to assess glacier accumulation regimes such as Maussion et al. (2014): ¨Although other applicable methods to assess glacier accumulation regimes in High Mountain Asia exist (e.g. Maussion et al., 2014), they are often based on climate reanalyses that have showed large disparities amongst them (Wortmann et al., 2018). More so, with the Glacier Index we intend to use a validation method fully independent from reanalysis data to validate our geodetic estimates¨.**

11. Along the line of your response to this point by R#2, please add 1-2 sentences explaining why you used Landsat imagery and not S-2 imagery.

    **L451: We included a brief text to explain the use of Landsat imagery over Sentinel-2 in the Glacier Index: ¨In the methodology developed by Huang et al (2022), the spatial resolution of the Sentinel-1 scenes is resampled to 30 m to reduce the speckle effect on SAR images, which may affect the firn and snow identification. Whilst 10 m spatial resolution Sentinel-2 images are available from Google Earth Engine, the method uses 30 m resolution Landsat images to match the resampled SAR images¨.**

12. Please mention which authors this applies to by adding their initials (as used in the Author contribution section).

    **L829: TB is the only person from the autor list to be part of TC editorial board.**

13. To thank the two anonymous reviewers as well, would be standard courtesy

    **L840: All three reviewers are now acknowleged accordingly.**

14. Missing ¨the¨

    **L844: Corrected accordingly.**

**ORCID numbers:**

**Daniel Falaschi 0000-0001-9232-2813**

**Lei Huang 0000-0003-1797-5667**

**Owen King 0000-0001-9232-2813**

**Kriti Mukherjee 0000-0001-9232-2813**

[Figure]

[Figure]

**References cited in this response:**

Falaschi, Daniel, Etienne Berthier, Joaquín M. C. Belart, Claudio Bravo, Mariano Castro, Marcelo Durand, and Ricardo Villalba. "Increased Mass Loss of Glaciers in Volcán Domuyo (Argentinian Andes) between 1962 and 2020, Revealed by Aerial Photos and Satellite Stereo Imagery." Journal of Glaciology 69, no. 273 (February 2023): 40–56. https://doi.org/10.1017/jog.2022.43.

Huang, Lei, Regine Hock, Xin Li, Tobias Bolch, Kun Yang, Ninglian Wang, Tandong Yao, Jianmin Zhou, Changyong Dou, and Zhen Li. "Winter Accumulation Drives the Spatial Variations in Glacier Mass Balance in High Mountain Asia." Science Bulletin, August 2022, S2095927322003644. https://doi.org/10.1016/j.scib.2022.08.019.

Maussion, Fabien, Dieter Scherer, Thomas Mölg, Emily Collier, Julia Curio, and Roman Finkelnburg. "Precipitation Seasonality and Variability over the Tibetan Plateau as Resolved by the High Asia Reanalysis*." Journal of Climate 27, no. 5 (March 1, 2014): 1910–27. https://doi.org/10.1175/JCLI-D-13-00282.1.

Muñoz-Sabater, Joaquín, Emanuel Dutra, Anna Agustí-Panareda, Clément Albergel, Gabriele Arduini, Gianpaolo Balsamo, Souhail Boussetta, et al. "ERA5-Land: A State-of-the-Art Global Reanalysis Dataset for Land Applications." Earth System Science Data 13, no. 9 (September 7, 2021): 4349–83. https://doi.org/10.5194/essd-13-4349-2021.

Nuth, Chris, and Andreas Kääb. "Co-Registration and Bias Corrections of Satellite Elevation Data Sets for Quantifying Glacier Thickness Change." The Cryosphere 5, no. 1 (March 29, 2011): 271–90. https://doi.org/10.5194/tc-5-271-2011.

Wortmann, Michel, Tobias Bolch, Christoph Menz, Jiang Tong, and Valentina Krysanova. "Comparison and Correction of High-Mountain Precipitation Data Based on Glacio-Hydrological Modeling in the Tarim River Headwaters (High Asia)." Journal of Hydrometeorology 19, no. 5 (May 1, 2018): 777–801. https://doi.org/10.1175/JHM-D-17-0106.1.

Dr. Daniel Falaschi
IANIGLA CONICET - CCT Mendoza
C.C. 330 - 5500 Mendoza, Argentina
E-mail: dfalaschi@mendoza-conicet.gob.ar

---

## Author Response (AR3)

IANIGLA

CONICET

U. N. C U Y O
GOBIERNO
DE MENDOZA
* * *
Dear Regula Frauenfelder
Handling Editor
The Cryosphere
Manuscript #TC-2022-264

Thank you very much for your decision. Please find enclosed the revised manuscript with your latest round of enquiries. Please find our response to your specific comments below.
Thank you very much for your assistance.

1.  You introduce an abbreviation for High Mountain Asia in the first sentence of the Introduction. However, you don't use it afterwards. Either you have to use this abbreviation in the rest of the text (thereby replacing High Mountain Asia with HMA) or you should delete the introduced abbreviation in the first sentence.

    **L45 We removed the HMA abbreviation as per your suggestion**

2.  Please get rid of the outermost bounding box around the figures. This will also allow you to make the figures a few mm larger each, which will increase their readability.

    **We eliminated the bounding box in Figures 3, 5, 7 and 8 as requested.**

3.  Missing ¨the¨

    **L569 Corrected accordingly.**

4.  Missing ´region (or range)´

    **L569 Here we opted for the term massif, as it is more suitable here in our opinion.**

5.  Remove ´in their detailed assessment´

    **L723 Removed accordingly**

6.  since 1973, with positive values between 2001 and 2009... How can it be "in-balance conditions since 1973" then? Do you mean "mostly in-balance conditions since 1973, however with..."? Please clarify.

    **L723 The mass balance value of +0.03 ±0.10 m w.e. a⁻¹ between 2001 and 2009 can be seen basically as in-balance, though writing ´slightly positive afterwards can lead to confusion in the context of this sentence. To avoid confusion, we rewrote it to ´ …while Bhattacharya et al. (2021) found almost balanced conditions since 1973 (-0.01 ±0.06 m w.e. a⁻¹ between 1973 and 2001 and +0.03 ±0.10 m w.e. a⁻¹ between 2001 and 2009)´.**

7.  Replace ´and´ with ´while´; add missing bracket; replace long dash with short dash

[Figure]

[Figure]

**L723 Corrected accordingly**

Dr. Daniel Falaschi
IANIGLA CONICET - CCT Mendoza
C.C. 330 - 5500 Mendoza, Argentina
E-mail: dfalaschi@mendoza-conicet.gob.ar